# An all-optical multidirectional mechano-sensor inspired by biologically mechano-sensitive hair sensilla

Yuxiang Li[1,6], Zhihe Guo [1,6], Xuyang Zhao [1], Sheng Liu[1], Zhenmin Chen [2], Wen-Fei Dong[3], Shixiang Wang [1] ✉, Yun-Lu Sun[1,4] ✉ & Xiang Wu [1,5] ✉

Mechano-sensitive hair-like sensilla (MSHS) have an ingenious and compact three-dimensional structure and have evolved widely in living organisms to perceive multidirectional mechanical signals. Nearly all MSHS are iontronic or electronic, including their biomimetic counterparts. Here, an all-optical mechano-sensor mimicking MSHS is prototyped and integrated based on a thin-walled glass microbubble as a flexible whispering-gallery-mode resonator. The minimalist integrated device has a good directionality of 32.31 dB in the radial plane of the micro-hair and can detect multidirectional displacements and forces as small as 70 nm and 0.9 μN, respectively. The device can also detect displacements and forces in the axial direction of the micro-hair as small as 2.29 nm and 3.65 μN, respectively, and perceive different vibrations. This mechano-sensor works well as a real-time, directional mechano-sensory whisker in a quadruped cat-type robot, showing its potential for innovative mechano-transduction, artificial perception, and robotics applications.

Optics and photonics for environmental perceptions, namely, artificial sensing, are increasingly used in photonic noses and tongues[1,2], photonic skins[3], and laser-based gyroscopes and LiDAR systems[4,5] for detecting molecules, pressures, spatiotemporal locations, visual information, and other data. Optical sensing devices and systems involve light transductions and have distinct capabilities in up-taking, transforming, and enhancing target signals[6–12]. Compared with electronic and electrical schemes, optical and photonic methods facilitate artificial perception with advantages such as high sensitivity, low loss, anti-electromagnetic interference, noncontact or remote operation, multiplexing capabilities, and spatiotemporal multidimensionality, many of which are challenging to realize with electronic systems[3,13]. In addition to visual perception, mechanical perception capabilities such as tactile and auditory sensing of forces and acoustic vibrations, especially those determining directionality or even vector properties (namely, signal amplitudes or strength and direction), are essential sensory functions of many living organisms[14] and important in artificial systems[15–19].

Organisms in nature have developed mechano-sensitive hair-like sensilla (MSHS), such as the cilia of arthropods, lateral line systems of fish, and whiskers and auditory receptors (e.g., cochlear hair cells) of mammals, to perceive mechanical or acoustic stimuli and information in the surrounding environment[14,20–28] (see more details in Supplementary Note 1 in the Supplementary Information (SI)). Biological MSHS have coincidently evolved with similar three-dimensional (3D) integrated configurations, even for different phyla. They are typically composed of one micro-hair antenna (whiskers, cilia, etc.) to probe and propagate mechanical signals and a network or array of sensory neurons and nerve endings wrapped around the micro-hair terminal that transform mechanical stimuli into electronic, or more precisely, iontronic nervous impulses[15,29,30]. The heterogeneously 3D-integrated structure of biological MSHS is essentially the only single-device configuration that enables multidirectional mechanical sensing of forces, vibrations, etc. Thus, this structure provides a highly evolutionarily optimized prototype reference for the man-made three-dimensional

[1]School of Information Science and Technology, Fudan University, Shanghai, China. [2]Peng Cheng Laboratory (PCL), Shenzhen, China. [3]CAS Key Laboratory of Bio Medical Diagnostics, Suzhou Institute of Biomedical Engineering and Technology, Chinese Academy of Sciences, Suzhou, China. [4]Institute for Stem Cell and Regeneration, Chinese Academy of Sciences, Beijing, China. [5]State Key Laboratory of Photovoltaic Science and Technology, Fudan University, Shanghai, China. [6]These authors contributed equally: Yuxiang Li, Zhihe Guo. ✉e-mail: wangsx@fudan.edu.cn; ylsun@fudan.edu.cn; wuxiang@fudan.edu.cn

or multidimensional integrated precision systems and multidirectional artificial mechano/vibration perception applications.

To our knowledge, almost all the reported MSHS, including biological and biomimetic sensilla, use the classic configuration (i.e., electrode array + micro-hair) to transform mechanical stimuli (strains, vibrations, etc.) into iontronic or electronic signals[15,31,32]. Compared with electrical systems using devices such as capacitors, an optical mechano-sensor micro-hair for innovative mechano-transduction and artificial perceptions applications (even an all-optical multi-perception integrated or fused system) would have advantages due to its simpler configuration, improved anti-interference ability, and high-frequency, multiplexing and multifunctional performance; however, to our knowledge, no such system has been reported. Microbubble structures are effective configurations for mechanical or acoustic transduction[33]. The flexible microstructures and heterogeneously 3D-integrated optics on a glass-microbubble whispering-gallery-mode (WGM) resonator enable this minimalist all-optical configuration for the effective multidirectional mechano/acousto-opto-transductions.

In this work, for the first time to our knowledge, a proof-of-concept prototype of an all-optical multidirectional mechano-sensor micro-hair inspired by biological MSHS is constructed that can perceive external multidirectional displacements, static forces, vibrations, and other information. This biomimetic mechano-sensor is assembled and integrated with three main parts: (i) a single thin-walled glass microbubble as the flexible WGM resonator for mechano-opto-transductions (namely, the hair terminals and tactile sensory neurons); (ii) a glass micro-hair as the flexible mechano-antenna placed on the top center of the microbubble for propagating the mechanical stimuli (namely, the mechano-probing hair); and (iii) a fiber taper for propagating the optical signals with the mechano-stimuli information (namely, the sensory nerves). With a good directionality of 32.31 dB, the sensitivity of the mechano-sensor changes with the direction of the external force within 360° in the radial plane of the micro-hair, which is a unique feature compared to the scalar mechano-sensor. When the direction of the external force and displacement is fixed at $\varphi = 180°$ (that is, the direction with the maximum positive force and displacement sensitivities), the leverage effect results in a maximum displacement sensitivity of 0.052 pm μm$^{-1}$ at $L = 3$ mm, and a maximum force sensitivity of 3.994 pm mN$^{-1}$ at $L = 15$ mm. Correspondingly, their detection limits ($DL$) are calculated to be 70 nm and 0.9 μN, respectively. In the axial direction of the micro-hair, the displacement sensitivity is 1.570 pm μm$^{-1}$ and the force sensitivity is 0.986 pm mN$^{-1}$. Their calculated detection limits in this direction are 2.29 nm and 3.65 μN, respectively. Furthermore, the device can perceive different vibrations in the axial direction and demonstrates the potential for multi-parameter (i.e., displacement and temperature) sensing within a single optical device. As a demonstration, this all-optical multidirectional mechano-sensor mimicking biological MSHS shows good performance as a real-time directional mechano-sensory whisker in a cat-like quadruped robot. The proposed mechano-sensor is potentially useful for constructing all-optical multifunctional perception systems with highly integrated structures and functions, which may share one or more light sources for diverse artificial sensing (vision, gesture, environment, multidirectional tactile sensing, etc.), promoting state-of-the-art technologies such as robotics, augmented reality and virtual reality (AR/VR) and the metaverse.

## Results

### Principles of photonic mechano-sensors

Animals use mechanically sensitive sensory organs, such as whiskers, cilia, and neuromasts, to perceive environmental changes and stimuli, as shown in Fig. 1a–c. For example, the whiskers of cats help perceive the environment and measure distances, as shown in Fig. 1a. The whisker mechanoreceptors, transmission nerves and brain form the tactile system (inset of Fig. 1a). Inspired by the biological structure and

tactile mechanism of these well-evolved mechano-sensory organs, an artificial biomimetic all-optical mechano-sensor and system were developed based on a glass-microbubble WGM resonator.

As schematically illustrated in Fig. 1d, to construct the bioinspired photonic mechano-sensor, a thin-walled glass microbubble with a glass micro-hair placed on its top center is coupled with a fiber taper at the equator of the microbubble. The 3D-integrated structure is embedded within a matrix of ultraviolet (UV)-crosslinked low-refractive-index polymer (MY Polymers, MY-133-V2000). The glass micro-hair (that is, the hair antenna) collects environmental mechanical stimuli such as forces or vibrations and propagates the stimuli to the microbubble. The thin-walled microbubble configuration is mechanically flexible, allowing these environmental mechanical stimuli to be visualized as shifts in the WGM resonance wavelength and light spectral signals (i.e., shifts in the dips). The optical fiber taper evanescently couples light into and out of the microbubble WGM resonator, and transmits the optical signals transformed from the mechanical stimuli to the information decoding system. The fiber taper functions similarly to nerve fibers connecting the MSHS and the central nervous system in the biological archetype. More importantly, when an external force $F_r$ with a component in the plane of the fiber taper and its coupled microbubble equator (or, a component perpendicular to the micro-hair) is applied to the head of the micro-hair, the presence of the fiber taper breaks the mechanical circular symmetry (the stress distribution, deformation, etc.) in the microbubble's equatorial section. Therefore, the strain effect and spectral response of the microbubble are different for $F_r$ applied in different directions, which is the main reason for the directionality of the mechano-sensor. The low-refractive-index polymer acts structurally as the so-called skin layer to elastically fix and protect the microbubble, the root of the hair antenna, and the fiber taper and works optically to implement refractive index matching for the WGM resonance of the laser light. Consequently, a bioinspired mechano-sensor is constructed, as shown in Fig. 1d and e (see more details in the Methods).

The mechano-sensor can detect touch and pressure sensations. The working principle of the photonic mechano-sensor is based on the condition of optical resonance, that is, the optical path length for a photon to complete a round trip in a sphere is an integer multiple of the light wavelength $\lambda$, which can be expressed as follows[34–36]:

$$m\lambda = 2\pi R n_{eff}. \tag{1}$$

where $R$ and $n_{eff}$ represent the equatorial radius and effective refractive index of the microbubble resonator, respectively, and $m$ ($\gg 1$) is an integer representing the azimuth quantum number. The force sensing mechanism is based on the strain effect of the glass microbubble, which has been experimentally demonstrated (see more details in Supplementary Fig. 1 and Table 1 in the SI). Therefore, a force applied to the micro-hair changes the $R$ and $n_{eff}$ values of the microbubble, causing the following shift in the resonance wavelength[37,38]:

$$\frac{d\lambda}{\lambda} = \frac{dR}{R} + \frac{dn_{eff}}{n_{eff}}. \tag{2}$$

Here, $dR/R$ represents the radial strain of the microbubble's equatorial cross-section, and $dn_{eff}/n_{eff}$ is the change in the effective refractive index due to the mechanical stress or strain, which is given by[39]:

$$\frac{dn_{eff}}{n_{eff}} = -\frac{n_{eff}^2}{2}[p_{11}\varepsilon_z + p_{12}(\varepsilon_x + \varepsilon_y)]. \tag{3}$$

where $\varepsilon_x$, $\varepsilon_y$ and $\varepsilon_z$ are the strains along the three coordinate axes for each point in space within the microbubble's equatorial cross-section, as indicated in Supplementary Fig. 2a (SI), and $p_{ij}$ are Pockel's elasto-

**Biological systems of mechano-sensory hairs**

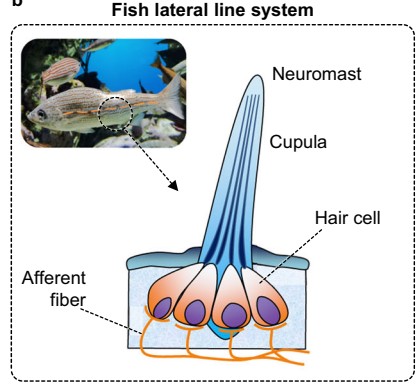

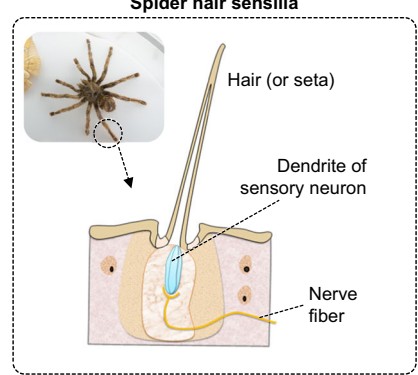

**a** Mammalian whiskers

**b** Fish lateral line system

**c** Spider hair sensilla

**Biomimetic photonic hair mechano-sensors**

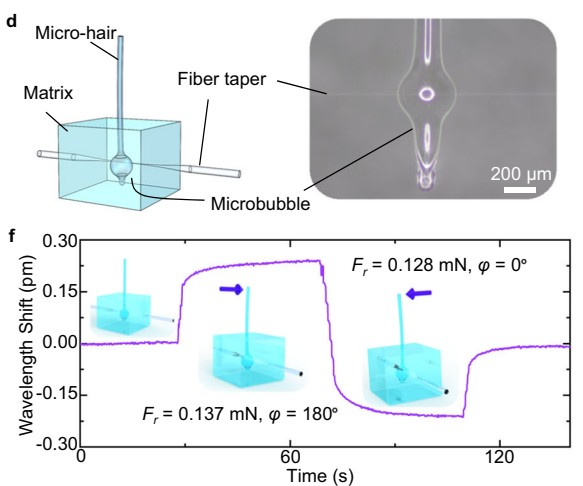

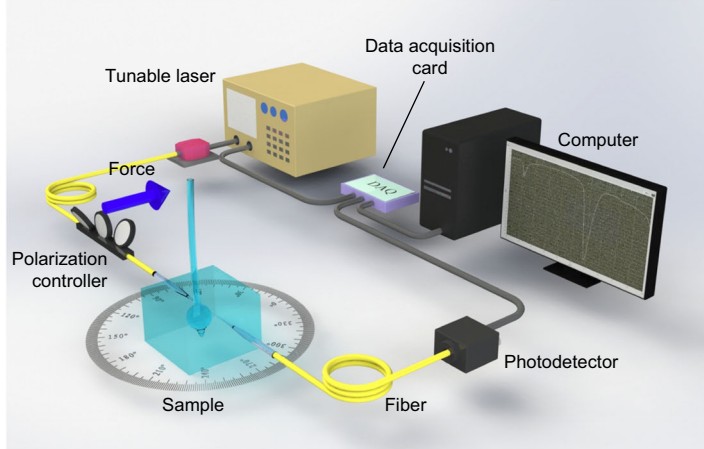

**Fig. 1 | Schematic illustration of the bioinspired mechano-sensor. a** Schematic illustration of the cat whisker sensory system and whisker mechanoreceptor anatomy. **b** Schematic diagram of a fish lateral line system and a neuromast cross-section. **c** Schematic diagram of a spider and the tactile hair protruding from its exoskeleton's surface. **d** Schematic diagram of the mechano-sensor. A microbubble with a micro-hair, similar to the whisker of a cat, is used to detect tactile stimuli, analogous to a sensory receptor. A fiber taper is used to transmit signals, which is similar to signal transmission by nerve fibers. The polymer matrix can fix the microbubble and fiber taper and protect them from environmental damage, which is similar to the function of the skin. The inset shows an image of a microbubble coupled with a fiber taper under a microscope. The scale bar represents 200 μm. **e** Experimental setup of the all-optical mechano-sensor system. The blue arrow indicates the direction of the external force. **f** Responses of the bioinspired mechano-sensor in different deformation states (i.e., $F_r = 0$ mN, 0.137 mN ($\varphi = 180°$), and 0.128 mN ($\varphi = 0°$)).

optic coefficients. The force and displacement sensitivity of the mechano-sensor can be defined as:

$$S_F = \frac{d\lambda}{dF}, S_D = \frac{d\lambda}{dD}. \tag{4}$$

where $F$ and $D$ are the external force and displacement applied to the mechano-sensor micro-hair, respectively.

In a typical test, the resonance wavelength remains constant in the initial vertical state of the mechano-sensor micro-hair ($F_r = 0$ mN). If the mechano-sensor micro-hair is swung or bent in other directions under an external force (i.e., $F_r = 0.137$ mN ($\varphi = 180°$) and 0.128 mN ($\varphi = 0°$), respectively), the resonance wavelength is shifted in the opposite direction due to the strain effect on the microbubble (Fig. 1f).

## Mechanical perception and analysis in the vertical plane of the micro-hair

A finite element method (FEM) model was used to analyze the responses of the mechano-sensor when external forces $F_r$ were applied to the micro-hair. The label $L$ in Fig. 2a represents the length of the

micro-hair subjected to the external forces $F_r$. The direction of the external force $F_r$ applied to the mechano-sensor in the $r$-$\varphi$ plane is shown in Fig. 2a. The elastic modulus $E$ and the Poisson's ratio $v$ of the polymer matrix are set as 5.2 MPa and 0.41, respectively. As a stimulus input in the simulation, an external force $F_r$ (blue arrow) is applied to the micro-hair along the $r$-axis of the mechano-sensor (Fig. 2b). The direction of the external force $F_r$ is set as $\varphi = 180°$, and the corresponding displacement is set as 2 μm. The stress field distribution in the equatorial cross-section of the mechano-sensor is shown in Fig. 2c. The microbubble and fiber taper move in the $\varphi = 180°$ direction. The stress on the microbubble wall is over 2-3 orders of magnitude higher than that on the polymer matrix; therefore, the stress effect of the polymer matrix is neglected. The radial strain field distribution (Fig. 2d) and the strain components (i.e., $\varepsilon_x$, $\varepsilon_y$ and $\varepsilon_z$) along the microbubble's equatorial cross-section can be directly obtained in the FEM simulation. According to Eq. (3), by using the reported values of $p_{11} = 0.131$ and $p_{12} = 0.26$ for fused silica, the field distribution of the strain-induced effective refractive index change is derived, as shown in Fig. 2e. The radial strain field distribution is negative in the range of $\varphi = 90°$ to 270° in the equatorial cross-section but positive in the range

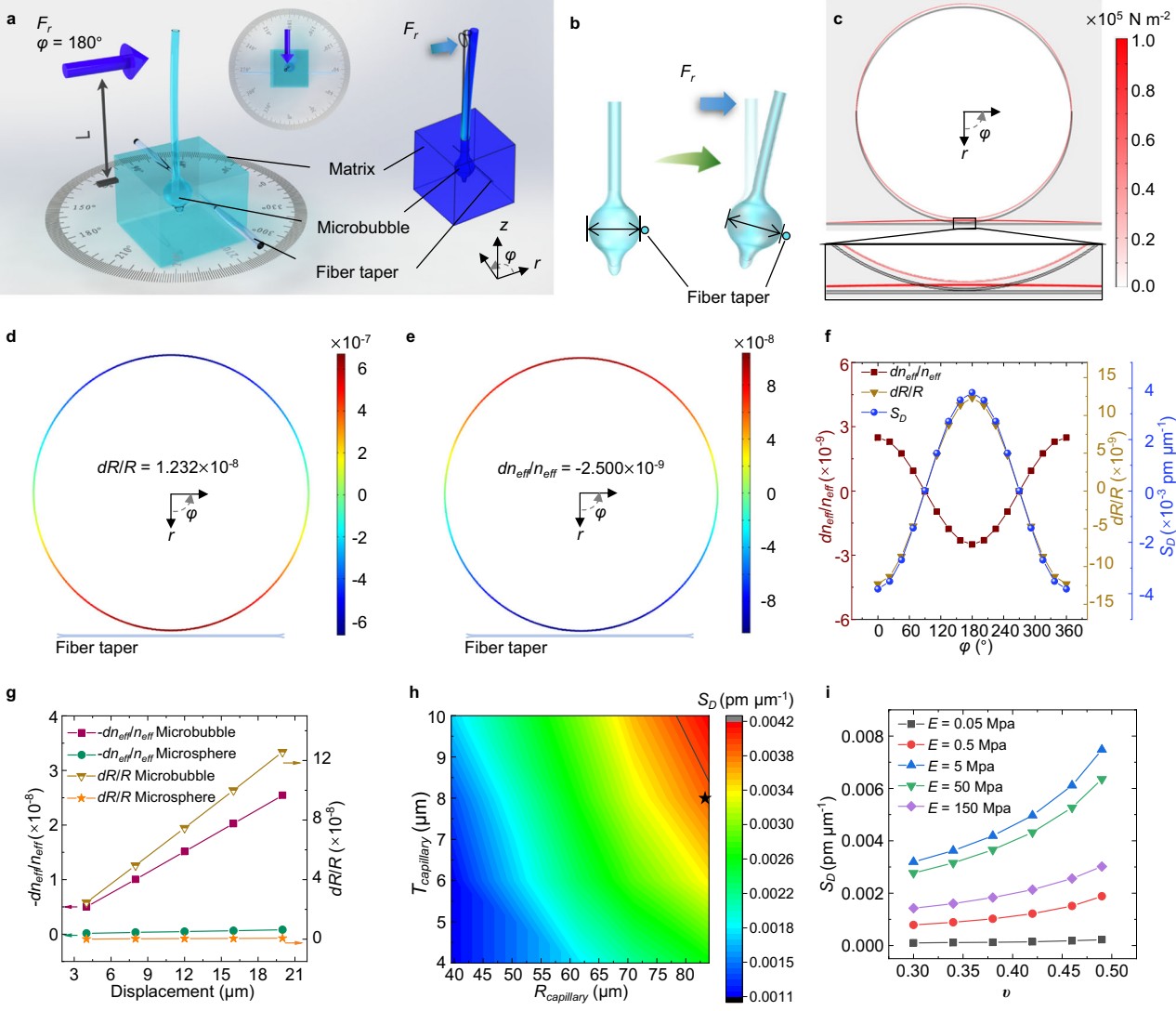

**Fig. 2 | Simulations of the mechano-sensor under an external force $F_r$.**
**a** Schematic illustration of the experimental device and FEM model of the mechano-sensor under an external force $F_r$ ($\varphi = 180°$). Inset: Top view of the mechano-sensor. **b** Schematic illustration of the deformation of the flexible microbubble under an external force $F_r$. Negative and positive radial strains are observed in the same and opposite directions of $F_r$, respectively. **c** Stress field distribution in the $r$-$\varphi$ plane under an external force $F_r$ ($\varphi = 180°$). **d, e** Field distributions of the radial strain and strain-induced effective refractive index change along the microbubble's equatorial cross-section under an external force $F_r$ ($\varphi = 180°$). The $dR/R$ and $dn_{eff}/n_{eff}$ values are the averages of their field distribution integrals. **f** Relations between the direction of

the external force $F_r$ and $dR/R$, $dn_{eff}/n_{eff}$, and the displacement sensitivity ($S_D$). **g** Comparisons of $dR/R$ and $dn_{eff}/n_{eff}$ between the microbubble-based mechano-sensor and the microsphere-based mechano-sensor under different external displacements ($\varphi = 180°$). **h** Relations between $S_D$ and the capillary size (i.e., radius ($R_{capillary}$) and wall thickness ($T_{capillary}$)) under an external force $F_r$ ($\varphi = 180°$). The solid black line corresponds to the contour of $R_{bubble} = 190$ μm. The star label corresponds to the $R_{capillary}$ and $T_{capillary}$ selected for the micro-hair. (**i**) Relations between $S_D$ and the mechanical properties of the polymer matrix (i.e., Poisson's ratio ($\upsilon$) and elastic modulus ($E$)) under an external force $F_r$ ($\varphi = 180°$).

of $\varphi = 270°$ to $450°$ (namely, $90°$) in the equatorial cross-section. The radial strain field distribution along the microbubble's equatorial cross-section has a maximum negative value at $\varphi = 180°$ (that is, the same direction of $F_r$), a maximum positive value at $\varphi = 0°$ (that is, the opposite direction of $F_r$), and values of approximately zero at $\varphi = 90°$ and $270°$ (that is, the direction perpendicular to $F_r$). The radial strain $dR/R$ and the effective refractive index change $dn_{eff}/n_{eff}$ in the microbubble's equatorial cross-section are the averages of their field distribution integrals, which are indicated in Fig. 2d, e, respectively. In theory, for a microbubble with a circular symmetric equatorial cross-section, the field distributions of the radial strain and strain-induced effective refractive index change should be antisymmetric (i.e., $dR/R \approx 0$, $dn_{eff}/n_{eff} \approx 0$), regardless of the direction of the force $F_r$ applied to the micro-hair. However, the presence of the fiber taper breaks the geometrical circular symmetry of the microbubble's

equatorial cross-section, amplifying the radial strain at the coupling position (that is, $\varphi = 0°$ in the microbubble's equatorial cross-section), resulting in $dR/R > 0$. Moreover, $dR/R$ is greater than $dn_{eff}/n_{eff}$, inducing a redshift in the resonance wavelength.

When the direction of the force $F_r$ changes, it can be inferred that the $dR/R$ depends on the radial strain at the coupling position of the microbubble. $dR/R$ has a maximum positive value (that is, the maximum resonance wavelength redshift) when an external force $F_r$ is applied in the $\varphi = 180°$ direction, a maximum negative value (that is, the maximum resonance wavelength blueshift) when an external force $F_r$ is applied in the $\varphi = 0°$ direction, and values of approximately zero (that is, no resonance wavelength shift) when an external force $F_r$ is applied in the $\varphi = 90°$ and $270°$ directions (see more details in Supplementary Fig. 3 and 4 in the SI). The $dR/R$, $dn_{eff}/n_{eff}$, and $S_D$ (displacement sensitivity of the mechano-sensor) vary with the direction

of force $F_r$ (Fig. 2f). $dR/R$ is greater than $dn_{eff}/n_{eff}$ and dominates $d\lambda/\lambda$. As a comparison, the hollow microbubble is replaced with a solid microsphere of the same size; in this case, the $dR/R$ and $dn_{eff}/n_{eff}$ values of the microsphere-based mechano-sensor are much smaller than those of the microbubble-based mechano-sensor, as shown in Fig. 2g. The results in Fig. 2g also indicate that for the displacement ($\varphi = 180°$) range considered, $dR/R$ is larger than $dn_{eff}/n_{eff}$, and the dependence of $d\lambda/\lambda$ on the displacement (i.e., force) is essentially linear.

To obtain a mechano-sensor with high displacement sensitivity, the geometric parameters of the micro-hair and microbubble and the mechanical properties of the polymer matrix were optimized through the FEM scanning parameters (Fig. 2h, i, Supplementary Fig. 5-7 and Table 1 in the SI). The strain effect of microbubble is the main reason for the shift in the resonance wavelength, indicating that reducing the wall thickness of the microbubble can effectively improve the displacement sensitivity of the mechano-sensor. When the wall thickness of the microbubble is fixed at 1.5 μm, the larger the radius and wall thickness of the capillary are, the larger the radius of the fabricated microbubble (Supplementary Fig. 5b, SI), and ultimately, the higher the displacement sensitivity of the mechano-sensor (Fig. 2h). The displacement sensitivity of the mechano-sensor is the highest when the elastic modulus of the polymer encapsulating the mechano-sensor is 5 MPa (Fig. 2i). An increase or decrease in the magnitude of the elastic modulus will reduce the displacement sensitivity of the mechano-sensor, which corresponds to the experimental results in Supplementary Fig. 1. Based on the FEM results and actual fabrication conditions, a fused silica capillary with a radius of 84 μm and a wall thickness of 8 μm was selected as the micro-hair (the star label in Fig. 2h), a microbubble with a radius of 185 μm and a wall thickness of 1.5 μm was fabricated for mechano-opto-transduction (Supplementary Fig. 5d, SI), and MY-133-V2000 polymer with an elastic modulus of

5.2 MPa and a Poisson's ratio of 0.41 was selected as the polymer matrix.

For an external force in the $r$-direction ($r$-$\varphi$ plane), different directional responses of the mechano-sensor can be observed. The quality factor ($Q$ factor) of the tracked WGM is approximately $1.8 \times 10^7$, as shown in Fig. 3a. Notably, a higher $Q$ factor can enhance the sensing resolution due to its narrower spectral linewidth[40]. The force direction indicated by the blue arrow is $\varphi = 180°$, with $L = 9$ mm. The input laser power is maintained as low as 4.16 μW to prevent opto-thermal effects. The corresponding resonance wavelength shift caused by the external force $F_r$ is shown in Fig. 3b and c. The value of the external force $F_r$ is recorded by a commercial force sensor mounted on the stepper motor stage. When the displacement-applying object is driven by a stepper motor with a step value of 60 μm, the full width at half maximum (FWHM) varies intricately (Supplementary Fig. 8a, SI), while the resonance wavelength shift generated by the mechano-sensor increases in a stepwise manner during the displacement application. The displacement sensitivity ($S_D$) and force sensitivity ($S_{Fr}$) of the mechano-sensor are 0.004 pm μm$^{-1}$ and 2.231 pm mN$^{-1}$, respectively (Fig. 3d). Furthermore, the relations between the displacement/force sensitivities at $\varphi = 180°$ and the position of the applied force were determined by changing the action point of the static force $F_r$ (maintaining $\varphi = 180°$), as shown in Fig. 3e. The spring constant ($k$) of the mechano-sensor varies from 61.861 N m$^{-1}$ to 0.412 N m$^{-1}$ as $L$ increases from 3 mm to 15 mm (Supplementary Fig. 8b, SI), indicating that the lever principle is important for customizing the force detection capability of the device. Due to the leverage effect, the displacement sensitivity decreases with increasing $L$, while the force sensitivity increases with increasing $L$. In the experiments, the displacement sensitivity and force sensitivity reach their maximum values of 0.052 pm μm$^{-1}$ and 3.994 pm mN$^{-1}$ when $L$ is 3 mm and 15 mm, respectively. The standard deviation

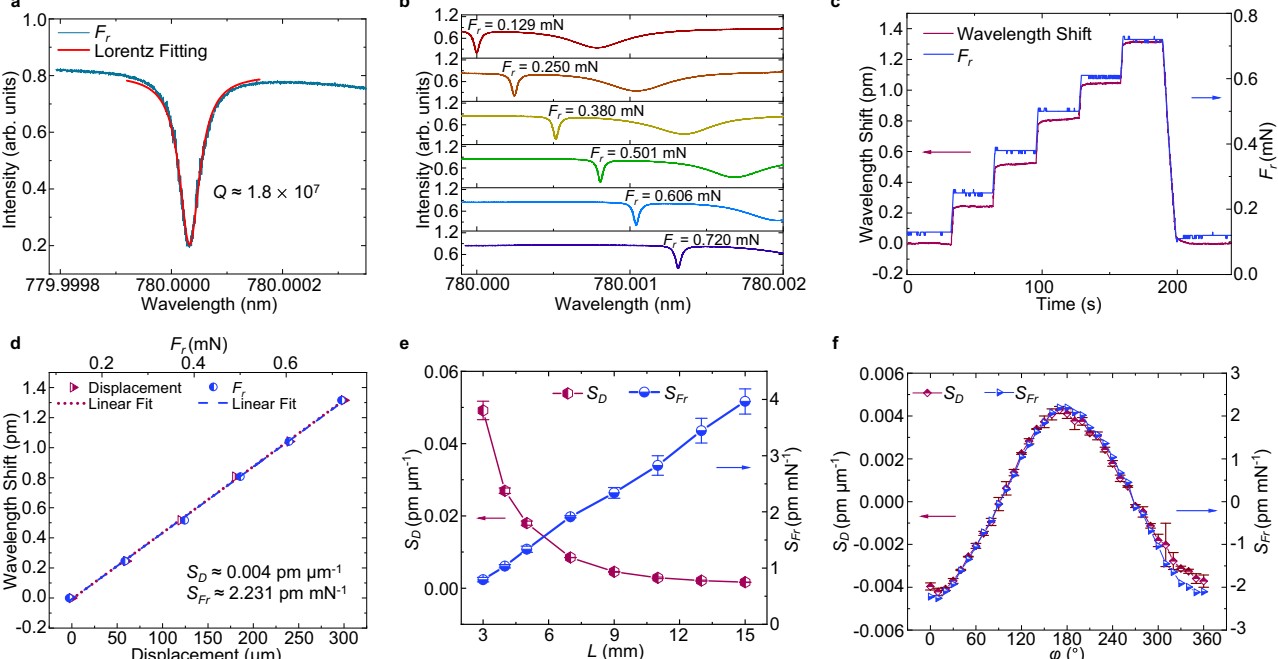

**Fig. 3 | Experiments with the mechano-sensor under an external force $F_r$.** **a** WGM transmission spectra for the mechano-sensor. The quality factor ($Q$ factor) of the WGM is approximately $1.8 \times 10^7$. arb. units, arbitrary units. **b** Spectral evolution of the mechano-sensor as the external force $F_r$ ($\varphi = 180°$) increases from 0.129 mN to 0.720 mN. arb. units, arbitrary units. **c** Responses of the mechano-sensor obtained by tracking the resonance wavelength shift (red curve) under various static forces $F_r$ ($\varphi = 180°$, blue curve). **d** Relations between the resonance

wavelength shift and the displacement/static force ($\varphi = 180°$). **e** Relations between the displacement (red curve) and force (blue curve) sensitivities and the position of the applied force ($L$). **f** Displacement (red curve) and force (blue curve) sensitivities of the mechano-sensor when the external force $F_r$ is applied in different directions ($\varphi$). Data in **e**–**f** are means from three independent experiments; error bars correspond to s.d.

$(\sigma)$ of the resulting spectral variation can be approximated by[41,42]:

$$\sigma \approx \frac{\Delta\lambda}{4.5 \times (SNR)^{0.25}}. \qquad (5)$$

where $\Delta\lambda$ is the FWHM of the resonance dip, which is related to the $Q$ factor. $SNR$ is the signal-to-noise ratio. The WGM shown in Fig. 3a has an $SNR$ of 48.7 dB and a $\sigma$ of 0.0012 pm. The theoretical detection limits of the displacement and force measurements are calculated as $DL_D = 3\sigma/S_D \approx 70$ nm and $DL_{Fr} = 3\sigma/S_{Fr} \approx 0.9$ μN, respectively. The values can be further improved by changing the position of the applied force. In practical applications, an external force usually acts directly on the entire mechano-sensor micro-hair, and the action point is not very significant.

By changing the angle of the external force applied in the $r$-$\varphi$ plane (rotated around the $z$-axis) at 10° intervals (Supplementary Fig. 9, SI), the sensitivity of the mechano-sensor to displacements and static forces applied in different directions ($\varphi$) was determined (maintaining $L = 9$ mm). As shown in Fig. 3f, the displacement and force sensitivities have maximum positive values when $\varphi = 180°$, maximum negative values when $\varphi = 0°$ or 360°, and values of approximately zero when $\varphi = 90°$ and 270°. These experimental results correspond well to the FEM simulation results (Fig. 2f), indicating that the mechano-sensor performance effectively depends on the direction of the applied stimulation. The directionality of the mechano-sensor can be expressed as[43]:

$$K \approx 20 \lg \frac{G'}{G}. \qquad (6)$$

where $G'$ and $G$ are the maximum and minimum force sensitivity values measured in the directionality of the mechano-sensor, respectively.

The measured directional results show that the proposed sensor has a good directionality of 32.31 dB in the range of $\varphi = 0 \sim 360°$ when $L = 9$ mm.

Furthermore, to demonstrate the anti-interference capability of the mechano-sensor, we introduced a temperature variable and immersed the mechano-sensor in sea water during the displacement sensing experiments (Supplementary Fig. 10 and 11, SI). The results show that the mechano-sensor can decouple temperature and displacement sensing and show stable performance in sea water.

## Mechanical perception and analysis in the axial direction of the micro-hair

To analyze the detection performance of the mechano-sensor for the external force $F_z$ applied on the micro-hair, an FEM model was also established, as shown in Fig. 4a. The external force $F_z$ (blue arrow) is applied along the $z$-axis of the mechano-sensor (Fig. 4b and Supplementary Fig.12 in the SI), and the corresponding displacement is set as 2 μm. The stress field distribution in the equatorial cross-section of the mechano-sensor under the external force $F_z$ is shown in Fig. 4c. The radial strain $dR/R$ and the strain-induced effective refractive index change $dn_{eff}/n_{eff}$ in the equatorial cross-section of the microbubble are the averages of their field distribution integrals (Fig. 4d and e) and linearly vary with increasing displacement (Fig. 4f). The radial strain field distribution along the microbubble's equatorial cross-section under an external force $F_z$ is approximately constant, which is different from the gradient distribution under an external force $F_r$. The FEM results indicate that the physical mechanism of the resonance wavelength shift caused by the external force $F_z$ is different from that caused by the external force $F_r$. The external force $F_z$ is transmitted into the microbubble by the micro-hair, which causes the microbubble to expand. As the microbubble expands, the equatorial radius increases, and the strain-induced effective refractive index decreases. The

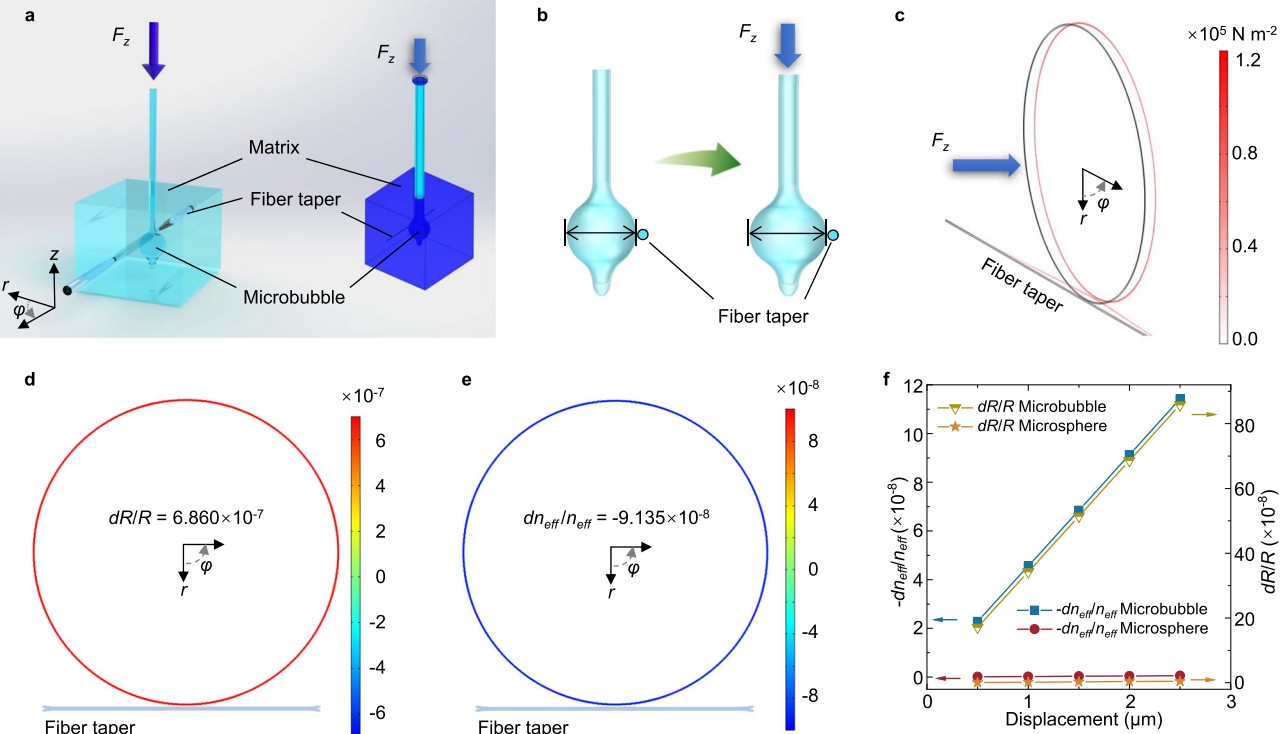

**Fig. 4 | Simulations of the mechano-sensor under an external force $F_z$.**
**a** Schematic illustration of the experimental device and FEM model of the mechano-sensor under an external force $F_z$. **b** Schematic illustration of the deformation of the flexible microbubble in the mechano-sensor under an external force $F_z$. The equatorial diameter of the microbubble increases. **c** Stress field distribution in the $r$-$\varphi$ plane under an external force $F_z$. **d**, **e** Field distributions of the radial strain and strain-induced effective refractive index change along the equatorial cross-section of the microbubble under an external force $F_z$. The $dR/R$ and $dn_{eff}/n_{eff}$ values are the averages of their field distribution integrals. **f** Comparisons of $dR/R$ and $dn_{eff}/n_{eff}$ between the microbubble-based mechano-sensor and the microsphere-based mechano-sensor under different external displacements of $F_z$.

change in the equatorial radius dominates the shift in the resonance wavelength, leading to a redshift in the resonance wavelength.

To verify that the mechano-sensor can detect touch and pressure stimuli, similar to the perception capability of cat whiskers, various static forces are applied along the z-axis to the end of the mechano-sensor micro-hair. Figure 5a shows the spectral evolution of the microbubble WGM resonator. The resonance wavelength redshifts with increasing displacement and external force $F_z$. Since the object applying the external force is driven by a stepper motor with a step value of 0.4 μm, the resonance wavelength shift of the mechano-sensor increases in a stepwise manner during the force application, as shown in Fig. 5b. As shown in Fig. 5c, the displacement and force sensitivities of the mechano-sensor are 1.570 pm μm⁻¹ and 0.986 pm mN⁻¹, respectively. The standard deviation is $\sigma \approx 0.0012$ pm, and the calculated detection limits of the displacement and force measurements are 2.29 nm and 3.65 μN, respectively. Furthermore, a piezo actuator is utilized to apply a cyclic force to the micro-hair along the z-axis to test the durability of the mechano-sensor. By repeatedly applying and releasing a normal force at different frequencies (0.05 Hz, 0.1 Hz, 0.2 Hz, 0.4 Hz, and 0.8 Hz) and displacements (0.054 μm, 0.099 μm, 0.152 μm, 0.249 μm, and 0.316 μm) with various waveforms (Square, Cubic, Dirac, Ramp, and Sine) at the end of the mechano-sensor micro-hair, the durability of the sensor is determined, as shown in Fig. 5d–f. The resonance wavelength shifts of the mechano-sensor caused by the changes in the frequency, amplitude, and waveform of the applied force are determined. These results demonstrate that the responses of the mechano-sensor are quite stable, with no obvious fluctuation with respect to the baseline intensity.

The durability test results of the mechano-sensor for forces with higher frequencies (1 Hz, 2 Hz, 4 Hz, 8 Hz, and 16 Hz) are shown in Supplementary Fig. 13b. To investigate the response of the mechano-sensor, monitoring the intensity change (Δ*I*) in the WGM signal was conducted. As shown in Supplementary Fig. 13d–f, the response and recovery times are just 1.24 ms and 1.26 ms, respectively, surpassing the highest limit of vibration sense for most biological MSHS. Moreover, these results are encouraging compared with the performance of existing mechano-sensors with comparable sizes and effective mechano-sensing ranges but no directionality[44,45].

### Responses to environmental stimuli and obstacle detection

The response of the mechano-sensor to weak signals demonstrates its potential as a device for monitoring weak disturbances in the environment, such as breezes and dripping water. Two independent experiments were conducted to verify the environmental perception capability of the mechano-sensor, and the results are shown in Fig. 6. Compressed air (5 bar) is used to produce a breeze three times towards the mechano-sensor micro-hair along the $\varphi = 180°$ and $0°$ directions, which causes bending and deformation of the mechano-sensor micro-hair and induces an opposite shift in the resonance wavelength, as shown in Fig. 6a, b. Moreover, a damping wavelength shift is observed when a drop of water ($\approx 18$ mg) is dripped on the mechano-sensor micro-hair along the $\varphi = 180°$ direction (Fig. 6c). The water droplet is immediately dispersed after contacting the mechano-sensor micro-hair, inducing vibrations. The wavelength shift is dampened and oscillated with the change in the micro-hair vibrations. The mechano-

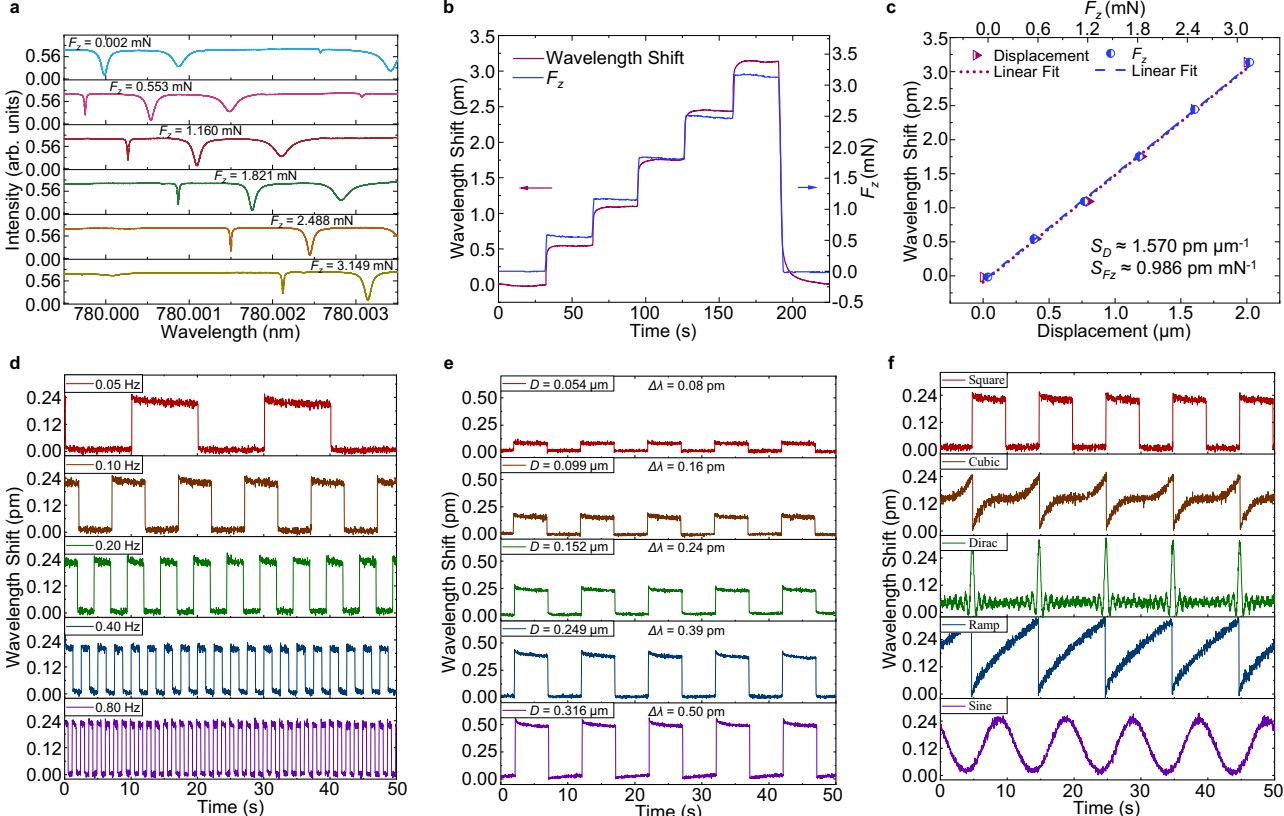

**Fig. 5 | Touch sensing capability of the mechano-sensor. a** Spectral evolution of the mechano-sensor as the external force $F_z$ increases from 0.002 mN to 3.149 mN. arb. units, arbitrary units. **b** Responses of the mechano-sensor obtained by tracking the resonance wavelength shift (red curve) under various static forces $F_z$ (blue curve). **c** Relation between the resonance wavelength shift of the mechano-sensor and the displacement/static force $F_z$ applied along the z-axis. **d** Frequency response of the mechano-sensor. Square signals are applied to the piezo actuator at frequencies of 0.05 Hz, 0.1 Hz, 0.2 Hz, 0.4 Hz, and 0.8 Hz. **e** Displacement response of the mechano-sensor. The 0.1 Hz Square signals are applied to the piezo actuator with displacements of 0.054 μm, 0.099 μm, 0.152 μm, 0.249 μm, and 0.316 μm. **f** Waveform response of the mechano-sensor. The 0.1 Hz signals with Square, Cubic, Dirac, Ramp, and Sine waveforms are applied to the piezo actuator.

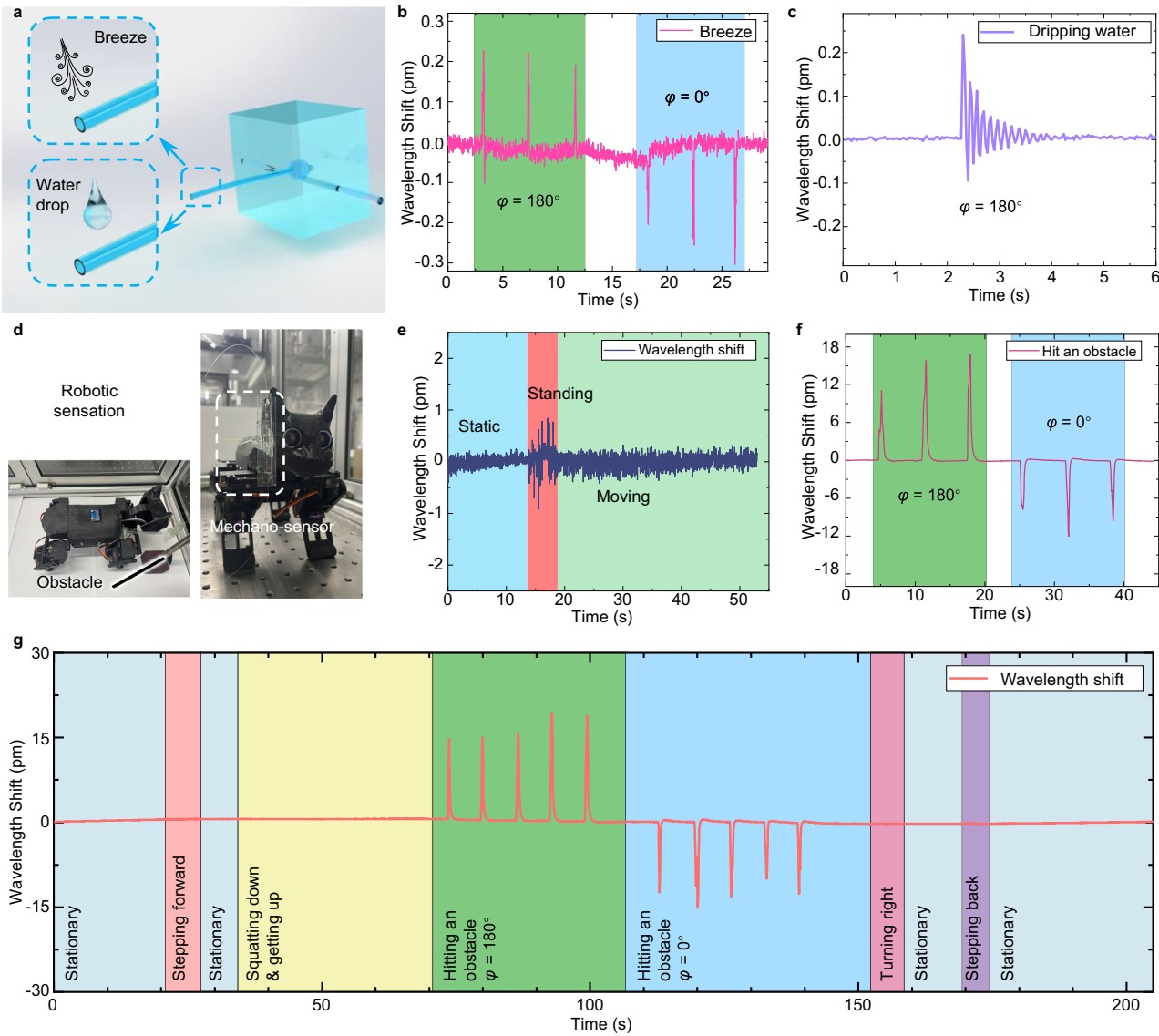

**Fig. 6 | Wavelength responses of the mechano-sensor to weak environmental stimuli and obstacle detection. a** Schematic illustration of breeze blowing and water drop dripping on the mechano-sensor micro-hair in the $\varphi = 180°$ direction. **b** Responses of the mechano-sensor to breezes in the $\varphi = 180°$ and 0° directions. **c** Response of the mechano-sensor as a water droplet is dripped on the micro-hair in the $\varphi = 180°$ direction. **d** Photograph of the quadruped cat robot equipped with the mechano-sensor system. **e** Responses of the mechano-sensor in stationary, standing, and moving states. **f** Responses of the mechano-sensor when hitting an obstacle while advancing ($\varphi = 180°$) and retreating ($\varphi = 0°$). **g** Wavelength shifts of the mechano-sensor caused by various actions, including remaining stationary, stepping forward, squatting down and getting up, hitting an obstacle at $\varphi = 180°$ and 0°, turning right, and stepping back. The background areas in gray, orange, yellow, green, blue, pink, and purple represent the mechano-sensor response when remaining stationary, stepping forward, squatting down and getting up, hitting an obstacle at $\varphi = 180°$, hitting an obstacle at $\varphi = 0°$, turning right, and stepping back, respectively.

sensor provides sensitive responses and recognizable outputs with specific characteristics for different stimuli, demonstrating its potential for perceiving, identifying, and monitoring diverse weak environmental signals.

Sensors with different functions have been widely used in robots for environmental monitoring. To mimic the sensing ability of mammalian whiskers, a mechano-sensor is mounted on a quadruped cat robot, and vibration signals are acquired in real time, as shown in Fig. 6d. A stable signal is obtained when the robot is in a stationary state (Fig. 6e). When the quadruped robot stands up from the stationary state, a small wavelength shift can be observed. As the robot moves, no obvious wavelength shift is observed. These results indicate that the sensor is insensitive to the jitter of the quadruped robot during its movement, and the jitter does not cause obvious interference for obstacle detection. Figure 6f demonstrates that the

mechano-sensor can detect obstacles (optical mounting posts) during the movement of the quadruped robot. The mechano-sensor micro-hair is bent and deformed as the robot contacts the obstacles. When the quadruped robot encounters an obstacle along the $\varphi = 180°$ direction during forward motion, the resonance wavelength is red-shifted. In contrast, when the quadruped robot retreats, the sensor encounters an obstacle along the $\varphi = 0°$ direction, which causes a resonance wavelength blueshift, as shown in Fig. 6f. Finally, the perception capability of the mechano-sensor is demonstrated by applying a series of actions to simulate situations that a cat may encounter in real life, including remaining stationary, stepping forward, squatting down and getting up, hitting an obstacle at $\varphi = 180°$ and 0°, turning right, and stepping back, as shown in Fig. 6g and Supplementary Video 1. The results indicate that the bioinspired all-optical mechano-sensor is a promising device for use in perceptual applications.

## Discussion

Supplementary Table 2 presents different force sensing methods that have been reported in recent years. Compared to most reported electronic whiskers, the innovative photonic whisker is not only easy-to-fabricate and cost-effective, but also achieves similar or even faster response and recovery times, along with lower *DL*.

In summary, for the first time to our knowledge, a proof-of-concept prototype of a bioinspired all-optical multidirectional mechano-sensor with micro-hair mimicking biological MSHS is demonstrated. The minimalist 3D configuration inside a polymer matrix includes a thin-walled glass-microbubble WGM resonator for mechano-opto-transduction, a glass micro-hair serving as the hair probe, and a fiber taper for evanescent coupling. The mechanically flexible and heterogeneously 3D-integrated micro-optics enable the photonic or all-optical mechano-sensor micro-hair, which has a simpler configuration than its analogs (i.e., the biological original references and other artificial electronic hair sensors). The mechano-sensor can detect displacements, forces and vibrations in multiple directions and with various driving frequencies, displacements, and waveforms. In the radial plane of the micro-hair, the mechano-sensor has a good directionality of 32.31 dB, and the maximum displacement and force sensitivities are 0.052 pm µm$^{-1}$ and 3.994 pm mN$^{-1}$, respectively. In the axial direction of the micro-hair, the displacement and force sensitivities are 1.570 pm µm$^{-1}$ and 0.986 pm mN$^{-1}$, respectively. The anti-interference capability of the mechano-sensor, including temperature and sea water interference, is demonstrated. The excellent temperature-displacement decoupling capability of the mechano-sensor demonstrates its potential for use in all-optical multifunctional perception systems with highly integrated structures and functions. Various stimuli, such as aperiodic airflows and water drops, are well detected, and information such as their strength, direction, frequency and characteristic spectral profiles of the peaks and pulses is obtained. The performance of the proposed all-optical mechano-sensor is further demonstrated by integrating it into a cat-like quadruped robot as a real-time mechano-sensory whisker with directionality. The mechano-sensor shows potential for use in state-of-the-art technologies and fields such as innovative artificial perception systems, vibration detection systems, robotics, AR/VR, and the metaverse, especially if advanced 3D top-down micro/nano-fabrication techniques are introduced in future work based on the prototype configuration.

## Methods

### Fabrication of the photonic mechano-sensor

The thin-wall microbubble was fabricated from a fused silica capillary (TSP100200, Zhengzhou INNOSEP Scientific, China) by using a fiber fusion splicer (FSU-975, Ericsson) through the fuse-and-blow method[46], and the details can be found in Supplementary Fig. 14 in the SI. Then, the microbubble was precisely controlled by five-dimensional stages to couple it with a fiber taper and packaged with a matrix of low-refractive-index polymer (MY Polymers Ltd., Israel)[47,48], and the details can be found in Supplementary Fig. 15 in the SI. The packaging method improves the stability of the system, reduces the effect of surrounding perturbations, and mimics the structural and functional characteristics of the MSHS in different organisms in nature. Finally, the bioinspired all-optical mechano-sensor was fabricated and used to detect displacements and forces.

### Experimental setup

A schematic of the experimental setup is shown in Fig. 1e. The light produced by a tunable laser (780 nm, UniQuanta, China) was coupled into the fiber and transmitted to the fiber taper for coupling with the microbubble. The polarization state of the laser light was adjusted with a polarization controller. The WGM signals from the mechano-sensor were monitored by a photodetector (APD430A/M, Thorlabs) and recorded by a data acquisition card (PCIe 6351, National Instruments).

A computer was used to locate the positions (minima) of the spectra after Lorentzian fitting to determine the exact resonance wavelengths of the WGMs. Static forces were applied to the mechano-sensor micro-hair along the *r*-axis and *z*-axis of the mechano-sensor. All sensing experiments were performed in an indoor environment unless otherwise noted (22 °C room temperature and 40% humidity).

### Force measurement

The static force applied to the mechano-sensor micro-hair was given by a commercial force sensor (FA597, FIBOS Measurement Technology, China). The commercial force sensor was mounted on a 3D stepper motor stage (9063-XYZ-PPP-M, Newport) and equipped with a digital force gauge (SBT970, Simbatouch, China). The object applying the external force was driven by the stepper motor, and the wavelength shift and the static force change were recorded by a computer and a digital force gauge in real time, respectively. The displacement value was obtained by the stepper motor. Schematic diagrams of the external forces applied to the mechano-sensor micro-hair along the *r*-axis and *z*-axis of the mechano-sensor are shown in Supplementary Fig. 9 and 12 (SI). A piezo actuator (P-280.20, Physik Instrumente (PI), Germany) was used for the touch sensing and durability tests.

## Data availability

The data generated in this study are provided as a Source Data file. Extra data that support the findings of this study are available from the corresponding authors upon request. Source data are provided in this paper.

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

## Acknowledgements

This work was financially supported by the National Natural Science Foundation of China under grants 62175035 (X.W.), Shanghai Pujiang Program under grants 22PJD008 (Y.-L.S.), Strategic Priority Research Program of the Chinese Academy of Sciences under grants XDA16021200 (Y.-L.S.), Natural Science Foundation of Shanghai under grants 21ZR1407400 (X.W.), and National Natural Science Foundation of China under grants 52205559 (S.-X.W.). The authors thank Dr. Jinliang Hu, Dr. Junhong Guo, Dr. Yi Zhou, Dr. Xing Niu, Mr. Zhiran Liu, and Ms. Man Luo for their help during the research.

## Author contributions

Y.-X.L. and Z.-H.G. conceived and fabricated the sensor, performed the experimental measurements, and analyzed the data, with assistance from X.-Y.Z., S.L., Z.-M.C., and S.-X.W. S.L. provided theoretical support. Y.-X.L. and Z.-H.G. wrote the paper with contributions from all other authors. Y.-L.S. proposed the study with contributions from Z.-H.G., X.W., and W.-F.D. X.W. and Y.-L.S. supervised the project.

## Competing interests

The authors declare no competing interests.
