## [Peer Review File · Nature Communications]

An All-Optical Multidirectional Mechano-Sensor Inspired by Biologically Mechano-Sensitive Hair SensillaREVIEWER COMMENTS

Reviewer #1 (Remarks to the Author):

The manuscript entitled An All-optical Vectorial Mechano-sensor Inspired by Biological Mechano-sensitive Hair Sensilla, by Zhihe Guo and co-workers, presents a very interesting and novel approach to a mechanosensor. The proposed sensor attempts to mimic hair sensilla to test forces and motion (to detect obstacles). The work is interesting and the device presented seems promising for purely optical integration in robotic or virtual reality devices. I think the device is interesting and could potentially be useful to the scientific community. Unfortunately, in its current form, this work does not provide the reader with sufficient information to enable this potential usefulness to be realised. The manuscript, in its present form, leaves too many unanswered questions about the device, its sensing principles, and its performance figures.

The manuscript uses too much space to explain biological sensilla. Although the sensor is bio-inspired in these organs, the sensor is based on simple physical parameters such as changes in the optical path, i.e. refractive index (not thoroughly discussed) and length. The pictures of animals are unnecessarily large and not very informative, and then the sensor is presented in a very small image where it is difficult to understand how it has been constructed. Is the bubble just a glass container filled with air? Is it a liquid with real bubbles in it? What are the dimensions of the glass walls?

In general, the manuscript needs a lot of work and the careful description of the devices and the measurements performed, so it can be useful to the readers and to allow the results to be reproduced elsewhere.

I will try to list most of the questions that I found were not fully or convincingly answered in the manuscript, so the authors have the opportunity to further clarify them.

Is the change in position of the glass vessel the main source of the change in the optical path? How does the refractive index change with applied stress? A detailed explanation and experimental proof of the effect of pressure on the refractive index is lacking. Is it possible to disentangle the bubble position (radius effect) from the refractive index effect?

It is not clear to me how the bubble is clamped. It seems that UV glue is holding the glass bubble in place. Is this UV glue a soft rubbery material or does it provide a hard clamp somewhere in the device?. How do the mechanical properties of this matrix affect the performance of the device, is this reproducible? How does it depend on the curing of the UV glue? In the supplementary information, the authors seem to indicate that the matrix clamps the capillary, leaving the bubble free to move in a leveraged fashion, but I cannot imagine how this can be achieved in the actual device with the information provided.

Line 135 The mechano-sensor can detect touch, pressure, tickle and itch sensations applied. How it can sense tickle and itch, and how it can discriminate between the two, is difficult to understand. I think the biological parallelism may have been carried too far. The measurements only show a capacity to convert applied forces or displacements (actually these two inputs could be considered equivalent) into a shift in the detected wavelength.

Figure 1 e. The authors present a schematic diagram (an actual experimental measurement would be more appealing), but they do not explain why or when a red or blue shift is expected. They do not quantify the expected shifts. In general, the manuscript is difficult to follow as key concepts are not presented in an orderly and clear manner.

Line 172 Stress distributions..... due to external forces have been characterised by FEM.

I would not use the term characterised to describe the FEM simulations.

Line 190 In the actual experiments, the fibre cone can be considered as an object with a higher refractive index than the UV adhesive.

Why did the authors not include this in the FEM simulations? The FEM simulations comparing a microbubble and a solid microsphere do not add much light to the devices or phenomena studied here.

I do not see any simulation results about the change in wavelength, only the stress distribution is shown in Figure 2. This is not a FEM model of the mechanosensor as stated in the figure caption, just a simulation of the effect of forces in a glass vessel. The effect on the refractive index or wavelength shift is not explained or simulated.

Figure 3a shows the typical spectrum of the WGM microbubble resonator,

How typical is that? Can you show some raw data spectra? How does this vary from device to device due to manufacturing differences in the dimensions of the bubble and walls?

The authors state that the Q factor is 1.0×10^7 , but why is this relevant? how does it translate into the performance of the device?

Line 230: "The mechano-sensor cannot recover from a relatively large deformation in a short time due to the overloaded external force or displacement.

I do not understand this sentence. Is the long recovery time a property of the sensor or an artefact of the way the sensor response is measured? Is this overload related to the force application setup or to the sensor under test?

When discussing the force sensitivity of the mechanosensor, the authors do not discuss the precision of the wavelength shift measurement. In fact, no details are given on how they measure the wavelength shift. In Figure 3a I can see the measured intensity in a-u for different wavelengths. How is the wavelength shift measured? Is it just a decrease in light intensity for a given detection wavelength? Are the full spectra recorded?. I would like to see more raw data measurements. The only plot where I can see intensity versus wavelength is Figure 4 a, and then all the plots are relative wavelength shift versus time. I would like to see how the full spectra change during the experiments. The full intensity versus wavelength spectra for the different experimental situations, just before displacement or force is applied, during force application, after force application... Do the spectra stay the same with just a shift in the peaks? Do the peaks broaden? This is not clear from the data presented. Also, why did the authors choose this resonance peak (in red in Fig. 4a) and not the deeper neighbouring one? An explanation of the physics behind the sensor performance and experimental details are missing.

What is the noise in the wavelength shift measurement?

The authors quote a standard deviation from the blank test in line 234, but this should be described more carefully and the sources for this standard deviation in the signal when no force is applied should be explained. The experimental data that led to this number should be shown (either in supplementary or main text).

234 The standard deviation for the blank test was $\sigma = 0.01$ pm. Therefore, the theoretical limits of detection for were calculated as $DL1 = 3\sigma/SD1 = 0.034$ μm and $DL2 = 3\sigma/SFz = 0.03$ (...)

This needs to be explained. How are they obtained? Are these experimental values? If so, why are the limits of detection called theoretical?

Figure 4 j. The applied force is quite noisy, but the wavelength shift does not reflect this. This should be explained.

This manuscript claims a 3D mechanosensor. The authors measure different sensitivities depending on the position of the force applied along the capillary and the directions, but it is not clear how a vectorial mechanosensor is obtained. How does the device discriminate between forces applied at different positions and angles? Several experimental situations may result in the same wavelength shift. If this is not the case, this should be explained and proven with experiments.

Line 353 states that the mechanosensor has the ability to identify/assess the direction of an external displacement/force within a 180° range.

I do not understand how the sensor can judge the direction of the force, a given value of the wavelength shift could correspond to a large force from one direction or a smaller force but from a different direction. How can the sensor tell the difference from just reading the wavelength shift?

From Figure 5, I can see that a red or blue shift is measured depending on the direction of the force, but this is only shown (proven) for two very specific directions (180 degrees and 0 degrees).

A clear explanation is needed, and FEM simulations of this would be more useful than the hard sphere simulations shown in Figure 2e (since no experimental devices composed of hard spheres are shown). I can infer that one angle produces a shorter gap and the other a larger gap and this can cause the red and blue shifts, but I would like to see this explained by the authors, the quantification of the shifts and some modeling, simulations and comparison with measurements carried out. Can the expected shift be predicted for known devices? What happens to angles different to 0 and 180 deg?.

Figure 5 b and Figure 5 f. Why does the sensor respond to the breeze or hitting an obstacle with multiple peaks? Why are there three or five peaks in different experiments?

Line 36, line 43 attitude λ , I do not understand what the authors are referring to here.

Reviewer #2 (Remarks to the Author):

An All-optical Vectorial Mechano-sensor Inspired by Biological Mechano-sensitive Hair Sensilla
Zhihe Guo, Yuxiang Li, Xuyang Zhao, Yi Zhou, Zhenmin Chen, Junhong Guo, Zhiran Liu, Man Luo, Xing Niu, Jinliang Hu, Wen-Fei Dong, Shixiang Wang, Yun-Lu Sun, and Xiang Wu

The authors report an all-optical force sensor inspired by mechano-sensitive biological sensilla, like mammalian whiskers. The principle of operation is an optomechanical coupling in between a microbubble and a tapered optical fiber. The microbubble is attached to a hollow glass capillary, which is used as the transducer of the sensor device.

I think that the work is very interesting and innovative. I especially appreciate the integration of the device, the design is elegant and makes sense, plus the chosen application is very relevant. I really think that the presented work could be of interest for the community. However, I cannot support the publication of the work at this time because I have identified some deficiencies in the manuscript related to both the clarity of presentation of the results and their interpretation. These are my comments that should be answered by the authors before my final recommendation.

1. The figures are not too clear. They have too many sub-panels that make it hard to follow them in the main text. I strongly suggest that the authors consider reducing the size of the figures or separating them into several
2. The values of sensitivity given by the authors are, somehow, meaningless. As a general consideration, when you are designing a force sensor, the sensitivity depends on the actual dimensions and material of the transducer, in this case the glass capillary. What is the spring constant of the system? I am pretty sure that the value obtained in this work can easily be exceeded by simply making the capillary longer, or thinner. The authors have to include a parametric sweep at the beginning of the study to justify the chosen geometry
3. The authors claim "higher sensitivity" by using this all-optical system when compared with the typical electrical configuration. What is the sensitivity of the electrical sensor? How many times are the all-optical one better? I feel that the authors did not do a good job of reviewing the existing literature.
4. General note, please, revise the English, some parts are very difficult to understand
5. What do authors mean when they state "The photonic mechano-sensor is of the displacement and force sensitivities of $0.87 \text{ pm} \cdot \mu\text{m}^{-1}$ and $1.07 \text{ pm} \cdot \text{mN}^{-1}$ in the axial direction of the "hair", respectively"? How are you defining sensitivity? Please include a paragraph defining sensitivity, because I think the authors are using responsivity instead of sensitivity.
6. The dimensions of the probe are $125 \mu\text{m}$ outer diameter, $100 \mu\text{m}$ inner diameter and 36 mm length, why it a hollow glass capillary? Can it be solid? Again, a study of the geometrical parameters needed. I also wonder if the capillary has any kind of polymer coating, because it seems from figure S1 that it is coated by polyimide, which greatly affects the mechanical properties of the device.
7. I guess the neff will be affected mainly by the deformation of the gap in between the microbubble and the fiber, therefore, the mechanical parameters (Young's modulus and Poisson ratio) of the polymer are just as important as the optical parameters (n and k). Could you please provide those numbers? I also think that a parametric sweep of the material properties would also be very useful. What is the ideal material?
8. I wonder why all the curves in Fig. 2 are linear, even when the authors state "Once an external force F_z was applied on the end of the mechano-sensor hair with a displacement of $20 \mu\text{m}$, the microbubble began to deform, and the major stress was located at the thin wall of the microbubble (Figure 2d)." Thus, it seems that there is a not shown threshold in the system.

9. Since a critical parameter is the evanescent light coupling, I am not sure that is justified to remove the fiber from the simulations, especially when the displacements are on the order of nanometers. I am pretty sure that the relative change in gap is modified if you include the fiber in FEM simulations.

10. The comparison in between the microbubble and the microsphere is not necessary in the main text, it can be summarized as the microsphere is stiffer. Again, it is important to make the calculation of the spring constant of the whole system

11. How do you calculate the force in Fig. 3? Are you using the Hooke's law? Explain the transformation of displacement into force, the calculation and/or the performed simulations, as well as the approximations that have been made.

12. I wonder if the authors have considered the frequency domain analysis of the sensor device. I am pretty sure that the study of the resonance of the "hair" can give more information, e.g., about vector force fields. In that sense I recommend to take a look to the existing literature of the field

13. Why do you have a red shifting at the beginning of the quadruped experiment? It seems to be related with temperature because it reaches a stationary value.

14. Since it seems that there is a transient state due to the temperature, I wonder what is the power of the laser that has been used in the experiments? Have the authors tried using different laser powers?

Reviewer #3 (Remarks to the Author):

This manuscript demonstrates an all-optical mechano-sensor for the detection of forces with different direction information. This work is meaningful and the mentioned device performs relatively good performances, however, it does not deserve a recommendation to be published in Nature Communications.

The footprint of the proposed device is very big, including laser, fiber, sensor, etc., which is not so-called minimalist and highly impedes its scale-up.

The signal decoupling of the simultaneously applied mechanical stimuli is seemingly impossible or difficult.

What is the Young's modulus of the UV matrix? Is it soft or hard? If it is hard, the deformation of the glass microbubble is not easy. If it is soft, the position of fiber taper will change during the force application process. The related optimization and discussion are suggested.

What is the response behavior of the device at high frequencies more than 1 Hz? The device shows a response time of 20 ms, which means it can respond at a relatively high frequency around 50 Hz. The related data is suggested. And a table of comprehensive comparison between this proposed device and others (reported or commercial ones, if possible) is suggested, in terms of key sensing index, configuration, energy consumption, etc.

For the durability test, the voltage amplitudes are suggested to be replaced with the practical displacements.

How about the anti-interference capability of the device to surrounding perturbations, in consideration of its high sensitivity to tiny mechanical stimuli?

MANUSCRIPT REVISIONS DETAILS

Ms. No.: NCOMMS-23-08485

Ms. Title: An All-Optical Multidirectional Mechano-sensor Inspired by Biologically Mechano-sensitive Hair Sensilla

Authors: Yuxiang Li, Zhihe Guo, Xuyang Zhao, Sheng Liu, Zhenmin Chen, Wen-Fei Dong, Shixiang Wang, Yun-Lu Sun, and Xiang Wu

We appreciate reviews by three reviewers. We have revised our manuscript and Supplementary Information according to the comments from reviewers. All changes made to the text are using the "Highlight" function with **blue color** in the revised manuscript and Supplementary Information.

Here, we present a point-by-point reply (in **blue**) to the comments (in **black**) from reviewers, as well as the new/rephrased sentences in the main manuscript and Supplementary Information (in **green**).

Reply to Reviewer #1

The manuscript entitled An All-optical Vectorial Mechano-sensor Inspired by Biological Mechano-sensitive Hair Sensilla, by Zhihe Guo and co-workers, presents a very interesting and novel approach to a mechanosensor. The proposed sensor attempts to mimic hair sensilla to test forces and motion (to detect obstacles). The work is interesting and the device presented seems promising for purely optical integration in robotic or virtual reality devices. I think the device is interesting and could potentially be useful to the scientific community. Unfortunately, in its current form, this work does not provide the reader with sufficient information to enable this potential usefulness to be realised. The manuscript, in its present form, leaves too many unanswered questions about the device, its sensing principles, and its performance figures.

We would like to thank the reviewer for the constructive comments and are delighted that the reviewer is interested in our results. We are happy to address all the comments below. We thank the reviewer for motivating those improvements.

Comment 1: *The manuscript uses too much space to explain biological sensilla. Although the sensor is bio-inspired in these organs, the sensor is based on simple physical parameters such as changes in the optical path, i.e. refractive index (not thoroughly discussed) and length. The pictures of animals are unnecessarily large and not very informative, and then the sensor is presented in a very small image where it is difficult to understand how it has been constructed. Is the bubble just a glass container filled with air? Is it a liquid with real bubbles in it? What are the dimensions of the glass walls?*

Answer 1:

1.1 For the graph modification:

Answer 1.1: As per the reviewer's suggestion, Figure 1 have been revised to make it clearer and more legible. We have reduced the size of the animal pictures and enlarged the image of the mechano-sensor in Figure 1. Additionally, we have simplified the introduction of biological sensilla to highlight the mechano-sensor.

We have revised Figure 1 in the revised manuscript as:

Fig. 1. Schematic illustration of the bioinspired mechano-sensor. **(a)** Schematic illustration of the cat whisker sensory system and whisker mechanoreceptor anatomy. **(b)** Schematic diagram of a fish lateral line system and a neuromast cross-section. **(c)** Schematic diagram of a spider and the tactile hair protruding from its exoskeleton's surface. **(d)** Schematic diagram of the mechano-sensor. A microbubble with a micro-hair, similar to the whisker of a cat, is used to detect tactile stimuli, analogous to a sensory receptor. A fiber taper is used to transmit signals, which is similar to signal transmission by nerve fibers. The polymer matrix can fix the microbubble and fiber taper and protect them from environmental damage, which is similar to the function of the skin. The inset shows an image of a microbubble coupled with a fiber taper under a microscope. The scale bar represents 200 μm . **(e)** Experimental setup of the all-optical mechano-sensor system. The blue arrow indicates the direction of the external force. **(f)** Responses of the bioinspired

mechano-sensor in different deformation states (i.e., $F_r = 0$ mN, 0.137 mN ($\varphi = 180^\circ$), and 0.128 mN ($\varphi = 0^\circ$)). on Page 28.

We have deleted the discussion of biological sensilla in the revised manuscript as:

"Mammals, including humans, rely on cochlear hair cells with stereocilia and kinocilia for the transduction of mechano/acoustic stimuli into membrane potential changes in receptor neurons^{1,2}, and some mammals (felid, murine, etc.) also use whiskers to provide nonvisual but acute sensory information, such as vibrations, under poor visibility conditions³. Moreover, the cilia of crickets detect mechanical signals in the surrounding environment and courtship and prey signals^{4,5}; fishes use their lateral line system to detect water motions and pressures for predator and prey detection, object avoidance and social behaviors⁶⁻⁸; and spiders use the deflections of the tactile hair protruding from their exoskeleton's surface to detect forces and prey or predator signals propagated through airflow or wet areas^{2,9}." on Page 3.

1.2 For the mechano-sensor configuration:

Answer 1.2: As schematically illustrated in Figure R1, to construct the bioinspired photonic mechano-sensor, a thin-walled glass microbubble with a glass micro-hair placed on its top center was coupled with a fiber taper at the equator of the microbubble. The 3D-integrated structure was embedded within a matrix of ultraviolet (UV)-crosslinked low-refractive-index polymer.

To obtain a mechano-sensor with high displacement sensitivity, the geometric parameters of the micro-hair and microbubble and the mechanical properties of the polymer matrix were optimized through the Finite Element Method (FEM) (see more details in the response to Question 2/Reviewer #2 and Question 5/Reviewer #1). The optimized radius $R_{capillary}$ and optimized wall thickness $T_{capillary}$ of the capillary were 84 μm and 8 μm , respectively. The optimized radius R_{bubble} and optimized wall thickness T_{bubble} of the microbubble were 185 μm and 1.5 μm , respectively. The MY-133-V2000 polymer with an elastic modulus of 5.2 MPa and a Poisson's ratio of 0.41 was selected as the optimized matrix.

Figure R1. Schematic diagram of the mechano-sensor. The inset shows an image of an optimized microbubble coupled with a fiber taper under a microscope. The scale bar represents 200 μm .

We have added the relevant discussion in the revised Supplementary Information as:

"The fabrication of the microbubble is shown in Supplementary Fig. 14. Initially, to prepare a thin-walled hollow microbubble, a fused silica capillary (Zhengzhou INNOSEP Scientific, China) needed to be corroded by hydrofluoric acid until the corroded wall thickness was 4~10 μm . Subsequently, a section of a silica capillary was placed at the electrodes of an optical fiber fusion splicer. When the electrodes were discharged, one end of the silica capillary was sealed. Then, the outer coating layer of the silica capillary was burnt with an alcohol burner and wiped with microscope lens paper. The silica capillary was placed at the center of the electrodes. The other end of the silica capillary was connected to a Teflon tube and a syringe. While the syringe was being pushed and the electrodes were discharging, the silica capillary was heated, melted and expanded, and finally, a hollow core microbubble structure with a long mechano-sensor micro-hair was fabricated." on Page S23 (Part 9 in SI).

Supplementary Fig. 14. Fabrication procedures for the microbubble resonator. (a) A silica capillary was used, and (b) the end of the capillary was sealed by electrode discharge. (c) The coating was burned, and the capillary was cleaned with microscope lens paper. (d) The capillary was connected to a syringe and placed under the electrodes. (e) The electrodes were discharged, and air was blown into the capillary, causing it to swell. (f) Finally, the prepared microbubble resonator with a long mechano-sensor micro-hair was obtained.

"The final version of the bioinspired optical mechano-sensor is shown in Supplementary Fig. 6, which was composed of a thin-walled glass microbubble integrated with a glass micro-hair that was optically coupled with a fiber taper at the equator of the microbubble resonator. To fit the outer contour curve of the microbubble with a Gaussian line shape and quantitatively control the geometric parameters of the microbubble, the prepared microbubble is approximately 200 μm away from the melting node where the glass capillary is sealed. Notably, such a short distance of 200 μm does not affect the displacement sensitivity of the mechano-sensor. The radius ($R_{\text{capillary}}$) and wall thickness ($T_{\text{capillary}}$) of the glass capillary are 84 μm and 8 μm , respectively. The radius (R_{bubble}) and wall thickness (T_{bubble}) of the glass-based hollow microbubble are 185 μm and 1.5 μm , respectively. The 50 mm long micro-hair collects information about the mechanical stimuli in the environment, such as forces and vibrations, and propagates the information to the microbubble. The thin-walled microbubble configuration is mechanically flexible, allowing the

device to transform these environmental mechanical stimuli into shifts in the WGM resonance wavelength and the spectral light signals (i.e., shifts in the dips)." on Page S11 (Part 4 in SI).

Supplementary Fig. 6. (a) Image of the mechano-sensor. The scale bar represents 10 mm. (b) Optical microscope image of a microbubble integrated with a glass micro-hair standing on its top center. The scale bar represents 200 μm .

Comment 2: *In general, the manuscript needs a lot of work and the careful description of the devices and the measurements performed, so it can be useful to the readers and to allow the results to be reproduced elsewhere. I will try to list most of the questions that I found were not fully or convincingly answered in the manuscript, so the authors have the opportunity to further clarify them.*

Answer 2: In response to the reviewer's feedback, we have comprehensively revised the manuscript, specifically enhancing the detailed descriptions of the devices and measurements. We appreciate the constructive feedback from the reviewer, and believe that the revised manuscript effectively addresses the raised concerns.

Comment 3: *Is the change in position of the glass vessel the main source of the change in the optical path?*

Answer 3: We thank the reviewer for the constructive suggestions. As pointed out by the reviewer, we have provided a detailed explanation of the resonance wavelength shift.

First, we prepared a solid microsphere-based mechano-sensor and a hollow microbubble-based mechano-sensor. Both mechano-sensors were encapsulated in the same polymer matrix (MY-133-V2000), and an external force (F_r , $\varphi = 180^\circ$) was applied at the same axial position.

The measured force sensitivity of the solid microsphere-based mechano-sensor was very low (Figure R2). In addition, we prepared four microbubble-based mechano-sensors encapsulated in four different commercial polymer matrices (Table R1). To study the effect of the polymer matrix on the force sensitivity of the mechano-sensor, the geometric parameters of the microbubbles and the capillaries were the same for the four mechano-sensors, and the external forces (F_r , $\varphi = 180^\circ$) were applied at the same axial position. As shown in Figure R2, the mechano-sensor encapsulated in MY-133-V2000 had the highest sensitivity. The mechano-sensors encapsulated in polymer matrices with smaller or larger elastic moduli had lower force sensitivities. Because the microsphere-based mechano-sensor and the microbubble-based mechano-sensor encapsulated in the polymer matrix with the smallest elastic modulus were not sensitive to external force, the change in the position of the glass vessel is not the main source of the change in the optical path.

The above experimental results indicated that the strain effect on the microbubble was the main source of the change in the optical path, that is, the shift in the resonance wavelength. Under the same external force, the strain effect on the solid microsphere was much weaker than that on the hollow microbubble. For the four microbubble-based mechano-sensors, the strain effect on the microbubble was not significant if the elastic modulus of the polymer matrix encapsulating the mechano-sensor was small, and the microbubble cannot deform if the elastic modulus of the polymer matrix encapsulating the mechano-sensor was too large. Therefore, the strain effect on the microbubble was strongest at an intermediate value of the elastic modulus.

Moreover, the FEM simulation results, derived from the theory of the strain effect on the microbubble, aligned well with the experimental findings. This serves as the additional confirmation of the theory's validity (see more details in the response to Question 5/Reviewer #1).

Table R1. Comparison of the properties of commercial UV-crosslinked low-refractive-index polymer.

Product name	RI at 950 nm	Elastic Modulus (MPa)	Poisson's Ratio	Hardness Shore
MY-132-A	1.322	0.4	0.495	30 A
MY-132-V15K	1.322	Very low	-	7
MY-133-V2000	1.329	5.2	0.41	70 A
NOA1348	1.348	158	-	30 D

Figure R2. Experimental results of the force sensitivity comparison among microsphere-based mechano-sensor encapsulated with MY-133-V2000 and microbubble-based mechano-sensors encapsulated with four different polymer matrices (MY-132-V15K, MY-132-A, NOA1348, and MY-133-V2000).

We discuss this issue in the revised Supplementary Information as:

"To analyze the main reason for the shift in the resonance wavelength, a solid microsphere-based mechano-sensor and a hollow microbubble-based mechano-sensor were prepared. Both mechano-sensors were packaged in the same polymer matrix (MY-133-V2000), and external forces (F_r , $\varphi = 180^\circ$) were applied at the same axial position. As shown in Supplementary Fig. 1, the solid microsphere-based mechano-sensor had very little force sensitivity. In addition, microbubble-based mechano-sensors encapsulated with four different commercial polymer matrices (**Supplementary Table 1**) were prepared. To study the effect of the polymer matrix on the force sensitivity of the mechano-sensor, the geometric parameters of the microbubbles and the micro-hairs were the same in all four mechano-sensors, and the external forces (F_r , $\varphi = 180^\circ$) were applied at the same axial position. As shown in Supplementary Fig. 1, the mechano-sensor encapsulated with MY-133-V2000 had the highest sensitivity. The mechano-sensors encapsulated with polymer matrices with smaller or larger elastic moduli had lower force sensitivity.

The above experimental results indicate that the strain effect on the microbubble is the main reason for the shift in the resonance wavelength. Under the same external force, the strain effect

on the solid microsphere is much weaker than that on the hollow microbubble. For the four microbubble-based mechano-sensors, the strain effect on the microbubble is not significant if the elastic modulus of the polymer matrix encapsulating the mechano-sensor is small, and the microbubble cannot deform if the elastic modulus of the polymer matrix encapsulating the mechano-sensor is too large. Therefore, the strain effect on the microbubble is strongest at an intermediate value of the elastic modulus." on Page S2 (Part 2 in SI).

Comment 4: *How does the refractive index change with applied stress? A detailed explanation and experimental proof of the effect of pressure on the refractive index is lacking. Is it possible to disentangle the bubble position (radius effect) from the refractive index effect?*

Answer 4:

4.1 For the refractive index change with applied force:

Answer 4.1: The working principle of the mechano-sensor was based on the condition of optical resonance, which can be expressed as follows:

(R3)

where R and n_{eff} represent the equatorial radius and effective refractive index of the microbubble resonator, respectively, and m ($\gg 1$) is an integer representing the azimuth quantum number. The force sensing mechanism was based on the strain effect of the glass microbubble, which was demonstrated in the experiments (see more details in Figure R2). Therefore, the application of a force through the micro-hair changes the R and n_{eff} values of the microbubble, causing the corresponding shift in the resonance wavelength:

(R4)

Here, dR/R represents the radial strain of the microbubble's equatorial cross-section, and dn_{eff}/n_{eff} is the change in the effective refractive index due to the mechanical stress or strain, which is given by [Ref: *Appl. Opt.* **32**, 3601-3609 (1993)]:

(R5)

where ε_x , ε_y and ε_z are the strains along the three coordinate axes for each point in space within

the microbubble's equatorial cross-section, as indicated in Figure R3, and p_{ij} are Pockel's elasto-optic coefficients of silica.

Figure R3. Schematic illustration of the strain components (ε_x , ε_y and ε_z) along the three coordinate axes in Cartesian coordinates.

We discuss this issue in the revised manuscript as:

"The force sensing mechanism is based on the strain effect of the glass microbubble, which has been experimentally demonstrated (see more details in Supplementary Fig. 1 and Table 1 in the SI). Therefore, a force applied to the micro-hair changes the R and n_{eff} values of the microbubble, causing the following shift in the resonance wavelength^{37,38}:

(2)

Here, dR/R represents the radial strain of the microbubble's equatorial cross-section, and dn_{eff}/n_{eff} is the change in the effective refractive index due to the mechanical stress or strain, which is given by³⁹:

(3)

where ε_x , ε_y and ε_z are the strains along the three coordinate axes for each point in space within the microbubble's equatorial cross-section, as indicated in Supplementary Fig. 2a (SI), and p_{ij} are Pockel's elasto-optic coefficients." on Page 8.

"The strain (ε_x , ε_y and ε_z) components along three coordinate axes in Cartesian coordinates are shown in Supplementary Fig. 2a. To simplify the mechano-sensor, the lever model is introduced (Supplementary Fig. 2b). The mechano-sensitive micro-hair served as a lever in the system. The UV glue (Norland, NOA68) with an elastic modulus of 138 MPa and a hardness shore of 60 D

was used for rigid fixation and served as the fulcrum in the lever system. The low-refractive-index polymer matrix (MY-133-V2000) with an elastic modulus of 5.2 MPa and a hardness shore of 70 A was used to elastically fix and protect the microbubble and the fiber taper." on Page S4 (Part 3 in SI).

Supplementary Fig. 2. (a) Schematic illustration of the strain components (ϵ_x , ϵ_y and ϵ_z) along the three coordinate axes in Cartesian coordinates. (b) Schematic illustration of the mechano-sensor under an external force F_r , showing the design of the lever model.

4.2 For a simulation to distinguish between dR/R and dn_{eff}/n_{eff} :

Answer 4.2: To analyze the impact of external forces on dR/R and dn_{eff}/n_{eff} , we established a finite element model.

Our mechano-sensor can detect the multidirectional displacement and force in both the radial plane and axial direction of the micro-hair. Consider the scenario of applying an external force F_z to the end of the micro-hair along the axial direction (Figure R4a-b). In the simulation, the displacement caused by the external force F_z was 2 μm . The stress field distribution in the equatorial cross-section of the mechano-sensor under the external force F_z was shown in Figure R4c. The stress on the microbubble wall was over 2-3 orders of magnitude higher than that on the polymer matrix, and therefore, the stress effect of the polymer matrix was neglected. The radial strain field distribution (Figure R4d) and the strain components (i.e., ϵ_x , ϵ_y and ϵ_z) along the microbubble's equatorial cross-section can be directly obtained through FEM simulations. According to Equation (R5), by using the reported values of $p_{11} = 0.131$ and $p_{12} = 0.26$ for fused silica, the field distribution of the strain-induced effective refractive index change was derived (Figure R4e). The radial strain dR/R and the effective refractive index change dn_{eff}/n_{eff} in the microbubble's entire equatorial cross-section were the averages of their field distribution integrals, which were indicated in Figure R4d and R4e, respectively. The micro-hair introduced an external force F_z into the microbubble, which led to an expansion of the microbubble. As the microbubble expanded, the equatorial radius increased, and the strain-induced effective

refractive index decreased. The change in the equatorial radius dominated the shift in the resonance wavelength, leading to a redshift in the resonance wavelength, which was consistent with the experimental results (see more details in Figure 5 in the revised manuscript and Figure R7).

As a comparison, the hollow microbubble was replaced by a solid microsphere of the same size, and the dR/R and dn_{eff}/n_{eff} values of the microsphere-based mechano-sensor were much smaller than those of the microbubble-based mechano-sensor, as shown in Figure R4f. The results in Figure R4f also indicated that for the displacement range considered, dR/R was larger than the dn_{eff}/n_{eff} , and the dependence of $d\lambda/\lambda$ on the displacement (that is, external force F_z) was essentially linear.

Figure R4. Simulations of the mechano-sensor under an external force F_z . **(a)** Schematic illustration of the experimental device (left) and FEM model of the mechano-sensor (right) under the external force F_z . **(b)** Schematic illustration of the deformation of the flexible microbubble in the mechano-sensor under an external force F_z . The equatorial diameter of the microbubble increased. **(c)** The stress field distribution in the r - ϕ plane under an external force F_z . **(d)** and **(e)** The field distributions of the radial strain and strain-induced effective refractive index change along the equatorial cross-section of the microbubble under an external force F_z . The dR/R and dn_{eff}/n_{eff} values were the averages of their field distribution integrals. **(f)** Comparisons of dR/R

and dn_{eff}/n_{eff} between the microbubble-based mechano-sensor and the microsphere-based mechano-sensor under different external displacements of F_z .

We discuss this issue in the revised manuscript as:

"2.3 Mechanical Perception and Analysis in the Axial Direction of the Micro-Hair

To analyze the detection performance of the mechano-sensor for the external force F_z applied on the micro-hair, an FEM model was also established, as shown in Fig. 4a. The external force F_z (blue arrow) was applied along the z -axis of the mechano-sensor (Fig. 4b and Supplementary Fig.12 in the SI), and the corresponding displacement was set as 2 μm . The stress field distribution in the equatorial cross-section of the mechano-sensor under the external force F_z is shown in Fig. 4c. The radial strain dR/R and the strain-induced effective refractive index change dn_{eff}/n_{eff} in the equatorial cross-section of the microbubble are the averages of their field distribution integrals (Fig. 4d and e) and linearly varied with increasing displacement (Fig. 4f). The radial strain field distribution along the microbubble's equatorial cross-section under an external force F_z was approximately constant, which was different from the gradient distribution under an external force F_r . The FEM results indicated that the physical mechanism of the resonance wavelength shift caused by the external force F_z was different from that caused by the external force F_r . The external force F_z was transmitted into the microbubble by the micro-hair, which caused the microbubble to expand. As the microbubble expanded, the equatorial radius increased, and the strain-induced effective refractive index decreased. The change in the equatorial radius dominated the shift in the resonance wavelength, leading to a redshift in the resonance wavelength." on Page 13.

Comment 5: *It is not clear to me how the bubble is clamped. It seems that UV glue is holding the glass bubble in place. Is this UV glue a soft rubbery material or does it provide a hard clamp somewhere in the device? How do the mechanical properties of this matrix affect the performance of the device, is this reproducible? How does it depend on the curing of the UV glue? In the supplementary information, the authors seem to indicate that the matrix clamps the capillary, leaving the bubble free to move in a leveraged fashion, but I cannot imagine how this can be achieved in the actual device with the information provided.*

Answer 5: We thank the reviewer for the constructive suggestions.

5.1 For clamping mechanism:

Answer 5.1: The microbubble can be clamped by the following fabrication procedures. The details of the fabrication procedure of the sensors have been described in Ref. [*Appl. Opt.* **55**, 395, (2016).] and [*Micromachines*, **13**, 592 (2022).]. The UV glue (Norland, NOA68) with an elastic modulus of 138 MPa and a hardness shore of 60 D was used for rigid fixation and a hard clamp. A UV-crosslinked low-refractive-index polymer matrix (MY-133-V2000) with an elastic modulus of 5.2 MPa and a hardness shore of 70 A was used to elastically fix and protect the microbubble and the fiber taper.

A schematic of the packaging process was shown in Figure R5.

First, the fiber taper was fixed on a glass scaffold with NOA68 glue (Figure R5a – R5c). Second, the gap between the fiber taper and glass scaffold was filled with the polymer matrix (Figure R5d). Next, the microbubble was accurately controlled by five-dimensional optic alignment stages to realize the best coupling condition with the fiber taper (Figure R5e), and the micro-hair was glued on the glass scaffold by NOA68 glue. Afterward, the optical coupling region between the fiber taper and the microbubble was completely wrapped with the polymer matrix to improve the stability of the sensor (Figure R5f). When the micro-hair was swung or bent under an external force, the NOA68 glue served as a rigid fixation and can be considered as the fulcrum in the lever system, and the microbubble wrapped in the polymer matrix underwent the elastic deformation.

Figure R5. Fabrication procedures for the bioinspired mechano-sensor.

We discuss this issue in the revised Supplementary Information as:

"The fabrication procedures for the bioinspired mechano-sensor are shown in Supplementary Fig. 15. The UV glue (Norland, NOA68) with an elastic modulus of 138 MPa and a hardness shore of 60 D was used for rigid fixation and a hard clamp. The polymer matrix (MY-133-V2000) with an elastic modulus of 5.2 MPa and a hardness shore of 70 A was used to elastically fix and protect the microbubble and the fiber taper. First, the fiber taper was fixed on a glass scaffold with NOA68 glue (Supplementary Fig. 15a–c). Second, the gap between the fiber taper and glass scaffold was filled with the polymer matrix (Supplementary Fig. 15d). Next, the microbubble was accurately controlled by five-dimensional optic alignment stages to realize the best coupling condition with the fiber taper (Supplementary Fig. 15e), and the micro-hair was glued on the glass scaffold by NOA68 glue. Afterward, the optical coupling region between the fiber taper and the microbubble was completely wrapped with the polymer matrix to improve the stability of the sensor (Supplementary Fig. 15f)." on Page S24 (Part 9 in SI).

5.2 For mechanical properties of polymer matrix:

Answer 5.2: The experimental results, indicating the influence of the polymer matrix's mechanical properties on device performance, were detailed in the response to Question 3/Reviewer #1. Additionally, the relevant FEM simulation model was established.

To obtain a mechano-sensor with higher displacement/force sensitivity, we systematically optimized the geometric parameters of the capillary and microbubble, along with the mechanical properties of the polymer matrix. We will not describe the optimization of the geometric parameters too much here; please see the response to Question 2/Reviewer #2 for more details.

The mechanical properties of the polymer matrix, including the elastic modulus E and Poisson's ratio ν , were optimized with fixed geometric parameters of the capillary and the microbubble. The external force F_r ($\varphi = 180^\circ$) was applied along the r -axis at the same axial position on the micro-hair. As shown in **Figure R6**, when the elastic modulus of the polymer matrix was 5 MPa, the strain effect was the strongest, and the displacement sensitivity of the mechano-sensor was the highest. An increase or decrease in the magnitude of the elastic modulus would reduce the displacement sensitivity of the mechano-sensor, which corresponded to the experimental results in Figure R2. Poisson's ratio had relatively little influence on the displacement sensitivity. Thus, considering the common commercial UV-crosslinked low-refractive-index polymers data (Table R1), the MY-133-V2000 polymer with an elastic modulus E of 5.2 MPa and Poisson's ratio ν of 0.41 was selected as the polymer matrix.

Figure R6. Relations between the displacement sensitivity (S_D) and the mechanical properties of the polymer matrix (i.e., Poisson's ratio (ν) and elastic modulus (E)) under an external force F_r ($\varphi = 180^\circ$).

We discuss this issue in the revised Supplementary Information as:

"Finally, the mechanical properties of the polymer matrix, including the elastic modulus E and Poisson's ratio ν , were optimized with fixed geometric parameters of micro-hair and microbubble. As shown in Fig. 2i, when the elastic modulus of the polymer matrix was 5 MPa, the strain effect was the strongest, and the displacement sensitivity of the mechano-sensor was the highest. An increase or decrease in the magnitude of the elastic modulus would reduce the displacement sensitivity of the mechano-sensor, which corresponded to the experimental results in Supplementary Fig. 1. The reason may be that the strain effect on the microbubble was not significant if the elastic modulus of the polymer matrix was too small, and the microbubble could not be deformed if the elastic modulus of the polymer matrix was too large. Poisson's ratio had relatively little influence on the displacement sensitivity. Thus, considering the commercial UV-crosslinked low-refractive-index polymers data (Supplementary Table 1), the MY-133-V2000 polymer with an elastic modulus E of 5.2 MPa and Poisson's ratio ν of 0.41 was selected as the polymer matrix." on Page S13 (Part 4 in SI).

5.3 For reproducibility:

Answer 5.3: By carefully controlling the experiment details, we ensured the device's reproducible performance. The laboratory environment was a ten-thousand grade clean room with constant temperature and humidity of $22 \pm 2^\circ\text{C}$ and 15.5%, respectively. The volume and

shape of both the NOA68 glue and polymer matrix (MY-133-V2000) were quantitatively controlled. The polymer matrix was fully cured with 350 nm UV light with a dose of 2000 mJ cm⁻². To test the stability of the polymer matrix, a newly prepared device was tested for 24 days. Figure R7 illustrated the relationship between the mechano-sensor's force sensitivity to external force (F_z) and time. Within two days post full curing, the solvent in the polymer matrix was completely volatilized, and the mechanical properties of the UV-crosslinked polymer matrix tended towards stability. This affirmed the reproducibility and stability of the device's performance.

Figure R7. Relations between the mean value of displacement sensitivity (S_D) varied with time under an external force F_z .

Comment 6: Line 135 *The mechano-sensor can detect touch, pressure, tickle and itch sensations applied.*

How it can sense tickle and itch, and how it can discriminate between the two, is difficult to understand. I think the biological parallelism may have been carried too far. The measurements only show a capacity to convert applied forces or displacements (actually these two inputs could be considered equivalent) into a shift in the detected wavelength.

Answer 6: We thank the reviewer for the positive suggestion, and we agree with it. In our experiments, we investigated the mechano-sensor's response to static forces mimicking long-term pressing and dynamic forces akin to touch, characterized by the instantaneous changes. The results not only demonstrate the conversion of applied forces or displacements but also highlighted the mechano-sensor's capability to identify mechanical vibration signals with different waveforms, displacements, and frequencies, resembling sensations of tickling and itching.

We acknowledge the concern that our initial description may have inadvertently emphasized the tickle and itch sensations excessively. To ensure the accurate conveyance of biological parallelism, we have revised our description in our revised manuscript "The mechano-sensor can detect touch and pressure sensations." on Page 7.

Comment 7: *Figure 1 e. The authors present a schematic diagram (an actual experimental measurement would be more appealing), but they do not explain why or when a red or blue shift is expected. They do not quantify the expected shifts. In general, the manuscript is difficult to follow as key concepts are not presented in an orderly and clear manner.*

Answer 7: We thank the reviewer for the constructive suggestions.

As per the reviewer's suggestion, we have restructured the manuscript to provide a more orderly and clear presentation of the key concepts. In Section 2.1 of the revised manuscript, we first briefly introduce the reason for mechano-sensing directionality to the external force F_r , that is, the presence of the fiber taper broke the mechanical circular symmetry (namely, the stress distribution, deformation, etc.) in the microbubble's equatorial section. Actual experimental results were then presented, demonstrating the mechano-sensor's response to the external forces F_r applied from two typical directions (i.e., $\varphi = 180^\circ$ and 0°).

Additionally, we have incorporated more detailed explanations and quantified results for the resonance wavelength shift based on FEM modeling. This information is now presented in Section 2.2 of the revised manuscript and in the response to Question 17/Reviewer #1.

We have added and revised the relevant discussion in the revised manuscript as:

"The fiber taper functions similarly to nerve fibers connecting the MSHS and the central nervous system in the biological archetype. More importantly, when an external force F_r with a component in the plane of the fiber taper and its coupled microbubble equator (or, a component perpendicular to the micro-hair) is applied to the head of the micro-hair, the presence of the fiber taper breaks the mechanical circular symmetry (the stress distribution, deformation, etc.) in the microbubble's equatorial section. Therefore, the strain effect and spectral response of the microbubble are different for F_r applied in different directions, which is the main reason for the directionality of the mechano-sensor." on Page 6.

"In a typical test, the resonance wavelength remained constant in the initial vertical state of the mechano-sensor micro-hair ($F_r = 0$ mN). If the mechano-sensor micro-hair was swung or bent in other directions under an external force (i.e., $F_r = 0.137$ mN ($\varphi = 180^\circ$) and 0.128 mN ($\varphi = 0^\circ$), respectively), the resonance wavelength was shifted in the opposite direction due to the strain effect on the microbubble (Fig. 1f)." on Page 8.

Fig. 1. (f) Responses of the bioinspired mechano-sensor in different deformation states (i.e., $F_r = 0$ mN, 0.137 mN ($\varphi = 180^\circ$), and 0.128 mN ($\varphi = 0^\circ$)).

Comment 8: Line 172 Stress distributions..... due to external forces have been characterised by FEM.

I would not use the term characterised to describe the FEM simulations.

Answer 8: According to the reviewer's suggestions, we have revised "The stress distributions of the mechano-sensor without and with external force F_r (with a displacement of $20 \mu\text{m}$) were characterized by FEM and are shown in Figure 4c and 4d, respectively." to "A finite element method (FEM) model was used to analyze the responses of the mechano-sensor when external forces F_r were applied to the micro-hair." in the revised manuscript (Page 8).

Comment 9: Line 190 In the actual experiments, the fibre cone can be considered as an object with a higher refractive index than the UV adhesive.

Why did the authors not include this in the FEM simulations? The FEM simulations comparing a microbubble and a solid microsphere do not add much light to the devices or phenomena studied here. I do not see any simulation results about the change in wavelength,

only the stress distribution is shown in Figure 2. This is not a FEM model of the mechano-sensor as stated in the figure caption, just a simulation of the effect of forces in a glass vessel. The effect on the refractive index or wavelength shift is not explained or simulated.

Answer 9: We thank the reviewer for the constructive suggestions. According to the reviewer's suggestions, we have added the fiber taper into our updated Finite Element Method (FEM) model. The addition of the fiber taper is crucial, as it breaks the mechanical circular symmetry (namely, the stress distribution, deformation, etc.) in the microbubble's equatorial section. This ensures a more accurate representation of the mechano-sensor's behavior, especially regarding mechano-sensing directionality to the external force F_r .

We acknowledge the importance of elucidating the mechano-sensor's response in terms of the refractive index and wavelength shift. In our response to Question 3 from Reviewer #1, we demonstrated that the strain effect on the microbubble was the main reason for the resonance wavelength shift. Further, in our responses to the 4th and 17th questions from Reviewer #1, we delved into more detailed simulations, elucidating the relationship between the applied forces and the resulting resonance wavelength shifts.

The field distributions of the radial strain and strain-induced effective refractive index change along the equatorial cross-section of the microbubble were determined through FEM simulations. The radial strain dR/R and the effective refractive index change dn_{eff}/n_{eff} in the microbubble's equatorial cross-section were the averages of their field distribution integrals, leading to the calculation of the resonance wavelength shift ($d\lambda/\lambda$).

Figure R8. FEM models including the fiber taper.

Comment 10: Figure 3a shows the typical spectrum of the WGM microbubble resonator. How typical is that? Can you show some raw data spectra? How does this vary from device to device

due to manufacturing differences in the dimensions of the bubble and walls? The authors state that the Q factor is 1.0×10^7 , but why is this relevant? how does it translate into the performance of the device?

Answer 10: We thank the reviewer for the positive suggestions.

10.1 For term "Typical":

Answer 10.1: Figure R9b showed the spectral evolution of the mechano-sensor as the external force F_r ($\varphi = 180^\circ$) increased from 0.129 mN to 0.720 mN. Each dip in the spectrum corresponded to an excited whispering-gallery-mode (WGM), and the mode field distributions varied for different WGMs. The higher the mode energy ratio in the microbubble wall is, the stronger the confinement of the mode energy by the resonator, and the higher the Q factor of the WGM. When we excite the WGMs of the microbubble through a fiber taper, WGMs with high and low Q factors were excited, so there was no "typical" result. We have deleted the term "typical" in our revised manuscript.

10.2 For raw data spectra:

Answer 10.2: According to the reviewer's suggestions, we have included the Fig. 3b and Fig. 5b in the revised manuscript, which presented the raw data spectra. This addition provided a more detailed and visualized responses of the mechano-sensor, and the raw data spectra was shown in Figure R9.

The microbubble's diameter determined the free spectral range (FSR) of the excited WGMs, with little impact on the Q factor unless there is a significant order of magnitude change. The thinner the wall thickness of the microbubble is, the weaker the binding on the mode, and the number of modes with high Q factors will be reduced.

While our device fabrication involves manual processes that may introduce manufacturing errors, it is noteworthy that mechanical sensing, unlike biochemical sensing, is reversible and dynamic. The manufacturing differences can be addressed through calibration experiments with various optional WGMs. The current configuration is only a proof-of-concept prototype, and we are committed to minimizing manufacturing differences in future iterations through the implementation of micro- and nanomachining technologies.

Figure R9. Experiments of the mechano-sensor under an external force F_r . **(a)** Transmission spectra for the mechano-sensor. The Q factor of the tracked WGM is approximately 1.8×10^7 . **(b)** Spectra evolution of the mechano-sensor as the external force F_r ($\varphi = 180^\circ$) increased from 0.129 mN to 0.720 mN.

To classify this issue, we have added the relevant figures into the revised manuscript as:

"Fig. 3. (b) Spectral evolution of the mechano-sensor as the external force F_r ($\varphi = 180^\circ$) increased from 0.129 mN to 0.720 mN." on Page 30.

"Fig. 5. (a) Spectral evolution of the mechano-sensor as the external force F_z increased from 0.002 mN to 3.149 mN." on Page 32.

10.3 For relevance of Q Factor:

Answer 10.3: We have provided additional information on the relevance of the Q factor to the

performance of the device. The force sensitivity ($S_F = d\lambda/dF$) and Q factor ($Q = \lambda/d\lambda$) jointly determine the minimum measurement resolution [*Appl. Opt.* **47**, 3009, (2008).; *Journal of applied physics* **105**.1, (2009).]. If we assume that the minimum measurable wavelength shift is $\Delta\lambda = \lambda/Q$, the resolution of force measurement is:

(R6)

The high Q factor is crucial for improving the resolution of force measurements and theoretical detection limits.

We discuss this issue in the revised manuscript as:

"The quality factor (Q factor) of the tracked WGM was approximately 1.8×10^7 , as shown in Fig. 3a. Notably, a higher Q factor can enhance the sensing resolution due to its narrower spectral linewidth." on Page 11.

"The standard deviation (σ) of the resulting spectral variation can be approximated by^{41,42}:

(5)

where $\Delta\lambda$ is the FWHM of the resonance dip, which is related to the Q factor. SNR is the signal-to-noise ratio. The WGM shown in Fig. 3a had an SNR of 48.7 dB and a σ of 0.0012 pm. The theoretical detection limits of the displacement and force measurements were calculated as $DL_D = 3\sigma/S_D \approx 70$ nm and $DL_{Fr} = 3\sigma/S_{Fr} \approx 0.9$ μ N, respectively." on Page 12.

Comment 11: *Line 230: "The mechano-sensor cannot recover from a relatively large deformation in a short time due to the overloaded external force or displacement. I do not understand this sentence. Is the long recovery time a property of the sensor or an artefact of the way the sensor response is measured? Is this overload related to the force application setup or to the sensor under test?"*

Answer 11: We thank the reviewer for the positive suggestions. First, the response time should be measured under transient external force changes, not slow force applications. Figure R10 (F_r , $\varphi = 180^\circ$) showed the resonance wavelength shift resulting from micro-hair bending. The object applying the external force was driven by the stepper motor. In this setup, the mechano-sensor's

measured response time was primarily determined by the stepper motor movement time. The force and displacement applied to the micro-hair were proportional to the movement time. In detail, $t_1 = 1.32$ s and $t_2 = 7.82$ s represent the stepper motor movement from initial position (P_0) to P_1 and from P_5 to P_0 , respectively. The relatively long recovery time is essentially the accumulation of the five deformation times. From this perspective, the long recovery time is an artefact of how the sensor response is measured.

Furthermore, we observed a trailing response when the sensor undergoes significant external force and displacement. This behavior may result from the accumulated elastic deformation of the polymer matrix and its slower recovery, but it was certainly not overload. The polymer matrix chosen for the experiment had favorable characteristics, including a low refractive index ensuring the presence of WGM resonance. And the convenience and rapidity of UV curing were particularly beneficial for packaging finely coupled systems consisting of a microbubble and fiber taper. Notably, the choice of polymers with higher elastic modulus can suppress the tailing response but may reduce sensitivity, so there was a trade-off between response time and force sensitivity.

Figure R10. Responses of the mechano-sensor obtained by tracking the wavelength shift (red curve) under various static forces F_r ($\varphi = 180^\circ$, blue curve).

We have removed "The mechano-sensor cannot recover from a relatively large deformation in a short time due to the overloaded external force or displacement." in our revised manuscript.

Comment 12: *When discussing the force sensitivity of the mechanosensor, the authors do not discuss the precision of the wavelength shift measurement. In fact, no details are given on how they measure the wavelength shift. In Figure 3a I can see the measured intensity in a-u for different wavelengths. How is the wavelength shift measured? Is it just a decrease in light intensity for a given detection wavelength? Are the full spectra recorded? I would like to see more raw data measurements.*

Answer 12: We thank the reviewer for the positive suggestions.

12.1 For mechano-sensor experimental setup:

Answer 12.1: As shown in Figure R11a, the light produced by a tunable laser (780 nm, UniQuanta, China) was coupled into the fiber and transmitted to the fiber taper for coupling with the microbubble. The polarization state of the laser light was adjusted with a polarization controller. When the scanning wavelengths of the laser light met the resonant condition of the microbubble resonator, the light energies of the resonance wavelengths were coupled into the microbubble, resulting in valleys at the resonance wavelengths in the transmission spectrum. Because the lifetime of photons in the resonator was not infinite, the valleys in the transmission spectrum appeared as Lorentz lines. The longer the lifetime of the photons was, the smaller the full width at half maximum (FWHM) of the Lorentz line, and the higher the Q factor of the WGM. The WGM signals from the device (mechano-sensor) were monitored by a photodetector (APD430A/M, Thorlabs) and recorded by a data acquisition card (PCIe 6351, National Instruments). A computer program was used to locate the positions (minima) of the spectra after Lorentzian fitting to determine the exact resonance wavelengths of the WGMs (Figure R11b).

We have added in the revised manuscript as:

" A computer is used to locate the positions (minima) of the spectra after Lorentzian fitting to determine the exact resonance wavelengths of the WGMs." on Page 18.

Figure R11. (a) Experimental setup of the all-optical mechano-sensor system. (b) WGM transmission spectra for the mechano-sensor. (c) Spectra evolution of the mechano-sensor as the external force F_r ($\varphi = 180^\circ$) increased from 0.129 mN to 0.720 mN. (d) Spectra evolution of the mechano-sensor as the external force F_z increased from 0.002 mN to 3.149 mN.

12.2 For force measurement:

Answer 12.2: Static forces were applied to the mechano-sensor micro-hair along the r -axis or z -axis of the mechano-sensor. The static force applied to the mechano-sensor micro-hair was given by a commercial force sensor (FA597, FIBOS Measurement Technology, China). The commercial force sensor was mounted on a 3D stepper motor stage (9063-XYZ-PPP-M, Newport) and equipped with a digital force gauge (SBT970, Simbatouch, China). The object applying the external force was driven by the stepper motor, and the wavelength shift and the static force change were recorded by a computer and a digital force gauge in real time, respectively. The displacement value was obtained by the stepper motor. Piezo actuators (P-280.20, Physik Instrumente (PI), Germany) were used for the touch sensing and durability tests. We have added the relevant discussion in the revised manuscript as:

"The displacement value was obtained by the stepper motor. Schematic diagrams of the external forces applied to the mechano-sensor micro-hair along the r -axis and z -axis of the mechano-sensor are shown in Supplementary Fig. 8 and 12 (SI). Piezo actuators (P-280.20, Physik Instrumente (PI), Germany) were used for the touch sensing and durability tests." on Page 18.

We have added the relevant discussion in the revised Supplementary Information as:

"To measure the directional characteristics of the mechano-sensor with external force (F_r), the mechano-sensor mounted on a rotating device, which drove the mechano-sensor to rotate 360° around the z-axis. The displacement and force sensitivities of the mechano-sensor were measured at 10° intervals, and the directional diagram of the mechano-sensor was finally obtained (Supplementary Fig. 9). The static force applied to the mechano-sensor micro-hair was given by a commercial force sensor. The commercial force sensor was mounted on a 3D stepper motor stage and equipped with a digital force gauge. The object applying the external force was driven by the stepper motor, and the wavelength shift and the static force change were recorded by a computer and a digital force gauge in real time, respectively. The displacement value was obtained by the stepper motor. The inset shows the position of the force action point (L)." on Page S15 (Part 5 of SI).

Supplementary Fig. 9. Measurement schematic diagram of the external force (F_r) applied to the mechano-sensor micro-hair in the r - φ plane. Inset: Front view of the mechano-sensor.

"As shown in the measurement schematic diagram in Supplementary Fig. 12, the external force (F_z) applied along the z-axis of the mechano-sensor was the same as that shown in Supplementary Fig. 8, except that the direction of the applied force was changed." on Page S19 (Part 7 of SI).

Supplementary Fig. 12. Measurement schematic diagram of the external force (F_z).

12.3 For the precision of the wavelength shift measurement:

Answer 12.3:

We recorded the full spectra of the mechano-sensor as the external forces (F_r, F_z) increased, as shown in Figure R11c-d. The resonance wavelength of the tracked WGM was changed by the external forces. The precision of the wavelength shift measurement, that is, the standard deviation of the wavelength shift, can be approximated by [*Opt. Express* **16**, 1020–1028 (2008).; *Opt. Lett.* **48**, 1922–1925 (2023).]:

(R7)

where $\Delta\lambda$ is the FWHM of the WGM, which is related to the Q factor. SNR is the signal-to-noise ratio. The WGM shown in Figure R11b has an SNR of 48.7 dB and a σ of 0.0012 pm.

Comment 13: *The only plot where I can see intensity versus wavelength is Figure 4 a, and then all the plots are relative wavelength shift versus time. I would like to see how the full spectra change during the experiments. The full intensity versus wavelength spectra for the different experimental situations, just before displacement or force is applied, during force application, after force application... Do the spectra stay the same with just a shift in the peaks? Do the peaks broaden? This is not clear from the data presented. Also, why did the authors choose this resonance peak (in red in Fig. 4a) and not the deeper neighbouring one? An explanation of the physics behind the sensor performance and experimental details are missing.*

Answer 13: We thank the reviewer for the positive suggestions.

13.1 For spectral evolution:

Answer 13.1:

Figures R12a and R12c show the spectral evolution of the mechano-sensor with the increasing external forces F_r ($\varphi = 180^\circ$) and F_z , respectively. Notably, the spectra remained approximately the same, with only a shift in the dips. Additionally, Figures R12b and R12d illustrate variations in the full width at half maximum (FWHM) of the tracked WGM as external forces F_r ($\varphi = 180^\circ$) and F_z increased, respectively. The FWHM varied intricately as the external forces increased, which can be attributed to complex influence factors (i.e., the strain effect of the microbubble, the scattering loss caused by the strain-induced refractive index change, and the change in the gap between the microbubble and the fiber taper).

Figure R12. (a) Spectral evolution of the mechano-sensor as the external force F_r ($\varphi = 180^\circ$) increased from 0.129 mN to 0.72 mN. (b) Full width at half maximum (FWHM) varied intricately as the external force F_r ($\varphi = 180^\circ$) increased. (c) Spectral evolution of the mechano-sensor as the external force F_z increased from 0.002 mN to 3.149 mN. (d) Full width at half maximum (FWHM) varied intricately as the external force F_z increased.

We have added the relevant discussion in the revised manuscript as:

"When the displacement-applying object was driven by a stepper motor with a step value of 60 μm , the full width at half maximum (FWHM) varied intricately (Supplementary Fig. 8a, SI), while the resonance wavelength shift generated by the mechano-sensor increased in a stepwise manner during the displacement application." on Page 11.

13.2 For selection of the tracked WGM:

Answer 13.2: The choice of the tracked WGM was a result of comprehensive considerations. External forces induced a unidirectional shift in the resonance wavelengths of all excited WGMs, but their force sensitivities differed due to distinct mode field distributions.

It was difficult to identify the WGMs with high force sensitivities by directly observing the valley shape (i.e., coupling depth and Q factor of the WGM) in the transmission spectrum. To find the WGM with a high force sensitivity for tracking, pre-experiments were conducted. The micro-hair was substantially bent, and the resulting resonance wavelength shift was observed. The WGM exhibiting the most significant resonance wavelength shift was chosen as the tracked WGM.

Furthermore, the selection process took into consideration the FWHM (i.e., the Q factor) and signal-to-noise ratio (SNR) of the resonance dip. These considerations, essential for precise wavelength shift measurements, played a crucial role in determining the tracked WGM.

Comment 14: *What is the noise in the wavelength shift measurement?*

The authors quote a standard deviation from the blank test in line 234, but this should be described more carefully and the sources for this standard deviation in the signal when no force is applied should be explained. The experimental data that led to this number should be shown (either in supplementary or main text).

234 The standard deviation for the blank test was $\sigma = 0.01 \text{ pm}$. Therefore, the theoretical limits of detection for were calculated as $DL1 = 3\sigma/SD1 = 0.034 \mu\text{m}$ and $DL2 = 3\sigma/SFz = 0.03$ (...)

This needs to be explained. How are they obtained? Are these experimental values? If so, why are the limits of detection called theoretical?

Answer 14: We thank the reviewer for the positive suggestions. The theoretical detection limit (DL) is defined as $DL = 3\sigma/S$, where σ is the standard deviation of resulting spectral variation and

S is sensitivity. The standard deviation of the resulting spectral variation σ can be approximated by:

(R8)

We have added in the revised manuscript as:

"The standard deviation (σ) of the resulting spectral variation can be approximated by [*Opt. Express* **16**, 1020–1028 (2008).; *Opt. Lett.* **48**, 1922–1925 (2023).]:

(5)

where $\Delta\lambda$ is the FWHM of the resonance dip, which is related to the Q factor. SNR is the signal-to-noise ratio. The WGM shown in Fig. 3a had an SNR of 48.7 dB and a σ of 0.0012 pm. The theoretical detection limits of the displacement and force measurements were calculated as $DL_D = 3\sigma/S_D \approx 70$ nm and $DL_{F_r} = 3\sigma/S_{F_r} \approx 0.9$ μ N, respectively." on Page 12.

Comment 15: *Figure 4 j. The applied force is quite noisy, but the wavelength shift does not reflect this. This should be explained.*

Answer 15: We thank the reviewer for the positive suggestions. In fact, the static force applied remained constant at each step, without fluctuations. The observed significant force noise was attributed to the poor performance of the previous commercial force sensor (SBT970, Simbatouch, China). Therefore, it was expected that the wavelength shift might not reflect the fluctuations in the noisy force values.

To address this issue, we replaced the SBT970 with a new commercial force sensor (FA597, FIBOS Measurement Technology, China), effectively reducing the noise level. The test results, presented in Figure R13, clearly illustrated a substantial suppression of noise in the force measurements.

Figure R13. Comparison of noise levels between the previous and new commercial force sensors.

We have revised "The static force applied to the mechano-sensor micro-hair was given by a commercial force sensor (FA597, FIBOS Measurement Technology, China)" (Page 18 in revised manuscript) and replaced the measured data in Figure 3 and Figure 5.

Comment 16: *This manuscript claims a 3D mechano-sensor. The authors measure different sensitivities depending on the position of the force applied along the capillary and the directions, but it is not clear how a vectorial mechano-sensor is obtained. How does the device discriminate between forces applied at different positions and angles? Several experimental situations may result in the same wavelength shift. If this is not the case, this should be explained and proven with experiments.*

Line 353 states that the mechano-sensor has the ability to identify/assess the direction of an external displacement/force within a 180° range.

I do not understand how the sensor can judge the direction of the force, a given value of the wavelength shift could correspond to a large force from one direction or a smaller force but from a different direction. How can the sensor tell the difference from just reading the wavelength shift?

Answer 16: We thank the reviewer for the positive suggestions. For this type of mechano-sensors, the directionality can be expressed as [*Microsyst. Nanoeng.* **9**, 65 (2023).]:

where G' and G are the maximum and minimum force sensitivity values measured in the directionality of the mechano-sensor, respectively.

With a good directionality of 32.31 dB, the force sensitivity of our mechano-sensor depended on the direction of the applied external force within 360° in the radial plane of the micro-hair. Moreover, our mechano-sensor can perceive the mechanical stimuli in the axial direction of the micro-hair. These unique features distinguished our mechano-sensor as a multidirectional sensor, distinct from the scalar mechano-sensors.

In our previous manuscript, the term "3D mechano-sensor" referred to the 3D assembly configuration of the device. We have modified the term from "vectorial mechano-sensor" to "multidirectional mechano-sensor" and omitted the term "3D mechano-sensor". This modification better reflected the one-dimensional vector property of our device. It can indicate the directional scope of an external force within 360° in the radial plane of the micro-hair based on the direction of the resonance wavelength shift. However, it was important to clarify that our device cannot accurately determine a force of any magnitude in any direction currently.

Additionally, we acknowledge the reviewer's insightful suggestion about a 3D vector sensing system. We are actively developing such a system by utilizing devices with multiple equators or arrays of multiple devices. However, achieving this goal requires more controllable micro-and nano-machining technologies, extending beyond the scope of the current proof-of-concept study.

In response to reviewer's query about how the device discriminates between forces applied at different positions, we made the following clarifications. To quantitatively study the mechano-sensor's characteristics for device optimization and customization, we measured its force and displacement sensitivities to external forces (F_r) at different axial positions. However, in practical applications, the biomimetic mechano-sensor works in the same way as the biological version, that is, an external force usually acts directly on the entire mechano-sensor hair or the tip of the hair, and the action point is not significant.

We have added and revised related discussion in revised Supplementary information as:

"To measure the directional characteristics of the mechano-sensor with external force (F_r), the mechano-sensor mounted on a rotating device, which drove the mechano-sensor to rotate 360° around the z -axis. The displacement and force sensitivities of the mechano-sensor were measured at 10° intervals, and the directional diagram of the mechano-sensor was finally obtained (Supplementary Fig. 9)." on Page S15 (Part 5 in SI).

We have added and revised related discussion in the revised manuscript as:

"By changing the angle of the external force applied in the r - φ plane (rotated around the z -axis) at 10° intervals (Supplementary Fig. 9, SI), the sensitivity of the mechano-sensor to displacements and static forces applied in different directions (φ) was determined (maintaining $L = 9$ mm). As shown in Fig. 3f, the displacement and force sensitivities had maximum positive values when $\varphi = 180^\circ$, maximum negative values when $\varphi = 0^\circ$ or 360° , and values of approximately zero when $\varphi = 90^\circ$ and 270° . These experimental results corresponded well to the FEM simulation results (Fig. 2f), indicating that the mechano-sensor performance effectively depends on the direction of the applied stimulation. The directionality of the mechano-sensor can be expressed as⁴³:

$$(6)$$

where G' and G are the maximum and minimum force sensitivity values measured in the directionality of the mechano-sensor, respectively. The measured directional results show that the proposed sensor has a directionality of 32.31 dB in the range of $\varphi = 0\sim 360^\circ$ when $L = 9$ mm." on Page12.

Fig. 2. (f) Relations between the direction of the external force F_r and dR/R , dn_{eff}/n_{eff} , and the displacement sensitivity (S_D).

Fig. 3. (f) Displacement (red curve) and force (blue curve) sensitivities of the mechano-sensor

when the external force F_r is applied in different directions (φ).

Comment 17: *From Figure 5, I can see that a red or blue shift is measured depending on the direction of the force, but this is only shown (proven) for two very specific directions (180 degrees and 0 degrees). A clear explanation is needed, and FEM simulations of this would be more useful than the hard sphere simulations shown in Figure 2e (since no experimental devices composed of hard spheres are shown). I can infer that one angle produces a shorter gap and the other a larger gap and this can cause the red and blue shifts, but I would like to see this explained by the authors, the quantification of the shifts and some modeling, simulations and comparison with measurements carried out. Can the expected shift be predicted for known devices? What happens to angles different to 0 and 180 deg?*

Answer 17: We thank the reviewer for the positive suggestions. In our reply to Question 3 / Reviewer #1, we concluded that the strain effect on microbubble is the main reason for the shift in the resonance wavelength. In response to this question, we presented a detailed explanation and Finite Element Method (FEM) simulations addressing the directionality of the mechano-sensor to external forces F_r . In the simulation, the corresponding displacement of external force F_r was set as 2 μm . The direction of the external force F_r applied to the mechano-sensor micro-hair is shown in Figure R14a.

17.1 For external forces F_r from $\varphi = 180^\circ$:

Answer 17.1: When an external force F_r was applied to the micro-hair in the direction of $\varphi = 180^\circ$, the resulting stress field distribution in the equatorial cross-section of the mechano-sensor was shown in Figure R14c. The microbubble and fiber taper moved in the $\varphi = 180^\circ$ direction. The stress on the microbubble wall was over 2-3 orders of magnitude higher than that on the polymer matrix, and therefore, the stress effect of the polymer matrix was neglected. The radial strain field distribution (Figure R14d) and the strain components (i.e., ε_x , ε_y and ε_z) along the microbubble's equatorial cross-section can be directly obtained through FEM simulations. According to Equation (R5), by using the reported values of $p_{11} = 0.131$ and $p_{12} = 0.26$ for fused silica, the field distribution of the strain-induced effective refractive index change was derived, as shown in Figure R14e. The radial strain field distribution was negative in the range of $\varphi = 90^\circ$ to 270° in the equatorial cross-section but positive in the range of $\varphi = 270^\circ$ to 450° (namely, 90°)

in the equatorial cross-section. The radial strain field distribution along the microbubble's equatorial cross-section had a maximum negative value at $\varphi = 180^\circ$ (that is, the same direction of F_r), a maximum positive value at $\varphi = 0^\circ$ (that is, the opposite direction of F_r), and a value of approximately zero at $\varphi = 90^\circ$ and 270° (that is, the direction perpendicular to F_r). The radial strain dR/R and the effective refractive index change dn_{eff}/n_{eff} in the microbubble's equatorial cross-section were the averages of their field distribution integrals. The dR/R and dn_{eff}/n_{eff} values were 1.232×10^{-8} and -2.5×10^{-9} , respectively. Accordingly, the calculated wavelength shift $d\lambda/\lambda$ was 0.982×10^{-8} . As a comparison, the hollow microbubble was replaced by a solid microsphere of the same size, and the dR/R and dn_{eff}/n_{eff} values of the microsphere-based mechano-sensor were much smaller than those of the microbubble-based mechano-sensor, as shown in Figure R14f. The results in Figure R14f also indicated that for the displacement ($\varphi = 180^\circ$) range considered, dR/R was larger than dn_{eff}/n_{eff} , and the dependence of $d\lambda/\lambda$ on the displacement (i.e., force) was essentially linear.

In theory, for a microbubble with a circular symmetric equatorial cross-section, the field distributions of both the radial strain and strain-induced effective refractive index change should be antisymmetric (i.e., $dR/R \approx 0$, $dn_{eff}/n_{eff} \approx 0$), regardless of the direction of the force F_r applied to the micro-hair. However, the presence of the fiber taper broke the geometrical circular symmetry of the microbubble's equatorial cross-section, amplifying the radial strain at the coupling position (that is, $\varphi = 0^\circ$ in the microbubble's equatorial cross-section), resulting in $dR/R > 0$. Moreover, dR/R was greater than dn_{eff}/n_{eff} , inducing a redshift in the resonance wavelength.

Figure R14. Simulations of the mechano-sensor under an external force F_r ($\phi = 180^\circ$). **(a)** Schematic illustration of the experimental device (left) and FEM model of the mechano-sensor (right) under an external force F_r ($\phi = 180^\circ$). A cylinder was used to apply the force in the FEM model. Inset: Top view of the mechano-sensor. **(b)** Schematic illustration of the deformation of the flexible microbubble under an external force F_r . Negative and positive radial strains were observed in the same and opposite directions of F_r , respectively. The fiber taper amplified the strain of the microbubble at the coupling position. **(c)** Stress field distribution in the r - ϕ plane of the mechano-sensor under an external force F_r ($\phi = 180^\circ$). **(d)** and **(e)** Field distributions of the radial strain and strain-induced effective refractive index change along the microbubble's equatorial cross-section under an external force F_r ($\phi = 180^\circ$). The dR/R and $dn_{\text{eff}}/n_{\text{eff}}$ values in the microbubble's entire equatorial cross-section were the averages of their field distribution integrals. **(f)** Comparisons of dR/R and $dn_{\text{eff}}/n_{\text{eff}}$ between the microbubble-based mechano-sensor and the microsphere-based mechano-sensor under different external displacements ($\phi = 180^\circ$).

We have added relevant discussion in the revised manuscript as:

"A finite element method (FEM) model was used to analyze the responses of the mechano-sensor when external forces F_r were applied to the micro-hair. The label L in Fig. 2a represents the length of the micro-hair subjected to the external forces F_r . The direction of the external force

F_r applied to the mechano-sensor in the r - φ plane is shown in Fig. 2a. The elastic modulus E and the Poisson's ratio ν of the polymer matrix were set as 5.2 MPa and 0.41, respectively. As a stimulus input in the simulation, an external force F_r (blue arrow) was applied to the micro-hair along the r -axis of the mechano-sensor (Fig. 2b). The direction of the external force F_r was set as $\varphi = 180^\circ$, and the corresponding displacement was set as 2 μm . The stress field distribution in the equatorial cross-section of the mechano-sensor is shown in Fig. 2c. The microbubble and fiber taper moved in the $\varphi = 180^\circ$ direction. The stress on the microbubble wall was over 2-3 orders of magnitude higher than that on the polymer matrix; therefore, the stress effect of the polymer matrix was neglected. The radial strain field distribution (Fig. 2d) and the strain components (i.e., ε_x , ε_y and ε_z) along the microbubble's equatorial cross-section can be directly obtained in the FEM simulation. According to Equation (3), by using the reported values of $p_{11} = 0.131$ and $p_{12} = 0.26$ for fused silica, the field distribution of the strain-induced effective refractive index change was derived, as shown in Fig. 2e. The radial strain field distribution was negative in the range of $\varphi = 90^\circ$ to 270° in the equatorial cross-section but positive in the range of $\varphi = 270^\circ$ to 450° (namely, 90°) in the equatorial cross-section. The radial strain field distribution along the microbubble's equatorial cross-section had a maximum negative value at $\varphi = 180^\circ$ (that is, the same direction of F_r), a maximum positive value at $\varphi = 0^\circ$ (that is, the opposite direction of F_r), and values of approximately zero at $\varphi = 90^\circ$ and 270° (that is, the direction perpendicular to F_r). The radial strain dR/R and the effective refractive index change $dn_{\text{eff}}/n_{\text{eff}}$ in the microbubble's equatorial cross-section were the averages of their field distribution integrals, which are indicated in Fig. 2d and e, respectively. In theory, for a microbubble with a circular symmetric equatorial cross-section, the field distributions of the radial strain and strain-induced effective refractive index change should be antisymmetric (i.e., $dR/R \approx 0$, $dn_{\text{eff}}/n_{\text{eff}} \approx 0$), regardless of the direction of the force F_r applied to the micro-hair. However, the presence of the fiber taper breaks the geometrical circular symmetry of the microbubble's equatorial cross-section, amplifying the radial strain at the coupling position (that is, $\varphi = 0^\circ$ in the microbubble's equatorial cross-section), resulting in $dR/R > 0$. Moreover, dR/R was greater than $dn_{\text{eff}}/n_{\text{eff}}$, inducing a redshift in the resonance wavelength." on Page 8.

Fig. 2. Simulations of the mechano-sensor under an external force F_r . **(a)** Schematic illustration of the experimental device (left) and FEM model of the mechano-sensor (right) under an external force F_r ($\varphi = 180^\circ$). A cylinder was used to apply the force in the FEM model. Inset: Top view of the mechano-sensor. **(b)** Schematic illustration of the deformation of the flexible microbubble under an external force F_r . Negative and positive radial strains were observed in the same and opposite directions of F_r , respectively. The fiber taper amplified the strain of the microbubble at the coupling position. **(c)** Stress field distribution in the r - φ plane under an external force F_r ($\varphi = 180^\circ$). **(d)** and **(e)** Field distributions of the radial strain and strain-induced effective refractive index change along the microbubble's equatorial cross-section under an external force F_r ($\varphi = 180^\circ$). The dR/R and dn_{eff}/n_{eff} values in the microbubble's equatorial cross-section were the averages of their field distribution integrals. **(f)** Relations between the direction of the external

force F_r and dR/R , dn_{eff}/n_{eff} , and the displacement sensitivity (S_D). (g) Comparisons of dR/R and dn_{eff}/n_{eff} between the microbubble-based mechano-sensor and the microsphere-based mechano-sensor under different external displacements ($\varphi = 180^\circ$). (h) Relations between S_D and the capillary size (i.e., radius ($R_{capillary}$) and wall thickness ($T_{capillary}$)) under an external force F_r ($\varphi = 180^\circ$). The solid black line corresponds to the contour of $R_{bubble} = 190 \mu\text{m}$. The star label corresponds to the $R_{capillary}$ and $T_{capillary}$ selected for the micro-hair. (i) Relations between S_D and the mechanical properties of the polymer matrix (i.e., Poisson's ratio (ν) and elastic modulus (E)) under an external force F_r ($\varphi = 180^\circ$).

17.2 For external forces F_r from $\varphi = 0^\circ$:

Answer 17.2: When the external force F_r was applied to the micro-hair in the direction of $\varphi = 0^\circ$ (Figure R15a), the stress field distribution in the equatorial cross-section of the mechano-sensor was shown in Figure R15b. The microbubble and fiber taper moved in the $\varphi = 0^\circ$ direction, and the stress effect of the polymer matrix was negligible. The radial strain dR/R and the strain-induced effective refractive index change dn_{eff}/n_{eff} in the equatorial cross-section of the microbubble were the averages of their field distribution integrals (Figure R15c and R15d), with values of -1.228×10^{-8} and 2.495×10^{-9} , respectively. Accordingly, the calculated resonance wavelength shift $d\lambda/\lambda$ was -0.9785×10^{-8} .

The radial strain field distribution along the microbubble's equatorial cross-section had a maximum negative value at $\varphi = 0^\circ$ (that is, the same direction of F_r). The fiber taper amplified the radial strain at the coupling position (that is, $\varphi = 0^\circ$ in the microbubble's equatorial cross-section), resulting in $dR/R < 0$ and a blueshift in the resonance wavelength.

Figure R15. Simulations of the mechano-sensor under an external force F_r ($\varphi = 0^\circ$). **(a)** Schematic illustration of the experimental device (left) and FEM model of the mechano-sensor (right) under an external force F_r ($\varphi = 0^\circ$). **(b)** Stress field distribution in the r - φ plane of the mechano-sensor under an external force F_r ($\varphi = 0^\circ$). **(c)** and **(d)** Field distributions of the radial strain and strain-induced effective refractive index change along the equatorial cross-section of microbubble under an external force F_r ($\varphi = 0^\circ$), respectively.

We have added relevant discussion in the revised Supplementary Information as:

"A finite element method (FEM) model was used to analyze the responses of the mechano-sensor to applied external forces F_r . As a stimulus input in the simulation, an external force F_r was applied in the $\varphi = 0^\circ$ direction (Supplementary Fig. 3a), and the corresponding displacement was 2 μm . The stress field distribution in the equatorial cross-section of the mechano-sensor is shown in Supplementary Fig. 3b. The microbubble and fiber taper moved in the $\varphi = 0^\circ$ direction, and the stress effect of the polymer matrix was negligible. The radial strain dR/R and the strain-induced effective refractive index change dn_{eff}/n_{eff} in the equatorial cross-section of the microbubble were the averages of their field distribution integrals (Supplementary Fig. 3c and d). The radial strain field distribution along the microbubble's equatorial cross-section had a maximum negative value at $\varphi = 0^\circ$ (that is, the same direction of F_r), a maximum positive value at $\varphi = 180^\circ$ (that is, the opposite direction of F_r), and values of approximately zero at $\varphi = 90^\circ$ and 270° (that is, the direction perpendicular to F_r). The fiber taper amplified the radial strain at the

coupling position (that is, $\varphi = 0^\circ$ in the microbubble's equatorial cross-section), resulting in $dR/R < 0$ and a blueshift in the resonance wavelength." on Page S5 (Part 3 in SI).

17.3 For external forces F_r from $\varphi = 90^\circ$:

Answer 17.3: When an external force F_r was applied in the $\varphi = 90^\circ$ direction (Figure R16a), the stress field distribution in the equatorial cross-section of the mechano-sensor was shown in Figure R16b. The microbubble and fiber taper moved in the $\varphi = 90^\circ$ direction. The radial strain dR/R and the strain-induced effective refractive index change dn_{eff}/n_{eff} in the equatorial cross-section of the microbubble were the averages of their field distribution integrals (Figure R16c and R16d), and both values were close to 0. This was because the radial strain field distribution in the microbubble's equatorial cross-section had a value of zero at $\varphi = 0^\circ$ (that is, the direction perpendicular to F_r), resulting in the negligible strain amplification effect introduced by the fiber taper. There was no shift in the resonance wavelength when the external force F_r was applied in the $\varphi = 90^\circ$ or 270° direction.

Figure R16. Simulations of the mechano-sensor under an external force F_r ($\varphi = 90^\circ$). **(a)** Schematic illustration of the experimental device (left) and FEM model of the mechano-sensor (right) under an external force F_r ($\varphi = 90^\circ$). **(b)** Stress field distribution in the r - φ plane of the mechano-sensor under an external force F_r ($\varphi = 90^\circ$). **(c)** and **(d)** Field distributions of the radial strain and strain-induced effective refractive index change along the equatorial cross-section of microbubble under an external force F_r ($\varphi = 90^\circ$), respectively.

We have added relevant discussion in the revised Supplementary Information as:

"Then, an external force F_r was applied in the $\varphi = 90^\circ$ direction (Supplementary Fig. 4a), and the stress field distribution in the equatorial cross-section of mechano-sensor is shown in Supplementary Fig. 4b. The microbubble and fiber taper moved in the $\varphi = 90^\circ$ direction, and the stress effect of the polymer matrix was also negligible in this case. The field distributions of the radial strain and strain-induced effective refractive index change along the equatorial cross-section of the microbubble are shown in Supplementary Fig. 4c and d, respectively. The radial strain field distribution was negative in the range of $\varphi = 0^\circ$ to 180° in the equatorial cross-section but positive in the range of $\varphi = 180^\circ$ to 360° . When the radial strain field integral distributed within the equatorial cross-section of the microbubble were averaged, dR/R and $dn_{\text{eff}}/n_{\text{eff}}$ were both approximately zero. This was because the radial strain at the coupling position (that is, $\varphi = 0^\circ$ in the microbubble's equatorial cross-section) was approximately zero, resulting in a negligible strain amplification effect introduced by the fiber taper. There was no resonance wavelength shift when the external force F_r was applied in the $\varphi = 90^\circ$ or 270° directions." on Page S6 (Part 3 in SI).

17.4 For external forces F_r from various directions:

Answer 17.4: Based on the above simulation results, it can be inferred that the dR/R depended on the radial strain at the coupling position (that is, $\varphi = 0^\circ$ in the microbubble's equatorial cross-section), which was related to the direction of force F_r . The dR/R , $dn_{\text{eff}}/n_{\text{eff}}$, and S_D (displacement sensitivity of the mechano-sensor) varied with the direction of force F_r (φ), as shown in Figure R17a. dR/R was greater than $dn_{\text{eff}}/n_{\text{eff}}$ and dominated $d\lambda/\lambda$. The dR/R had a maximum positive value (that is, maximum resonance wavelength redshift) when an external force F_r was applied in the $\varphi = 180^\circ$ direction, a maximum negative value (that is, maximum resonance wavelength blueshift) when an external force F_r was applied in the $\varphi = 0^\circ$ direction, and values of approximately zero (that is, no resonance wavelength shift) when an external force F_r was applied in the $\varphi = 90^\circ$ and 270° directions. Moreover, the simulation results of mechano-sensing directionality correspond well with the experimental results (Figure R17b).

Figure R17. (a) Simulations of the relationship between the directions of the external force (ϕ) and dR/R , dn_{eff}/n_{eff} and S_D . (b) Experimental results of the relationship between the directions of the external force (ϕ) and the displacement sensitivity (red curve) and force sensitivity (blue curve) of the mechano-sensor.

We have added relevant discussion in the revised manuscript as:

"When the direction of the force F_r changes, it can be inferred that the dR/R depends on the radial strain at the coupling position of the microbubble. dR/R had a maximum positive value (that is, the maximum resonance wavelength redshift) when an external force F_r was applied in the $\phi = 180^\circ$ direction, a maximum negative value (that is, the maximum resonance wavelength blueshift) when an external force F_r was applied in the $\phi = 0^\circ$ direction, and values of approximately zero (that is, no resonance wavelength shift) when an external force F_r was applied in the $\phi = 90^\circ$ and 270° directions (see more details in Supplementary Fig. 3 and 4 in the SI). The dR/R , dn_{eff}/n_{eff} , and S_D (displacement sensitivity of the mechano-sensor) varied with the direction of force F_r (Fig. 2f). dR/R was greater than dn_{eff}/n_{eff} and dominated $d\lambda/\lambda$." on Page 9.

Comment 18: *Figure 5 b and Figure 5 f. Why does the sensor respond to the breeze or hitting an obstacle with multiple peaks? Why are there three or five peaks in different experiments?*

Answer 18: We thank the reviewer for the positive suggestions. In the experiment, compressed air was used to blow air on the micro-hair. Each instance of blowing air induced a wavelength shift, evident as a peak in Figure R18a. Repeatedly blowing air on the micro-hair led to the appearance of multiple peaks in the real-time curve. Similarly, in the impact experiment (as demonstrated in Supplementary Movie 1), each impact of the micro-hair against an obstacle

resulted in the observation of a peak (Figure R18b).

Figure R18. (a) Responses of the mechano-sensor to breezes in the $\varphi = 180^\circ$ and 0° directions. (b) Responses of the mechano-sensor when hitting an obstacle while advancing ($\varphi = 180^\circ$) and retreating ($\varphi = 0^\circ$).

We have revised "Compressed air (5 bar) was used to produce a breeze three times towards the mechano-sensor micro-hair along the $\varphi = 180^\circ$ and 0° directions, which caused bending and deformation of the mechano-sensor micro-hair and induced an opposite shift in the resonance wavelength." in our revised manuscript (Page 15).

Comment 19: Line 36, line 43 attitude, I do not understand what the authors are referring to here.

Answer 19: We thank the reviewer for the positive suggestions. In our manuscript, when referring to the capabilities of the mechano-sensor in perceiving motion states such as acceleration, deceleration, and turning, we initially used the term "attitude". However, we recognize that this term may introduce ambiguity. Consequently, we have revised the description in our updated manuscript for clarity.

We have revised the sentence "Optics and photonics systems for environmental perception applications, namely, artificial sensing, have become increasingly used in photonic noses and tongues^{1,2}, photonic skins³, and laser-based gyroscopes and LiDAR systems^{4,5} for detecting molecules, pressures, spatiotemporal locations, visual information, and other data." and "In addition to visual perception, mechanical perception capabilities such as tactile and auditory sensing of forces and acoustic vibrations, especially those determining directionality or even vector properties (namely, signal amplitudes or strength and direction), are essential sensory

functions of many living organisms¹⁴ and important in artificial systems¹⁵⁻¹⁹."(Page 3 in the revised manuscript).

In summary, we have addressed all comments made by the Reviewer #1. Accordingly, the manuscript and supplementary materials have been carefully revised. We hope these answers and efforts yield a more complete and insightful manuscript, and we are also very grateful for the reviewer's essential contribution in motivating these improvements.

Reply to Reviewer #2

The authors report an all-optical force sensor inspired by mechano-sensitive biological sensilla, like mammalian whiskers. The principle of operation is an optomechanical coupling in between a microbubble and a tapered optical fiber. The microbubble is attached to a hollow glass capillary, which is used as the transducer of the sensor device. I think that the work is very interesting and innovative. I especially appreciate the integration of the device, the design is elegant and makes sense, plus the chosen application is very relevant. I really think that the presented work could be of interest for the community. However, I cannot support the publication of the work at this time because I have identified some deficiencies in the manuscript related to both the clarity of presentation of the results and their interpretation. These are my comments that should be answered by the authors before my final recommendation.

We would like to thank the reviewer for the constructive comments and are delighted that the reviewer is interested in our results. We are happy to address all the comments below. We thank the reviewer for motivating those improvements.

Comment 1: *The figures are not too clear. They have too many sub-panels that make it hard to follow them in the main text. I strongly suggest that the authors consider reducing the size of the figures or separating them into several.*

Answer 1: We thank the reviewer for the positive suggestions. We have made significant revisions to the figures, aiming to enhance clarity and improve the overall readability of the manuscript. We have either reduced the size of the figures or separated them into multiple sub-panels.

Comment 2: *The values of sensitivity given by the authors are, somehow, meaningless. As a general consideration, when you are designing a force sensor, the sensitivity depends on the actual dimensions and material of the transducer, in this case the glass capillary. What is the spring constant of the system? I am pretty sure that the value obtained in this work can easily be exceeded by simply making the capillary longer, or thinner. The authors have to include a*

parametric sweep at the beginning of the study to justify the chosen geometry.

Answer 2: We thank the reviewer for the constructive suggestions.

As described in our responses to Question 3/Reviewer #1 and Question 6/Reviewer #2, the solid microsphere-based mechano-sensor was insensitive to external force, and the microbubble-based mechano-sensor, encapsulated with a polymer matrix possessing an intermediate value of the elastic modulus, had the highest force sensitivity. These experimental results indicated that the strain effect on the microbubble was the main reason for the shift in the resonance wavelength.

Furthermore, we conducted Finite Element Method (FEM) modeling to optimize the geometric parameters of the glass capillary and microbubble, which significantly influenced the displacement and force sensitivities. According to the actual fabrication procedures, the microbubble was the result of spherical expansion of the fused silica capillary. Therefore, the geometric parameters of the microbubble and capillary were interconnected and should be jointly optimized. The curves equation for the inner and outer contours of the microbubble produced by the fiber fusion splicer (FSU-975) can be formulated as:

(R10)

Here, $T_{capillary}$ and T_{bubble} are the wall thickness of the capillary and microbubble, respectively. $R_{capillary}$ and R_{bubble} are the radius of the microbubble, respectively, and z refers to the axial extension of the microbubble. Equation (R10) was derived from the Gaussian fitting of the outer contour of the prepared microbubble and the constant volume of glass during the fabrication of a hollow capillary into a hollow microbubble (see more details in part 4 of the Supplementary Information (SI)). The strain effect on the microbubble was the force sensing mechanism, indicating that reducing T_{bubble} can effectively improve the displacement and force sensitivities of the mechano-sensor. In practice, considering that T_{bubble} was too thin to constrain the optical WGM and the stability of the preparation process, the thinnest T_{bubble} we fabricated was fixed at 1.5 μm in the following simulations and experiments.

During the fabrication of a glass capillary into a glass microbubble, the volume of glass remained constant. Once the geometric parameters of the capillary (i.e., $R_{capillary}$ and $T_{capillary}$) and T_{bubble} were determined, the radius of the prepared microbubble (R_{bubble}) was obtained using Equation (R10). To optimize the displacement sensitivity (S_D) of the mechano-sensor, $R_{capillary}$

and $T_{capillary}$ values were scanned. The simulation results indicated that the larger the radius and wall thickness of the capillary were, the larger the radius of the prepared microbubble (Figure R19a), and ultimately, the higher the displacement sensitivity of the mechano-sensor (Figure R19b). Because the circular symmetry of the prepared microbubble would deteriorate if R_{bubble} exceeded the limited discharge area of the fiber fusion splicer (FSU-975), R_{bubble} was limited to less than 190 μm (black solid line in Figure R19a-b). Finally, a fused silica capillary with a radius of 84 μm and a wall thickness of 8 μm was selected as the micro-hair (the star label in Figure R19a-b), and a microbubble with a radius of 185 μm and a wall thickness of 1.5 μm was fabricated for mechano-opto-transduction. The outer contour curve of the prepared microbubble (Figure R19c) was fitted with a Gaussian line shape (red line):

$$(R11)$$

where 83.45 μm is basically consistent with the capillary outer radius of 84 μm , and 184.48 μm is basically consistent with the microbubble outer radius of 185 μm . This result indicated the rationality of the process for calculating the radius of the microbubble (Figure R19a and Equation (R10)).

Figure R19. Simulations of the mechano-sensor under an external force F_r ($\varphi = 180^\circ$). **(a)** Relations between R_{bubble} and the geometric parameters of the capillary (i.e., $R_{capillary}$ and $T_{capillary}$). T_{bubble} is fixed at 1.5 μm . **(b)** Relations between the displacement sensitivity (S_D) and the geometric parameters of the capillary. T_{bubble} is fixed at 1.5 μm . The solid black line corresponds to the contour of $R_{bubble} = 190 \mu\text{m}$. The star label corresponds to the $R_{capillary}$ and $T_{capillary}$ values selected for the micro-hair. **(c)** Outer contour of the microbubble fabricated for mechano-opto-transduction was fitted with a Gaussian line shape (red line). The scale bar represents 100 μm .

The theoretical calculation formula of the spring constant of the mechano-sensor can be expressed as:

(R12)

Here, k ($\text{N}\cdot\text{m}^{-1}$) is the spring constant of the mechano-sensor, F_r (N) is the external force loaded on the action point of the micro-hair, and L (m) is the position of the force action point. The spring constant of the mechano-sensor (k) varied from $61.861 \text{ N}\cdot\text{m}^{-1}$ to $0.412 \text{ N}\cdot\text{m}^{-1}$ as L increased from 3 mm to 15 mm (Figure R20).

Figure R20. Relations between the spring constant of the mechano-sensor (k) and the position of the force action point (L). k is inversely proportional to the third power of L .

We discuss this issue in the revised Supplementary Information as:

"To obtain a mechano-sensor with high displacement/force sensitivity, the geometric parameters of the micro-hair (that is, the glass capillary) and microbubble and the mechanical properties of the polymer matrix were optimized through the FEM scanning parameters. The outer contour of the microbubble can be approximately fitted with a Gaussian line shape:

(S1)

where $R_{capillary}$ is the radius of the capillary, R_{bubble} is the radius of the microbubble, and α is related to the expansion length of the microbubbles along the z -axis and is only determined by the discharge area of the fiber fusion splicer. Supplementary Fig. 5a shows a microbubble with a radius of $160 \mu\text{m}$ and a wall thickness of $1.5 \mu\text{m}$ prepared from a fused silica capillary with a radius of $61.5 \mu\text{m}$ and a corroded thickness of $10 \mu\text{m}$. The outer contour of this microbubble was fitted with a Gaussian line shape (red line):

(S2)

where $58.54 \mu\text{m}$ is basically consistent with the capillary outer radius of $61.5 \mu\text{m}$, and $160.14 \mu\text{m}$ is basically consistent with the microbubble outer radius of $160 \mu\text{m}$. When the hollow fused silica capillaries are used to fabricate the hollow microbubbles, the volume of the glass remains constant, so the curve for the inner contour of the microbubble can be formulated as:

(S3)

Supplementary Fig. 5. Parametric scanning of the geometric parameters of the capillary and microbubble. **(a)** The outer contour of a microbubble was fitted with a Gaussian line shape (red line). R_{bubble} and T_{bubble} are $160 \mu\text{m}$ and $1.5 \mu\text{m}$, respectively. $R_{capillary}$ and $T_{capillary}$ are $61.5 \mu\text{m}$ and $10 \mu\text{m}$, respectively. **(b)** Relations between R_{bubble} and the geometric parameters of the capillary (i.e., $R_{capillary}$ and $T_{capillary}$). T_{bubble} was fixed at $1.5 \mu\text{m}$. The solid black line corresponds to the contour of $R_{bubble} = 190 \mu\text{m}$. The star label corresponds to the $R_{capillary}$ and $T_{capillary}$ selected for the micro-hair. **(c)** Simulations of the relations between the displacement sensitivity of the mechano-sensor (S_D) and the position of the applied force (L) under an external force F_r ($\varphi = 180^\circ$). **(d)** Outer contour of the microbubble fabricated for mechano-opto-transduction was fitted with a Gaussian line shape (red line). R_{bubble} and T_{bubble} are $185 \mu\text{m}$ and $1.5 \mu\text{m}$, respectively. $R_{capillary}$ and $T_{capillary}$ are $84 \mu\text{m}$ and $8 \mu\text{m}$, respectively. The scale bars in **(a)** and **(d)** represent $100 \mu\text{m}$.

In conclusion, the curves equation for the inner and outer contours of the microbubble produced by the fiber fusion splicer (FSU-975) can be formulated as:

(S4)

where $T_{capillary}$ and T_{bubble} are the thicknesses of the capillary and microbubble walls, respectively. As shown in Supplementary Fig. 1, the strain effect on the microbubble is the main reason for the shift in the resonance wavelength, indicating that reducing T_{bubble} can effectively improve the displacement/force sensitivity of the mechano-sensor. In practice, considering that T_{bubble} is too thin to constrain the optical WGM and the stability of the preparation process, the thinnest T_{bubble} we could fabricate was 1.5 μm . Therefore, T_{bubble} was fixed at 1.5 μm in the following simulations and experiments.

Based on the principle of volume conservation during the preparation of the microbubbles and **Equation (S4)**, once the geometric parameters of the capillary, including $R_{capillary}$ and $T_{capillary}$, are determined, the radius of the prepared microbubble (R_{bubble}) can be obtained (Supplementary Fig. 5b). $R_{capillary}$ and $T_{capillary}$ were determined to optimize the displacement sensitivity of the mechano-sensor (S_D). The FEM simulation results indicated that the larger the radius and wall thickness of the capillary are, the larger the radius of the prepared microbubble (Supplementary Fig. 5b), and ultimately, the higher the displacement sensitivity of the mechano-sensor (Fig. 2h). Because the circular symmetry of the prepared microbubble will be broken if R_{bubble} exceeds the limited discharge area of the fiber fusion splicer (FSU-975), R_{bubble} was limited to less than 190 μm (black solid line in Supplementary Fig. 5b and Fig. 2h). Finally, a fused silica capillary with a radius of 84 μm and a wall thickness of 8 μm was selected as the micro-hair (the star label in Supplementary Fig. 5b and Fig. 2h), and a microbubble with a radius of 185 μm and a wall thickness of 1.5 μm was fabricated for mechano-opto-transduction. In the FEM simulation, S_D varied with the action point of the external force (L), as shown in Supplementary Fig. 5c. The outer contour curve of the prepared microbubble (Supplementary Fig. 5d) was fitted with a Gaussian line shape (red line):

(S5)

where 83.45 μm is basically consistent with the capillary outer radius of 84 μm , and 184.48 μm is basically consistent with the microbubble outer radius of 185 μm . This result indicates the rationality of the process for calculating the radius of the microbubble (Supplementary Fig. 5b and **Equation (S4)**).

The final version of the bioinspired optical mechano-sensor is shown in Supplementary Fig. 6, which was composed of a thin-walled glass microbubble integrated with a glass micro-

hair that was optically coupled with a fiber taper at the equator of the microbubble resonator. To fit the outer contour curve of the microbubble with a Gaussian line shape and quantitatively control the geometric parameters of the microbubble, the prepared microbubble is approximately 200 μm away from the melting node where the glass capillary is sealed. Notably, such a short distance of 200 μm does not affect the displacement sensitivity of the mechano-sensor. The radius ($R_{\text{capillary}}$) and wall thickness ($T_{\text{capillary}}$) of the glass capillary are 84 μm and 8 μm , respectively. The radius (R_{bubble}) and wall thickness (T_{bubble}) of the glass-based hollow microbubble are 185 μm and 1.5 μm , respectively. The 50 mm long micro-hair collects information about the mechanical stimuli in the environment, such as forces and vibrations, and propagates the information to the microbubble. The thin-walled microbubble configuration is mechanically flexible, allowing the device to transform these environmental mechanical stimuli into shifts in the WGM resonance wavelength and the spectral light signals (i.e., shifts in the dips).

Supplementary Fig. 6. (a) Image of the mechano-sensor. The scale bar represents 10 mm. (b) Optical microscope image of a microbubble integrated with a glass micro-hair standing on its top center. The scale bar represents 200 μm ." on Page S8 (Part 4 in SI).

"The theoretical calculation formula of the spring constant of the mechano-sensor can be expressed as:

$$(S6)$$

Here, k ($\text{N}\cdot\text{m}^{-1}$) is the spring constant of the mechano-sensor, F_r (N) is the external force loaded at the action point of the micro-hair, and L (m) is the position of the force action point. The spring constant of the mechano-sensor (k) varied from 61.861 $\text{N}\cdot\text{m}^{-1}$ to 0.412 $\text{N}\cdot\text{m}^{-1}$ as L increased from 3 mm to 15 mm (Supplementary Fig. 8b).

Supplementary Fig. 8. (a) Full width at half maximum (FWHM) varied intricately as the external force F_r increased from 0.12 mN to 0.72 mN. (b) Relations between the spring constant of the mechano-sensor (k) and the position of the force action point (L). k is inversely proportional to the third power of L ." on Page S14 (Part 5 in SI).

Comment 3: *The authors claim "higher sensitivity" by using this all-optical system when compared with the typical electrical configuration. What is the sensitivity of the electrical sensor? How many times are the all-optical one better? I feel that the authors did not do a good job of reviewing the existing literature.*

Answer 3: We thank the reviewer for the positive suggestions. We have included a comprehensive comparison table in the revised Supplementary Information. Supplementary Table 2 listed the properties of various mechano-sensors, including their configurations, response/recovery times, sensitivities, and detection limits (DLs).

Due to the differences of sensitivity calculation methods, it does not make sense to directly compare the sensitivity between our optical mechano-sensors and electrical mechano-sensors. For example, for electrical mechano-sensors relying on resistance change, the sensitivity can be described as the change in resistance ($\Delta R \cdot R^{-1}$) per unit force (N), unit pressure (Pa), or unit displacement (m). In contrast, for optical mechano-sensors relying on the resonance wavelength shift, the sensitivity can be described as the shift in resonance wavelength (pm) per unit force (N), unit pressure (Pa), or unit displacement (m) [Ref: *Light: Sci. Appl.* **10**, 1–12 (2021)]. Considering this distinction, it is more reasonable for us to compare the detection limits (DLs) between these two systems.

Our all-optical bioinspired mechano-sensor, serving as a proof-of-concept prototype, did not rely on advanced top-down micro- and nano- fabrication techniques. Despite this, when compared to most of the reported electronic whiskers, our novel photonic whisker was not only easy-to-fabricate and cost-effective, but also achieved similar or even faster response and recovery times, along with lower *DL*. Moreover, our device exhibited additional advantages, including certain directionality, improved anti-interference ability, and high-frequency performance. Especially, our experiments demonstrated the effective decoupling of temperature and displacement measurements, highlighting the potential for constructing highly integrated all-optical multiplexing and multifunctional perception systems (see more details in the reply to Question 6/Reviewer #3).

Furthermore, all-optical mechano-sensors that were not hair-like configurations, such as fiber-optic nanomechanical probes fabricated by two-photon polymerization (TPP) nanolithography [Ref: *Light: Sci. Appl.* **10**, 1–12 (2021); *Int. J. Extreme Manuf.* **5**, 015005 (2023)], have been proven to exhibit an extremely high sensitivity and low detection limits. That is, optical versions of mechano-sensors have unique properties and advantages for precision mechanical sensing and detection.

In conclusion, our study demonstrated a novel proof-of-concept prototype of an all-optical multidirectional mechano-sensor inspired by biological mechano-sensitive hair sensilla. We are committed to further advancing this research through ongoing and future work, leveraging advanced micro-and nano-machining technologies to achieve even higher performances.

We have added in the revised manuscript as:

"Supplementary Table 2 presents different force sensing methods that have been reported in recent years. Compared to most reported electronic whiskers, the novel photonic whisker is not only easy-to-fabricate and cost-effective, but also achieves similar or even faster response and recovery times, along with lower *DL*." on Page 16.

We have added in the revised Supplementary Information as:

"**Supplementary Table 2.** Comparison of the sensing performance of different types of force sensors." on Page S23 (Part 8 in SI).

Structures	Platform/Material	Response time/ms	Recovery time/ms	Sensitivity	DL	Refs.
------------	-------------------	------------------	------------------	-------------	----	-------

Dual-modal piezotronic transistor	ZnO nano/microwire	360	360	221.5 N ⁻¹	21 mN	12	
Electrospun micropylamid arrays on-skin devices	Poly(vinylidene fluoride) film	0.8	-	19 kPa ⁻¹	0.05 Pa (13 mN)	13	
Electronic whiskers	Shape memory polymer and gold strain gauges ¹⁴		0.25 ¹⁴				
		Pizeoresistor ¹⁵	16 ¹⁵		1.129 μN ²¹		
		MEMS barometers ¹⁶	37 ¹⁶		3.33 μN ¹⁶		
		Graphite pencil trace ¹⁷	50 ¹⁷	65 ¹⁵	46 Ω·mN ⁻¹ ²³	632 μN ²²	14-23
		CNT-Ag NP film ^{18,19}	90 ¹⁸	76 ¹⁷	80 kPa ⁻¹ ¹⁹	1.31 mN ²³	
		Graphene ²⁰	100 ¹⁹				
		Fluorinated ethylene propylene ²¹	220 ²⁰				
	Giant magnetoresistive sensor ^{22,23}						
Photonic whisker	Silica microbubble	1.24	1.26	3.994 pm·mN ⁻¹	0.9 μN	This work	

Ref:

12. Ge, R., Yu, Q., Zhou, F., Liu, S. & Qin, Y. Dual-modal piezotronic transistor for highly sensitive vertical force sensing and lateral strain sensing. *Nat. Commun.* **14**, 6315 (2023).
13. Zhang, J. et al. Versatile self-assembled electrospun micropylamid arrays for high-performance onskin devices with minimal sensory interference. *Nat. Commun.* **13**, 5839 (2022).
14. Reeder, Jonathan T. et al. 3D, reconfigurable, multimodal electronic whiskers via directed air assembly. *Adv. Mater.* **30**, 1706733 (2018).
15. Wang, Q. et al. Mechano-Sensor for proprioception inspired by ultrasensitive trigger hairs of Venus flytrap. *Cyborg Bionic Syst.*
16. Deer, W., & Pounds, P. E. Lightweight whiskers for contact, pre-contact, and fluid velocity sensing. *IEEE Robot Autom Let.* **4**, 1978-1984 (2019).
17. Hua, Q. et al. Bioinspired Electronic Whisker Arrays by Pencil - Drawn Paper for Adaptive Tactile Sensing. *Adv. Electron. Mater.* **2**, 1600093 (2016).
18. Harada, S. et al. Fully printed, highly sensitive multifunctional artificial electronic whisker arrays integrated with strain and temperature sensors. *ACS nano* **8**, 3921-3927 (2014).
19. Takei, K. et al. Highly sensitive electronic whiskers based on patterned carbon nanotube and silver nanoparticle composite films. *Proc. Natl. Acad. Sci. U. S. A.* **111**, 1703-1707 (2014).

20. Gul, J. Z. et al. Fully 3D printed multi-material soft bio-inspired whisker sensor for underwater-induced vortex detection. *Soft Robot.* **5**, 122-132 (2018).
21. An, J. et al. Biomimetic hairy whiskers for robotic skin tactility. *Adv. Mater.* **33**, 2101891 (2021).
22. Ribeiro, P. et. al. A miniaturized force sensor based on hair-like flexible magnetized cylinders deposited over a giant magnetoresistive sensor. *IEEE Trans. Magn.* **53**, 1-5 (2017).
23. Alfadhel, A. et. al. A magnetoresistive tactile sensor for harsh environment applications. *Sensors* **16**, 650 (2016).

Comment 4: *General note, please, revise the English, some parts are very difficult to understand.*

Answer 4: To improve the clarity of the manuscript, we followed the reviewer's suggestions, and many sentences have been modified.

Additionally, the paper was edited and polished by Springer Nature Author Services (Submission ID: HYKWMGS2). All changes made to the manuscript are indicated using the "Highlight" function with blue color in the revised manuscript and revised Supplementary Information. We trust that these revisions contribute significantly to the overall clarity and readability of the manuscript. We appreciate your consideration of our revised submission and hope that it now meets the required standards for acceptance.

Figure R21. Editing certificate.

Comment 5: *What do authors mean when they state "The photonic mechano-sensor is of the displacement and force sensitivities of $0.87 \text{ pm}\cdot\mu\text{m}^{-1}$ and $1.07 \text{ pm}\cdot\text{mN}^{-1}$ in the axial direction of the "hair", respectively"? How are you defining sensitivity? Please include a paragraph defining sensitivity, because I think the authors are using responsivity instead of sensitivity.*

Answer 5: We thank the reviewer for the positive suggestions. We would like to clarify that the terms "displacement sensitivity (S_D)" and "force sensitivity (S_F)" in our study are determined using established methods based on current published articles [*Light: Sci. Appl.* **10**, 1–12 (2021)].

The displacement sensitivity (S_D) and force sensitivity (S_F) are determined as $S_D = d\lambda/dD$ and $S_F = d\lambda/dF$, respectively. The unit $\text{pm}\cdot\mu\text{m}^{-1}$ represents the ratio of the resonance wavelength shift (pm) of the mechano-sensor to the external displacement of force (μm) applied to the mechano-sensor micro-hair. Similarly, the unit $\text{pm}\cdot\text{mN}^{-1}$ represents the ratio of the resonance wavelength shift (pm) of the mechano-sensor to the external force (mN) applied to the mechano-sensor micro-hair.

We have added "The force and displacement sensitivity of the mechano-sensor can be defined as:

(4)

where F and D are the external force and displacement applied to the mechano-sensor micro-hair, respectively" (Page 8 in revised manuscript).

Comment 6: *The dimensions of the probe are 125 μm outer diameter, 100 μm inner diameter and 36 mm length, why it a hollow glass capillary? Can it be solid? Again, a study of the geometrical parameters needed. I also wonder if the capillary has any kind of polymer coating, because it seems from figure S1 that it is coated by polyimide, which greatly affects the mechanical properties of the device.*

Answer 6: We thank the reviewer for the positive suggestions. We provided the following clarifications and explanations regarding the hollow glass capillary used in our study:

6.1 For choice of a hollow glass capillary:

Answer 6.1: The choice of a hollow glass capillary was crucial for the fabrication process of our mechano-sensor, especially in producing thin-walled hollow microbubbles. The fabrication procedures for the microbubbles were shown in Figure R22. The glass microbubble was the result of spherical expansion of the hollow capillary. First, to prepare a thin-walled hollow microbubble, a fused silica capillary (Zhengzhou INNOSEP Scientific, China) needed to be corroded by hydrofluoric acid until the corroded wall thickness was 4~10 μm . Second, a section of a silica capillary was placed at the electrodes of an optical fiber fusion splicer. When the electrodes were discharged, one end of the silica capillary was sealed. Third, the outer coating layer of the silica capillary was burnt with an alcohol burner and wiped with microscope lens paper. The silica capillary was placed at the center of the electrodes. The other end of the silica capillary was connected to a Teflon tube and a syringe. While the syringe was pushed and the electrodes discharge, the capillary silica capillary was heated, melted and expanded, and finally, a hollow core microbubble structure with a long mechano-sensor micro-hair was fabricated. Therefore, the presence of gas/liquid channels within the hollow capillary was essential for this process.

Additionally, the strain effect on the microbubble was the main reason for the shift in resonance wavelength, and the force sensitivity of the mechano-sensor increased with decreasing wall thickness. If a solid capillary were used instead, it would result in a solid microsphere-based mechano-sensor with minimal strain effect and low mechanical sensitivity.

Figure R22. Fabrication procedures for the microbubble resonator. (a) A silica capillary was

used, and (b) the end of the capillary was sealed by electrode discharge. (c) The coating was burned, and the capillary was cleaned with microscope lens paper. (d) The capillary was connected to a syringe and placed under the electrodes. (e) The electrodes were discharged, and air was blown into the capillary, causing it to swell. (f) Finally, the prepared microbubble resonator with a long mechano-sensor micro-hair was obtained.

We have added the fabrication procedures for the microbubble resonator on Page S24 in the SI.

6.2 For optimization of the capillaries' geometrical parameters:

Answer 6.2: As addressed in our response to Question 2/Reviewer #2, the optimization of force/displacement sensitivity for the mechano-sensor was conducted through FEM modeling, scanning the radius ($R_{capillary}$) and corroded wall thickness ($T_{capillary}$) of the capillary. Because the volume of glass was constant during the fabrication of hollow capillaries into hollow microbubbles, once the geometric parameters of the capillary (i.e., $R_{capillary}$ and $T_{capillary}$) and T_{bubble} were determined, the radius of the prepared microbubble (R_{bubble}) could be derived. When the wall thickness of the microbubble was fixed at $1.5 \mu\text{m}$, the larger the radius and wall thickness of the capillary were, the larger the radius of the fabricated microbubble (Figure R23a), and ultimately, the higher the displacement sensitivity of the mechano-sensor (Figure R23b). Due to the limited discharge area of the fiber fusion splicer, the circular symmetry of the prepared microbubble would deteriorate if its radius (R_{bubble}) exceeded the discharge area (black solid line in Figure R23a). Based on the FEM results and actual fabrication conditions, a fused silica capillary with a radius of $84 \mu\text{m}$ and a wall thickness of $8 \mu\text{m}$ was selected as the micro-hair (the star label in Figure R24a-b), and a microbubble with a radius of $185 \mu\text{m}$ and a wall thickness of $1.5 \mu\text{m}$ was fabricated for mechano-opto-transduction (Figure R23c).

Figure R23. Simulations of the mechano-sensor under an external force F_r ($\varphi = 180^\circ$). (a) Relations between R_{bubble} and the geometric parameters of the capillary (i.e., $R_{capillary}$ and $T_{capillary}$). T_{bubble} is fixed at $1.5 \mu\text{m}$. (b) Relations between the displacement sensitivity (S_D) and

the geometric parameters of the capillary. T_{bubble} is fixed at 1.5 μm . The solid black line corresponds to the contour of $R_{bubble} = 190 \mu\text{m}$. The star label corresponds to the $R_{capillary}$ and $T_{capillary}$ values selected for the micro-hair. (c) Optical microscope image of a microbubble integrated with a glass micro-hair standing on its top center. The scale bar represents 200 μm . We have added in the revised Supplementary Information as:

"To obtain a mechano-sensor with high displacement/force sensitivity, the geometric parameters of the micro-hair (that is, the glass capillary) and microbubble and the mechanical properties of the polymer matrix were optimized through the FEM scanning parameters. The outer contour of the microbubble can be approximately fitted with a Gaussian line shape:

(S1)

where $R_{capillary}$ is the radius of the capillary, R_{bubble} is the radius of the microbubble, and α is related to the expansion length of the microbubbles along the z -axis and is only determined by the discharge area of the fiber fusion splicer. Supplementary Fig. 5a shows a microbubble with a radius of 160 μm and a wall thickness of 1.5 μm prepared from a fused silica capillary with a radius of 61.5 μm and a corroded thickness of 10 μm . The outer contour of this microbubble was fitted with a Gaussian line shape (red line):

(S2)

where 58.54 μm is basically consistent with the capillary outer radius of 61.5 μm , and 160.14 μm is basically consistent with the microbubble outer radius of 160 μm . When the hollow fused silica capillaries are used to fabricate the hollow microbubbles, the volume of the glass remains constant, so the curve for the inner contour of the microbubble can be formulated as:

(S3)

Supplementary Fig. 5. Parametric scanning of the geometric parameters of the capillary and microbubble. **(a)** The outer contour of a microbubble was fitted with a Gaussian line shape (red line). R_{bubble} and T_{bubble} are $160 \mu\text{m}$ and $1.5 \mu\text{m}$, respectively. $R_{capillary}$ and $T_{capillary}$ are $61.5 \mu\text{m}$ and $10 \mu\text{m}$, respectively. **(b)** Relations between R_{bubble} and the geometric parameters of the capillary (i.e., $R_{capillary}$ and $T_{capillary}$). T_{bubble} was fixed at $1.5 \mu\text{m}$. The solid black line corresponds to the contour of $R_{bubble} = 190 \mu\text{m}$. The star label corresponds to the $R_{capillary}$ and $T_{capillary}$ selected for the micro-hair. **(c)** Simulations of the relations between the displacement sensitivity of the mechano-sensor (S_D) and the position of the applied force (L) under an external force F_r ($\varphi = 180^\circ$). **(d)** Outer contour of the microbubble fabricated for mechano-opto-transduction was fitted with a Gaussian line shape (red line). R_{bubble} and T_{bubble} are $185 \mu\text{m}$ and $1.5 \mu\text{m}$, respectively. $R_{capillary}$ and $T_{capillary}$ are $84 \mu\text{m}$ and $8 \mu\text{m}$, respectively. The scale bars in **(a)** and **(d)** represent $100 \mu\text{m}$.

In conclusion, the curves equation for the inner and outer contours of the microbubble produced by the fiber fusion splicer (FSU-975) can be formulated as:

$$(S4)$$

where $T_{capillary}$ and T_{bubble} are the thicknesses of the capillary and microbubble walls, respectively. As shown in Supplementary Fig. 1, the strain effect on the microbubble is the main reason for the shift in the resonance wavelength, indicating that reducing T_{bubble} can effectively improve the

displacement/force sensitivity of the mechano-sensor. In practice, considering that T_{bubble} is too thin to constrain the optical WGM and the stability of the preparation process, the thinnest T_{bubble} we could fabricate was 1.5 μm . Therefore, T_{bubble} was fixed at 1.5 μm in the following simulations and experiments.

Based on the principle of volume conservation during the preparation of the microbubbles and **Equation (S4)**, once the geometric parameters of the capillary, including $R_{capillary}$ and $T_{capillary}$, are determined, the radius of the prepared microbubble (R_{bubble}) can be obtained (Supplementary Fig. 5b). $R_{capillary}$ and $T_{capillary}$ were determined to optimize the displacement sensitivity of the mechano-sensor (S_D). The FEM simulation results indicated that the larger the radius and wall thickness of the capillary are, the larger the radius of the prepared microbubble (Supplementary Fig. 5b), and ultimately, the higher the displacement sensitivity of the mechano-sensor (Fig. 2h). Because the circular symmetry of the prepared microbubble will be broken if R_{bubble} exceeds the limited discharge area of the fiber fusion splicer (FSU-975), R_{bubble} was limited to less than 190 μm (black solid line in Supplementary Fig. 5b and Fig. 2h). Finally, a fused silica capillary with a radius of 84 μm and a wall thickness of 8 μm was selected as the micro-hair (the star label in Supplementary Fig. 5b and Fig. 2h), and a microbubble with a radius of 185 μm and a wall thickness of 1.5 μm was fabricated for mechano-opto-transduction. In the FEM simulation, S_D varied with the action point of the external force (L), as shown in Supplementary Fig. 5c. The outer contour curve of the prepared microbubble (Supplementary Fig. 5d) was fitted with a Gaussian line shape (red line):

(S5)

where 83.45 μm is basically consistent with the capillary outer radius of 84 μm , and 184.48 μm is basically consistent with the microbubble outer radius of 185 μm . This result indicates the rationality of the process for calculating the radius of the microbubble (Supplementary Fig. 5b and **Equation (S4)**).

The final version of the bioinspired optical mechano-sensor is shown in Supplementary Fig. 6, which was composed of a thin-walled glass microbubble integrated with a glass micro-hair that was optically coupled with a fiber taper at the equator of the microbubble resonator. To fit the outer contour curve of the microbubble with a Gaussian line shape and quantitatively control the geometric parameters of the microbubble, the prepared microbubble is approximately 200 μm away from the melting node where the glass capillary is sealed. Notably, such a short distance of 200 μm does not affect the displacement sensitivity of the mechano-sensor. The radius ($R_{capillary}$) and wall thickness ($T_{capillary}$) of the glass capillary are 84 μm and 8 μm ,

respectively. The radius (R_{bubble}) and wall thickness (T_{bubble}) of the glass-based hollow microbubble are 185 μm and 1.5 μm , respectively. The 50 mm long micro-hair collects information about the mechanical stimuli in the environment, such as forces and vibrations, and propagates the information to the microbubble. The thin-walled microbubble configuration is mechanically flexible, allowing the device to transform these environmental mechanical stimuli into shifts in the WGM resonance wavelength and the spectral light signals (i.e., shifts in the dips).

Supplementary Fig. 6. (a) Image of the mechano-sensor. The scale bar represents 10 mm. (b) Optical microscope image of a microbubble integrated with a glass micro-hair standing on its top center. The scale bar represents 200 μm ." on Page S8 (Part 4 in SI).

6.3 For polyimide coating:

Answer 6.3: Regarding the polyimide coating on the capillary, it is essential to clarify that, prior to microbubble preparation, the polyimide coating on the surface of the microbubble section on the capillary was removed. The polyimide coating could be removed from the entire capillary surface or only selectively from the microbubble surface. To analyze how the polyimide coating affected the displacement sensitivity, FEM modeling was performed. As shown in Figure R24, when the capillary was not coated in part of the polymer matrix, the proportion of polyimide coating had little effect on the displacement sensitivity of the mechano-sensor. However, when the proportion of polyimide coating present was large enough that the capillary was coated in the part of the polymer matrix, the displacement sensitivity was greatly reduced. Therefore, it was crucial to ensure that the capillary in the polymer matrix was not coated after encapsulation to maintain displacement sensitivity.

Figure R24. Effect of the coating removal ratio on the displacement sensitivity of mechano-sensor.

We have added in the revised Supplementary Information as:

"As shown in Supplementary Fig. 14, prior to the preparation of a microbubble from a fused silica capillary, the coating on the capillary surface needs to be removed. The coating can be removed from the entire capillary surface or only selectively from the microbubble surface. To analyze the effect of the coating removal ratio on the displacement sensitivity, an FEM model was used. As shown in Supplementary Fig. 7, when the capillary had no coating in part of the polymer matrix, the proportion of coating had little effect on the displacement sensitivity of the mechano-sensor. However, when the proportion of coating present was large enough that the capillary was coated in the part of the polymer matrix, the displacement sensitivity was greatly reduced. Therefore, it is necessary to ensure that the capillary in the polymer matrix is not coated after encapsulation." on Page S12 (Part 4 in SI).

Comment 7: *I guess the n_{eff} will be affected mainly by the deformation of the gap in between the microbubble and the fiber, therefore, the mechanical parameters (Young's modulus and Poisson ratio) of the polymer are just as important as the optical parameters (n and k). Could you please provide those numbers? I also think that a parametric sweep of the material properties would also be very useful. What is the ideal material?*

Answer 7: We thank the reviewer for the positive suggestions. According to the reviewer's suggestions, we conducted a comprehensive set of experiments and simulations to clarify this issue.

7.1 Experimental proof of the force sensing mechanism:

Answer 7.1: First, we prepared a solid microsphere-based mechano-sensor and a hollow microbubble-based mechano-sensor. Both mechano-sensors were encapsulated in the same polymer matrix (MY-133-V2000), and an external force (F_r , $\varphi = 180^\circ$) was applied at the same axial position. The measured force sensitivity of the solid microsphere-based mechano-sensor was very low (Figure R25). In addition, we prepared four microbubble-based mechano-sensors encapsulated in four different commercial polymer matrices (Table R2). To study the effect of the polymer matrix on the force sensitivity of the mechano-sensor, the geometric parameters of the microbubbles and the capillaries were the same for the four mechano-sensors, and the external forces (F_r , $\varphi = 180^\circ$) were applied at the same axial position. As shown in Figure R25, the mechano-sensor encapsulated in MY-133-V2000 had the highest sensitivity. The mechano-sensor encapsulated in polymer matrices with smaller or larger elastic modulus had lower force sensitivities. Because both the microsphere-based mechano-sensor and the microbubble-based mechano-sensor encapsulated in the polymer matrix with the smallest elastic modulus were not sensitive to external force, the change in the gap between the microbubble and the fiber was not the main source of the change in the optical path.

The above experimental results indicated that the strain effect on the microbubble was the main source of the change in the optical path, that is, the shift in the resonance wavelength. Under the same external force, the strain effect on the solid microsphere was much weaker than that on the hollow microbubble. For the four microbubble-based mechano-sensors, the strain effect on the microbubble was not significant if the elastic modulus of the polymer matrix encapsulating the mechano-sensor was small, and the microbubble cannot deform if the elastic modulus of the polymer matrix encapsulating the mechano-sensor was too large. Therefore, the strain effect on the microbubble was strongest with an intermediate value of the elastic modulus.

Therefore, n_{eff} was affected mainly by the strain effect on the microbubble, rather than by the deformation of the gap between the microbubble and the fiber. Even so, it was important and necessary to optimize the mechanical parameters of the polymer matrix.

Table R2. Comparison of the properties of commercial polymer matrices.

Product name	RI at 950 nm	Elastic Modulus (MPa)	Poisson's Ratio	Hardness Shore
MY-132-A	1.322	0.4	0.495	30 A
MY-132-V15K	1.322	Very low	-	7

MY-133-V2000	1.329	5.2	0.41	70 A
NOA1348	1.348	158	-	30 D

Figure R25. Experimental results of the force sensitivity comparison among microsphere-based mechano-sensor encapsulated with MY-133-V2000 and microbubble-based mechano-sensors encapsulated with four different polymer matrices (MY-132-V15K, MY-132-A, NOA1348, and MY-133-V2000).

We have added in revised Supplementary Information as:

"To analyze the main reason for the shift in the resonance wavelength, a solid microsphere-based mechano-sensor and a hollow microbubble-based mechano-sensor were prepared. Both mechano-sensors were packaged in the same polymer matrix (MY-133-V2000), and external forces (F_r , $\varphi = 180^\circ$) were applied at the same axial position. As shown in Supplementary Fig. 1, the solid microsphere-based mechano-sensor had very little force sensitivity. In addition, microbubble-based mechano-sensors encapsulated with four different commercial polymer matrices (**Supplementary Table 1**) were prepared. To study the effect of the polymer matrix on the force sensitivity of the mechano-sensor, the geometric parameters of the microbubbles and the micro-hairs were the same in all four mechano-sensors, and the external forces (F_r , $\varphi = 180^\circ$) were applied at the same axial position. As shown in Supplementary Fig. 1, the mechano-sensor encapsulated with MY-133-V2000 had the highest sensitivity. The mechano-sensors encapsulated with polymer matrices with smaller or larger elastic moduli had lower force sensitivity.

The above experimental results indicate that the strain effect on the microbubble is the main reason for the shift in the resonance wavelength. Under the same external force, the strain effect on the solid microsphere is much weaker than that on the hollow microbubble. For the four microbubble-based mechano-sensors, the strain effect on the microbubble is not significant if the elastic modulus of the polymer matrix encapsulating the mechano-sensor is small, and the microbubble cannot deform if the elastic modulus of the polymer matrix encapsulating the mechano-sensor is too large. Therefore, the strain effect on the microbubble is strongest at an intermediate value of the elastic modulus." on Page S2 (Part 2 in SI).

7.2 Parametric sweep of the material properties:

Answer 7.2: FEM simulations were performed to optimize the mechanical parameters of the polymer matrix. The geometric parameters of the mechano-sensor were set as fixed values. The optimized radius $R_{capillary}$ and optimized wall thickness $T_{capillary}$ of the capillary were 84 μm and 8 μm , respectively. The optimized radius R_{bubble} and optimized wall thickness T_{bubble} of the microbubble were 185 μm and 1.5 μm , respectively. The external force F_r ($\varphi = 180^\circ$) was applied along the r -axis at the same axial position on the micro-hair. As shown in Figure R26, when the elastic modulus of the polymer matrix was 5 MPa, the strain effect was the strongest, and the displacement sensitivity of the mechano-sensor was the highest. A larger or smaller elastic modulus would reduce the displacement sensitivity of the mechano-sensor, which corresponded to the experimental results shown in Figure R25. Poisson's ratio had little influence on the displacement sensitivity. Thus, referring to the common commercial UV-crosslinked low-refractive-index polymers (Table R2), MY-133-V2000 polymer with an elastic modulus E of 5.2 MPa and a Poisson's ratio ν of 0.41 was selected for the optimized polymer matrix.

Figure R26. FEM simulations of the relations between the displacement sensitivity (S_D) and

Poisson's ratio (ν) and elastic modulus (E) of the polymer matrix under an external force F_r ($\varphi = 180^\circ$).

We have added and revised in our revised manuscript as:

"To obtain a mechano-sensor with high displacement sensitivity, the geometric parameters of the micro-hair and microbubble and the mechanical properties of the polymer matrix were optimized through the FEM scanning parameters (Fig. 2h and i, Supplementary Fig. 5-7 and Table 1 in the SI)." on Page 10.

"The displacement sensitivity of the mechano-sensor is the highest when the elastic modulus of the polymer encapsulating the mechano-sensor is 5 MPa (Fig. 2i). An increase or decrease in the magnitude of the elastic modulus will reduce the displacement sensitivity of the mechano-sensor, which corresponds to the experimental results in Supplementary Fig. 1." on Page 10.

"MY-133-V2000 polymer with an elastic modulus of 5.2 MPa and a Poisson's ratio of 0.41 was selected as the polymer matrix" on Page 10.

Comment 8: *I wonder why all the curves in Fig. 2 are linear, even when the authors state "Once an external force F_z was applied on the end of the mechano-sensor hair with a displacement of $20 \mu\text{m}$, the microbubble began to deform, and the major stress was located at the thin wall of the microbubble (Figure 2d)." Thus, it seems that there is a not shown threshold in the system.*

Answer 8: We thank the reviewer for the positive suggestions.

In reality, the mechano-sensor system had no threshold. The microbubble started to deform as soon as an external force F_z was applied to the mechano-sensor micro-hair, regardless of how much force was applied. We emphasized that small displacements or forces, which caused invisible wavelength shifts in the experiment, can indeed induce deformations in the microbubble. However, due to the inherent detection limit (DL) of our experimental system, these small displacements or forces may not be discernible in the presented data. The detection limit (DL), which is given by:

(R13)

where S is the displacement/force sensitivity of the mechano-sensor and σ is the standard

deviation of the resulting spectral variation. σ can be approximated by:

(R14)

where $\Delta\lambda$ is the full width at half maximum (FWHM) of the resonance dip, which is related to the Q factor. SNR is the signal-to-noise ratio.

Furthermore, to demonstrate the displacement perception capability of our device, we chose to set the displacement of the stepper motor at 20 μm . Notably, the displacement can be set at various values, provided they exceed the displacement DL and stepper motor resolution.

Comment 9: *Since a critical parameter is the evanescent light coupling, I am not sure that is justified to remove the fiber from the simulations, especially when the displacements are on the order of nanometers. I am pretty sure that the relative change in gap is modified if you include the fiber in FEM simulations.*

Answer 9: We thank the reviewer for the constructive suggestions. According to the reviewer's suggestions, we extended our FEM model to include the fiber taper and subsequently verified that the mechano-sensor's directionality to external force F_r was related to the fiber taper.

According to the simulation and experimental results described in the response to Question 7/Reviewer #2, the strain effect on the microbubble was the main reason for the shift in resonance wavelength. We conducted additional simulations and experiments to investigate the impact of the fiber taper on the performance of mechano-sensor. The FEM simulation and experimental results of the mechanical perception to the external force F_r in the vertical plane of the micro-hair were shown in Figure R27a-b. Our findings indicated that the presence of the fiber taper indeed influenced the mechanical circular symmetry of the microbubble's equatorial cross-section. Consequently, the strain effect and spectral responses of the WGM microbubble were different for forces F_r applied from different directions, contributing to the mechano-sensor's directionality.

Specifically, for a microbubble with a circular symmetric equatorial cross-section, the field distributions of the radial strain and strain-induced effective refractive index change should be antisymmetric (i.e., $dR/R \approx 0$, $dn_{\text{eff}}/n_{\text{eff}} \approx 0$), regardless of which direction of F_r was applied to the micro-hair. However, the fiber taper amplified the radial strain at the coupling position (that

is, $\varphi = 0^\circ$ in the microbubble's equatorial cross-section), resulting in the radial strain dR/R in the microbubble's equatorial cross-section depending on the radial strain at the coupling position of the microbubble.

The dR/R , $dn_{\text{eff}}/n_{\text{eff}}$ (strain-induced effective refractive index change in the microbubble's equatorial cross-section), and S_D (displacement sensitivity of the mechano-sensor) varied with the direction of force F_r (φ), as shown in Figure R27a. dR/R was greater than $dn_{\text{eff}}/n_{\text{eff}}$ and dominated $d\lambda/\lambda$ (shift in the resonance wavelength). dR/R had a maximum positive value (that is, maximum resonance wavelength redshift) when an external force F_r was applied in the $\varphi = 180^\circ$ direction, a maximum negative value (that is, maximum resonance wavelength blueshift) when an external force F_r was applied in the $\varphi = 0^\circ$ direction, and values of approximately zero (that is, no resonance wavelength shift) when external forces F_r was applied in the $\varphi = 90^\circ$ and 270° directions (see more details in the response to Question 17/Reviewer #1 and Figure R14-R17). Moreover, the simulation results of mechano-sensing directionality correspond well with the experimental results (Figure R27b)

We believed that these additional simulations and experiments, taking into account the fiber taper, enhance the accuracy of our theoretical model in describing the mechano-sensor's behaviors.

Figure R27. (a) Simulations of the relationship between the directions of the external force (φ) and dR/R , $dn_{\text{eff}}/n_{\text{eff}}$ and S_D . (b) Experimental results of the relationship between the directions of the external force (φ) and the displacement (red curve) and force sensitivities (blue curve) of the mechano-sensor.

We have added in revised manuscript as:

"A finite element method (FEM) model was used to analyze the responses of the mechano-sensor when external forces F_r were applied to the micro-hair. The label L in Fig. 2a represents the length of the micro-hair subjected to the external forces F_r . The direction of the external force

F_r applied to the mechano-sensor in the r - φ plane is shown in Fig. 2a. The elastic modulus E and the Poisson's ratio ν of the polymer matrix were set as 5.2 MPa and 0.41, respectively. As a stimulus input in the simulation, an external force F_r (blue arrow) was applied to the micro-hair along the r -axis of the mechano-sensor (Fig. 2b). The direction of the external force F_r was set as $\varphi = 180^\circ$, and the corresponding displacement was set as 2 μm . The stress field distribution in the equatorial cross-section of the mechano-sensor is shown in Fig. 2c. The microbubble and fiber taper moved in the $\varphi = 180^\circ$ direction. The stress on the microbubble wall was over 2-3 orders of magnitude higher than that on the polymer matrix; therefore, the stress effect of the polymer matrix was neglected. The radial strain field distribution (Fig. 2d) and the strain components (i.e., ε_x , ε_y and ε_z) along the microbubble's equatorial cross-section can be directly obtained in the FEM simulation. According to Equation (3), by using the reported values of $p_{11} = 0.131$ and $p_{12} = 0.26$ for fused silica, the field distribution of the strain-induced effective refractive index change was derived, as shown in Fig. 2e. The radial strain field distribution was negative in the range of $\varphi = 90^\circ$ to 270° in the equatorial cross-section but positive in the range of $\varphi = 270^\circ$ to 450° (namely, 90°) in the equatorial cross-section. The radial strain field distribution along the microbubble's equatorial cross-section had a maximum negative value at $\varphi = 180^\circ$ (that is, the same direction of F_r), a maximum positive value at $\varphi = 0^\circ$ (that is, the opposite direction of F_r), and values of approximately zero at $\varphi = 90^\circ$ and 270° (that is, the direction perpendicular to F_r). The radial strain dR/R and the effective refractive index change $dn_{\text{eff}}/n_{\text{eff}}$ in the microbubble's equatorial cross-section were the averages of their field distribution integrals, which are indicated in Fig. 2d and e, respectively. In theory, for a microbubble with a circular symmetric equatorial cross-section, the field distributions of the radial strain and strain-induced effective refractive index change should be antisymmetric (i.e., $dR/R \approx 0$, $dn_{\text{eff}}/n_{\text{eff}} \approx 0$), regardless of the direction of the force F_r applied to the micro-hair. However, the presence of the fiber taper breaks the geometrical circular symmetry of the microbubble's equatorial cross-section, amplifying the radial strain at the coupling position (that is, $\varphi = 0^\circ$ in the microbubble's equatorial cross-section), resulting in $dR/R > 0$. Moreover, dR/R was greater than $dn_{\text{eff}}/n_{\text{eff}}$, inducing a redshift in the resonance wavelength.

When the direction of the force F_r changes, it can be inferred that the dR/R depends on the radial strain at the coupling position of the microbubble. dR/R had a maximum positive value (that is, the maximum resonance wavelength redshift) when an external force F_r was applied in the $\varphi = 180^\circ$ direction, a maximum negative value (that is, the maximum resonance wavelength

blueshift) when an external force F_r was applied in the $\varphi = 0^\circ$ direction, and values of approximately zero (that is, no resonance wavelength shift) when an external force F_r was applied in the $\varphi = 90^\circ$ and 270° directions (see more details in Supplementary Fig. 3 and 4 in the SI). The dR/R , dn_{eff}/n_{eff} , and S_D (displacement sensitivity of the mechano-sensor) varied with the direction of force F_r (Fig. 2f). dR/R was greater than dn_{eff}/n_{eff} and dominated $d\lambda/\lambda$." on Page 8.

Comment 10: *The comparison in between the microbubble and the microsphere is not necessary in the main text, it can be summarized as the microsphere is stiffer. Again, it is important to make the calculation of the spring constant of the whole system.*

Answer 10: We thank the reviewer for the positive suggestions.

10.1 Comparison of microbubble and microsphere:

Answer 10.1: In the main text, the comparison between the microbubble and microsphere was valuable. Both experimental findings and simulation results demonstrated that the solid microsphere-based mechano-sensor was insensitive to external forces. This observation contributed to the understanding that the sensing mechanism of the hollow microbubble-based mechano-sensor is due to the strain effect on the microbubble. In addition, the detailed comparisons presented in Figure R28 highlighted that the radial strain dR/R was larger than the strain-induced effective refractive index change dn_{eff}/n_{eff} , and the dependence of $d\lambda/\lambda$ on the displacement (that is, force) was essentially linear. Therefore, we have retained the comparison between the microbubble- the microsphere-based mechano-sensors in the revised manuscript to convey this valuable information.

However, as pointed out by the reviewer, the force-insensitive nature of the microsphere-based mechano-sensor can be attributed to its stiffness. We also recognize the importance of conciseness and clarity in the manuscript. If the reviewer still believes that the detailed comparison between microbubble and microsphere is not essential for the main text, we are open to revising the manuscript accordingly.

Figure R28. Simulations of the mechano-sensor under external forces. **(a)** Comparisons of dR/R and dn_{eff}/n_{eff} between the microbubble-based mechano-sensor and the microsphere-based mechano-sensor under different external displacements of forces F_r ($\varphi = 180^\circ$). **(b)** Comparisons of dR/R and dn_{eff}/n_{eff} between the microbubble-based mechano-sensor and the microsphere-based mechano-sensor under different external displacements of forces F_z .

10.2 Spring constant of the mechano-sensor:

Answer 10.2: The theoretical calculation formula of the spring constant of the mechano-sensor can be expressed as:

$$(R15)$$

Here, k ($\text{N}\cdot\text{m}^{-1}$) is the spring constant of the mechano-sensor, F_r (N) is the external force loaded on the action point of the micro-hair, and L (m) is the position of the force action point. The spring constant of the mechano-sensor (k) varied from $61.861 \text{ N}\cdot\text{m}^{-1}$ to $0.412 \text{ N}\cdot\text{m}^{-1}$ as L increased from 3 mm to 15 mm (Figure R29).

Figure R29. Relations between the spring constant of the mechano-sensor (k) and the position of the force action point (L). k is inversely proportional to the third power of L .

We have revised in revised manuscript as:

"As a comparison, the hollow microbubble was replaced with a solid microsphere of the same size; in this case, the dR/R and dn_{eff}/n_{eff} values of the microsphere-based mechano-sensor were much smaller than those of the microbubble-based mechano-sensor, as shown in Fig. 2g. The results in Fig. 2g also indicate that for the displacement ($\varphi = 180^\circ$) range considered, dR/R was larger than dn_{eff}/n_{eff} , and the dependence of $d\lambda/\lambda$ on the displacement (i.e., force) was essentially linear." on Page 10.

"The radial strain dR/R and the strain-induced effective refractive index change dn_{eff}/n_{eff} in the entire equatorial cross-section of the microbubble are the averages of their field distribution integrals (Fig. 4d and e) and linearly varied with increasing displacement (Fig. 4f)." on Page 13.

Comment 11: *How do you calculate the force in Fig. 3? Are you using the Hooke's law? Explain the transformation of displacement into force, the calculation and/or the performed simulations, as well as the approximations that have been made.*

Answer 11: We thank the reviewer for the positive suggestions. We appreciate the opportunity to provide a detailed explanation of the methodology employed in our study.

In our experimental setup, the static force applied to the mechano-sensor micro-hair was given by a commercial force sensor. The commercial force sensor was mounted on a 3D stepper motor stage and equipped with a digital force gauge. The object applying the external force was driven by the stepper motor, and the wavelength shift and the static force change were recorded by a computer and a digital force gauge in real time, respectively. The displacement value was obtained by the stepper motor. The inset showed the position of the force action point (L).

While we did not explicitly use Hooke's law in our calculations, we can theoretically verify that the principles underlying our experiments align with Hooke's law.

Figure R30. Measurement schematic diagram of the external force (F_r) applied to the mechano-sensor micro-hair in the r - ϕ plane. Inset: Front view of the mechano-sensor.

We have added in the revised Supplementary Information as:

"To measure the directional characteristics of the mechano-sensor with external force (F_r), the mechano-sensor mounted on a rotating device, which drove the mechano-sensor to rotate 360° around the z -axis. The displacement and force sensitivities of the mechano-sensor were measured at 10° intervals, and the directional diagram of the mechano-sensor was finally obtained (Supplementary Fig. 9). The static force applied to the mechano-sensor micro-hair was given by a commercial force sensor. The commercial force sensor was mounted on a 3D stepper motor stage and equipped with a digital force gauge. The object applying the external force was driven by the stepper motor, and the wavelength shift and the static force change were recorded by a computer and a digital force gauge in real time, respectively. The displacement value was obtained by the stepper motor. The inset shows the position of the force action point (L)." on Page S15 (Part 5 in SI).

Comment 12: *I wonder if the authors have considered the frequency domain analysis of the sensor device. I am pretty sure that the study of the resonance of the "hair" can give more information, e.g., about vector force fields. In that sense I recommend to take a look to the existing literature of the field.*

Answer 12: We thank the reviewer for the positive suggestions. The frequency domain analysis has indeed been applied in the vibration analysis of nanowire and cantilever beam systems. For example, Wei, Z. *et al.* used the whisker sensor with the frequency domain analysis method to discriminate the texture [2019 IEEE International Conference on Cyborg and Bionic Systems

(CBS), Munich, Germany, 2019, pp. 222-227.]; Braakman and Poggio used a nanowire cantilever to demonstrate the sensitive vectorial force and mass detection [*Nanotechnology*, **30**, 332001 (2019).].

When navigate forward and probe the object surface, rodents actively move their whiskers at a frequency of $\sim 5\text{--}12$ Hz [*J Neurosci* **10**, 2638–2648, (1990); *Somatosens Mot Res* **18**, 211–222, (2001).]. In our study, we focused on investigating the mechano-sensor's responses to low-frequency stimulation (ranging from 0.05 Hz to 16 Hz) to align with the typical frequency range of biological whisker movements during object exploration. Given that most environmental objects were in a static state (means being in a low-frequency vibration status), analyzing frequencies beyond this range was not a primary focus in our current work scenario.

However, we sincerely appreciate the reviewer's suggestion regarding frequency domain analysis and acknowledge its potential value in exploring vector force fields. We will certainly consider incorporating this perspective in our future work.

Comment 13: *Why do you have a red shifting at the beginning of the quadruped experiment? It seems to be related with temperature because it reaches a stationary value.*

Answer 13: We thank the reviewer for the positive suggestions. Our experiments have demonstrated that the microbubble resonator was sensitive to temperature variations. Because the thermo-optical coefficient of the polymer matrix was negative and larger than that of glass wall, we observed a redshift in the WGM resonance wavelength with decreasing temperature.

As pointed out by the reviewer, the redshift observed at the beginning of the quadruped experiment was related to the temperature decrease in the surrounding environment. To reduce the influence of temperature changes, we kept the mechano-sensor away from heat sources, such as human interference and laser fluctuations. Subsequently, in Figure R31, we conducted the quadruped experiment once the surrounding environment had stabilized, resulting in a flattened wavelength shift.

It was crucial to emphasize that the interference caused by temperature fluctuations did not impact the performance of our device because the device can decouple the temperature interference signal from the mechanical signal very well (see more details in our reply to Question 6/Reviewer #3).

Figure R31. Wavelength shifts of the mechano-sensor caused by various actions, including remaining stationary, stepping forward, squatting down and getting up, hitting an obstacle at $\varphi = 180^\circ$ and 0° , turning right, and stepping back. The background areas in gray, orange, yellow, green, blue, pink, and purple represent the mechano-sensor response when remaining stationary, stepping forward, squatting down and getting up, hitting an obstacle at $\varphi = 180^\circ$, hitting an obstacle at $\varphi = 0^\circ$, turning right, and stepping back, respectively.

Comment 14: *Since it seems that there is a transient state due to the temperature, I wonder what is the power of the laser that has been used in the experiments? Have the authors tried using different laser powers?*

Answer 14: We thank the reviewer for the positive suggestions. The power of the laser was maintained as small as $4.16 \mu\text{W}$ to prevent thermo-optic effects. We also attempted using different laser powers. Notably, the thermo-optic coefficient of the polymer matrix was not only negative but also greater than the positive thermo-optic coefficient of fused silica. Excessive laser power led to a blueshift in the resonance wavelength, creating interference with the wavelength shift induced by mechanical stimuli. In addition, the low laser power needed by the mechano-sensor will contribute to the preparation of highly integrated devices in the future. We have added "The input laser power was maintained as low as $4.16 \mu\text{W}$ to prevent opto-thermal effects." (Page 11 in revised manuscript).

In summary, we have addressed all comments made by the Reviewer #2. Accordingly, the manuscript and supplementary materials have been carefully revised. We hope these answers and

efforts yield a more complete and insightful manuscript, and we are also very grateful for the reviewer's essential contribution in motivating these improvements.

Reply to Reviewer #3

This manuscript demonstrates an all-optical mechano-sensor for the detection of forces with different direction information. This work is meaningful and the mentioned device performs relatively good performances, however, it does not deserve a recommendation to be published in Nature Communications.

We would like to thank the reviewer for the constructive comments and are encouraged by the reviewer's positive remarks. We understand the reviewer's viewpoint regarding the manuscript not meeting the publication criteria of *Nature Communications*. However, we are committed to improving the quality of our work to align with the journal's standards. We are happy to address all the comments below. We thank the reviewer for motivating those improvements.

Comment 1: *The footprint of the proposed device is very big, including laser, fiber, sensor, etc., which is not so-called minimalist and highly impedes its scale-up.*

Answer 1: We thank the reviewer for the comments. In our manuscript, the term "minimalism" referred to the configuration and principle of our mechano-sensor. To provide clarity, the construction of the device involved a thin-walled glass microbubble with a glass micro-hair, coupled with a fiber taper at the equator of the microbubble. This 3D-integrated structure was embedded within a matrix of ultraviolet (UV)-crosslinked low-refractive-index polymer.

Our bioinspired photonic mechano-sensor, serving as a proof-of-concept prototype, validated the feasibility of this novel hair-like configuration both theoretically and physically. Although the current experimental setups and methodology may not align well with the minimalist concept, we foresee that future developments in integrated photonics will greatly facilitate the integration of various components, addressing scalability concerns. Much like the early development of the crystal field-effect transistor (FET), which initially had an awkward and fragile configuration but eventually evolved into the most integrated basic component in chips, we anticipate a similar progress for the integration of the all-optical mechano-sensor system.

Comment 2: *The signal decoupling of the simultaneously applied mechanical stimuli is seemingly*

impossible or difficult.

Answer 2: We thank the reviewer for the positive suggestions. In the case of multiple static forces being simultaneously applied, the mechano-sensitive hair sensilla, including both biological and biomimetic versions, encountered challenges in decoupling these static forces due to the inherent configuration of the device. However, we would like to emphasize that this limitation is specific to static forces. When it came to multiple dynamic forces, the mechano-sensitive hair sensilla could effectively decouple these mechanical signals. This distinction was crucial in understanding the capabilities of our device under various mechanical stimuli. Furthermore, our all-optical mechano-sensor exhibited an excellent ability to decouple other physical signals (i.e., temperature) from the mechanical signal (see more details in our reply to Question 6/Reviewer #3).

Comment 3: *What is the Young's modulus of the UV matrix? Is it soft or hard? If it is hard, the deformation of the glass microbubble is not easy. If it is soft, the position of fiber taper will change during the force application process. The related optimization and discussion are suggested.*

Answer 3: We thank the reviewer for the constructive suggestions. To answer this question, we performed a series of experiments to analyze the main source of the change in the optical path (i.e., dR and dn_{eff}). As pointed out by the reviewer, the fiber and the microbubble moved slightly together. However, the main factor influencing the wavelength shift was the strain effect on the microbubble.

3.1 Experimental proof of the force sensing mechanism:

Answer 3.1: First, we prepared a solid microsphere-based mechano-sensor and a hollow microbubble-based mechano-sensor. Both mechano-sensors were encapsulated in the same polymer matrix (MY-133-V2000), and an external force (F_r , $\varphi = 180^\circ$) was applied at the same axial position. The measured force sensitivity of the solid microsphere-based mechano-sensor was very low (Figure R32). In addition, we also prepared four microbubble-based mechano-sensors encapsulated in four different commercial polymer matrices (Table R3). To study the effect of the polymer matrix on the force sensitivity of the mechano-sensor, the geometric

parameters of the microbubbles and the capillaries were the same for the four mechano-sensors, and the external forces (F_r , $\varphi = 180^\circ$) were applied at the same axial position. As shown in Figure R32, the mechano-sensor encapsulated in MY-133-V2000 had the highest sensitivity. The mechano-sensor encapsulated in polymer matrices with smaller or larger elastic modulus had lower force sensitivities. Because both the microsphere-based mechano-sensor and the microbubble-based mechano-sensor encapsulated in the polymer matrix with the smallest elastic modulus were insensitive to external force, the change in the gap between the microbubble and the fiber was not the main source of the change in the optical path.

The above experimental results indicated that the strain effect on the microbubble was the main source of the change in the optical path, that is, the shift in resonance wavelength. Under the same external force, the strain effect on the solid microsphere was much weaker than that on the hollow microbubble. For the four microbubble-based mechano-sensors, the strain effect on the microbubble was not significant if the elastic modulus of the polymer matrix encapsulating the mechano-sensor was small, and the microbubble cannot deform if the elastic modulus of the polymer matrix encapsulating the mechano-sensor was too large. Therefore, the strain effect on the microbubble was strongest with an intermediate value of the elastic modulus. Even so, it was important and necessary to optimize the mechanical parameters of the polymer matrix.

Table R3. Comparison of the properties of commercial polymer matrices.

Product name	RI at 950 nm	Elastic Modulus (MPa)	Poisson's Ratio	Hardness Shore
MY-132-A	1.322	0.4	0.495	30 A
MY-132-V15K	1.322	Very low	-	7
MY-133-V2000	1.329	5.2	0.41	70 A
NOA1348	1.348	158	-	30 D

Figure R32. Experimental results of the force sensitivity comparison among microsphere-based mechano-sensor encapsulated with MY-133-V2000 and microbubble-based mechano-sensors encapsulated with four different polymer matrices (MY-132-V15K, MY-132-A, NOA1348, and MY-133-V2000).

We discuss this issue in the revised Supplementary Information as:

"To analyze the main reason for the shift in the resonance wavelength, a solid microsphere-based mechano-sensor and a hollow microbubble-based mechano-sensor were prepared. Both mechano-sensors were packaged in the same polymer matrix (MY-133-V2000), and external forces (F_r , $\varphi = 180^\circ$) were applied at the same axial position. As shown in Supplementary Fig. 1, the solid microsphere-based mechano-sensor had very little force sensitivity. In addition, microbubble-based mechano-sensors encapsulated with four different commercial polymer matrices (Supplementary Table 1) were prepared. To study the effect of the polymer matrix on the force sensitivity of the mechano-sensor, the geometric parameters of the microbubbles and the micro-hairs were the same in all four mechano-sensors, and the external forces (F_r , $\varphi = 180^\circ$) were applied at the same axial position. As shown in Supplementary Fig. 1, the mechano-sensor encapsulated with MY-133-V2000 had the highest sensitivity. The mechano-sensors encapsulated with polymer matrices with smaller or larger elastic moduli had lower force sensitivity." on Page S2 (Part 2 in SI).

"The above experimental results indicate that the strain effect on the microbubble is the main reason for the shift in the resonance wavelength. Under the same external force, the strain effect on the solid microsphere is much weaker than that on the hollow microbubble. For the four microbubble-based mechano-sensors, the strain effect on the microbubble is not significant if the

elastic modulus of the polymer matrix encapsulating the mechano-sensor is small, and the microbubble cannot deform if the elastic modulus of the polymer matrix encapsulating the mechano-sensor is too large. Therefore, the strain effect on the microbubble is strongest at an intermediate value of the elastic modulus." on Page S3 (SI).

3.2 Optimization of the polymer matrix:

Answer 3.2: FEM simulations were also performed to optimize the mechanical parameters of the polymer matrix. The geometric parameters of the mechano-sensor were set as fixed values. The optimized radius $R_{capillary}$ and optimized wall thickness $T_{capillary}$ of the capillary were 84 μm and 8 μm , respectively. The optimized radius R_{bubble} and optimized wall thickness T_{bubble} of the microbubble were 185 μm and 1.5 μm , respectively. The external force F_r ($\varphi = 180^\circ$) was applied along the r -axis at the same axial position of the micro-hair. As shown in Figure R33, when the elastic modulus of the polymer matrix was 5 MPa, the strain effect was the strongest, and the displacement sensitivity of the mechano-sensor was the highest. A larger or smaller elastic modulus would reduce the displacement sensitivity of the mechano-sensor, which corresponded to the experimental results in Figure R32. Poisson's ratio had little influence on the displacement sensitivity. Thus, referring to the common commercial UV-crosslinked low-refractive-index polymers (Table R3), the MY-133-V2000 polymer with elastic modulus $E = 5.2$ MPa and Poisson's ratio $\nu = 0.41$ was selected as the optimized polymer matrix.

Figure R33. FEM simulations of the relations between the displacement sensitivity (S_D) and Poisson's ratio (ν) and elastic modulus (E) of the polymer matrix under an external force F_r ($\varphi = 180^\circ$).

We discuss this issue in the revised Supplementary Information as:

"Finally, the mechanical properties of the polymer matrix, including the elastic modulus E and

Poisson's ratio ν , were optimized with fixed geometric parameters of micro-hair and microbubble. As shown in Fig. 2i, when the elastic modulus of the polymer matrix was 5 MPa, the strain effect was the strongest, and the displacement sensitivity of the mechano-sensor was the highest. An increase or decrease in the magnitude of the elastic modulus would reduce the displacement sensitivity of the mechano-sensor, which corresponded to the experimental results in Supplementary Fig. 1. The reason may be that the strain effect on the microbubble was not significant if the elastic modulus of the polymer matrix was too small, and the microbubble could not be deformed if the elastic modulus of the polymer matrix was too large. Poisson's ratio had relatively little influence on the displacement sensitivity. Thus, considering the commercial UV-crosslinked low-refractive-index polymers data (Supplementary Table 1), the MY-133-V2000 polymer with an elastic modulus E of 5.2 MPa and Poisson's ratio ν of 0.41 was selected as the polymer matrix." on Page S13 (Part 4 in SI).

Fig. 2. Simulations of the mechano-sensor under an external force F_r . **(a)** Schematic illustration of the experimental device (left) and FEM model of the mechano-sensor (right) under an external force F_r ($\varphi = 180^\circ$). A cylinder was used to apply the force in the FEM model. Inset: Top view of the mechano-sensor. **(b)** Schematic illustration of the deformation of the flexible microbubble under an external force F_r . Negative and positive radial strains were observed in the same and opposite directions of F_r , respectively. The fiber taper amplified the strain of the microbubble at the coupling position. **(c)** Stress field distribution in the r - φ plane under an external force F_r ($\varphi = 180^\circ$). **(d)** and **(e)** Field distributions of the radial strain and strain-induced effective refractive index change along the microbubble's equatorial cross-section under an external force F_r ($\varphi = 180^\circ$). The dR/R and dn_{eff}/n_{eff} values in the microbubble's equatorial cross-section were the averages of their field distribution integrals. **(f)** Relations between the direction of the external

force F_r and dR/R , dn_{eff}/n_{eff} , and the displacement sensitivity (S_D). (g) Comparisons of dR/R and dn_{eff}/n_{eff} between the microbubble-based mechano-sensor and the microsphere-based mechano-sensor under different external displacements ($\varphi = 180^\circ$). (h) Relations between S_D and the capillary size (i.e., radius ($R_{capillary}$) and wall thickness ($T_{capillary}$)) under an external force F_r ($\varphi = 180^\circ$). The solid black line corresponds to the contour of $R_{bubble} = 190 \mu\text{m}$. The star label corresponds to the $R_{capillary}$ and $T_{capillary}$ selected for the micro-hair. (i) Relations between S_D and the mechanical properties of the polymer matrix (i.e., Poisson's ratio (ν) and elastic modulus (E)) under an external force F_r ($\varphi = 180^\circ$).

Comment 4: *What is the response behavior of the device at high frequencies more than 1 Hz? The device shows a response time of 20 ms, which means it can respond at a relatively high frequency around 50 Hz. The related data is suggested. And a table of comprehensive comparison between this proposed device and others (reported or commercial ones, if possible) is suggested, in terms of key sensing index, configuration, energy consumption, etc.*

Answer 4: We thank the reviewer for the constructive suggestions. There are two methods to investigate the durability and response of the mechano-sensor: tracking the resonance wavelength shift of the WGM ($\Delta\lambda$) and monitoring the intensity change in the WGM (ΔI) (Figure R34a).

When tracking $\Delta\lambda$ (Figure R34b), the output wavelength of the tunable laser was tuned by an external triangular wave, covering the WGM spectrum to extract the resonance wavelength. The response time and recovery time of the frequency response by applying a square waveform with a frequency of 0.8 Hz to the piezo actuator were calculated as 18 ms and 20 ms, respectively (Figure R34c). The mechano-sensor can detect a square waveform with a frequency up to 16 Hz, as demonstrated in Figure R34b. However, it was inaccurate to measure the response time by tracking $\Delta\lambda$. Due to the limitation of the maximum scanning frequency of the external triangular wave (below 40 Hz), the minimum time interval between data points in Figure R34c was 25 ms, exceeding the calculated response time of 18 ms. Therefore, tracking $\Delta\lambda$ did not accurately capture the response time, which was likely less than 18 ms.

Figure R34. (a) Schematic diagram for tracking the wavelength shift ($\Delta\lambda$) of the WGM and changes in the WGM intensity (ΔI). (b) Higher frequency responses of the mechano-sensor by tracking the wavelength shifts. Square signals were applied to the piezo actuator at frequencies of 1 Hz, 2 Hz, 4 Hz, 8 Hz, and 16 Hz, respectively. (c) Response time and recovery time of the mechano-sensor, determined by tracking the wavelength shifts. (d) Low frequency responses of the mechano-sensor by monitoring the intensity changes. Square signals were applied to the piezo actuator at frequencies of 0.2 Hz, 0.4 Hz, 0.8 Hz, 1.6 Hz, 3.2 Hz, and 6.4 Hz. (e) Higher frequency responses of the mechano-sensor by monitoring the intensity changes. Square signals were applied to the piezo actuator at frequencies of 10 Hz, 20 Hz, 40 Hz, 80 Hz, 160 Hz, and 320 Hz, respectively. (f) Response time and recovery time of the mechano-sensor, determined by monitoring the intensity changes in the WGM.

In contrast, when monitoring ΔI (Figure R34a), there was no need to scan the entire WGM spectrum. Instead, the output wavelength remained at the rising or falling edge of the

resonance dip, and the response time of the mechano-sensor can be accurately obtained by monitoring ΔI through the oscilloscope. The accurate response and recovery times of 1.24 ms and 1.26 ms, respectively, were obtained by applying a square waveform with a frequency of 0.2 Hz to the piezo actuator (Figure R34f). The mechano-sensor can detect a square waveform with a frequency up to 320 Hz, as demonstrated in Figure R34d and e.

Notably, while monitoring ΔI can accurately capture the response time of the mechano-sensor, this method was strongly dependent on the selection of the start position and the nonlinearity change in the intensity. Thus, it was more reasonable for us to investigate the perception capability of the mechano-sensor using the wavelength shift approach.

We have added the relevant discussion in the revised Supplementary Information as:

"There are two methods to investigate the durability and response of the mechano-sensor: tracking the resonance wavelength shift of the WGM ($\Delta\lambda$) and monitoring the intensity change in the WGM (ΔI).

When tracking $\Delta\lambda$ (Fig. 5d and Supplementary Fig. 13b), the output wavelength of the tunable laser is tuned by an external triangular wave, covering the WGM spectrum to extract the resonance wavelength. The response time and recovery time of the frequency response by applying a square waveform with a frequency of 0.8 Hz to the piezo actuator were calculated as 18 ms and 20 ms, respectively (Supplementary Fig. 13c). The mechano-sensor can detect a square waveform with a frequency up to 16 Hz, as demonstrated in Supplementary Fig. 13b. However, it is inaccurate to measure the response time by tracking $\Delta\lambda$. Due to the limitation of the maximum scanning frequency of the external triangular wave (below 40 Hz), the minimum time interval between data points in Supplementary Fig. 13c is 25 ms, exceeding the calculated response time of 18 ms. Therefore, tracking $\Delta\lambda$ does not accurately capture the response time, which is likely less than 18 ms.

In contrast, when monitoring ΔI (Supplementary Fig. 13a), there is no need to scan the entire WGM spectrum. Instead, the output wavelength remains at the rising or falling edge of the resonance dip, and the response time of the mechano-sensor can be accurately obtained by monitoring ΔI through the oscilloscope. The accurate response and recovery times of 1.24 ms and 1.26 ms, respectively, were obtained by applying a square waveform with a frequency of 0.2 Hz to the piezo actuator (Supplementary Fig. 13f). The mechano-sensor can detect a square waveform with a frequency up to 320 Hz, as demonstrated in Supplementary Fig. 13d and e.

2Notably, while monitoring ΔI can accurately capture the response time of the mechano-sensor, this method is strongly dependent on the selection of the start position and the nonlinearity change in the intensity. Thus, it is more reasonable for us to investigate the perception capability of the mechano-sensor using the wavelength shift approach." on Page S2 (Part 8 in SI).

We have added the relevant discussion in the revised manuscript as:

"The durability test results of the mechano-sensor for forces with higher frequencies (1 Hz, 2 Hz, 4 Hz, 8 Hz, and 16 Hz) are shown in Supplementary Fig. 13b. To investigate the response of the mechano-sensor, monitoring the intensity change (ΔI) in the WGM signal was conducted. As shown in Supplementary Fig. 13d-f, the response and recovery times were just 1.24 ms and 1.26 ms, respectively, surpassing the highest limit of vibration sense for most biological MSHS. Moreover, these results are encouraging compared with the performance of existing mechano-sensors with comparable sizes and effective mechano-sensing ranges but no directionality^{44,45}." on Page 14.

"Supplementary Table 2 presents different force sensing methods that have been reported in recent years. Compared to most reported electronic whiskers, the novel photonic whisker is not only easy-to-fabricate and cost-effective, but also achieves similar or even faster response and recovery times, along with lower DL ." on Page 16.

"**Supplementary Table 2.** Comparison of the sensing performance of different types of force sensors." on Page S23 (Part 8 in SI).

Structures	Platform/Material	Response time/ms	Recovery time/ms	Sensitivity	DL	Refs.
Dual-modal piezotronic transistor	ZnO nano/microwire	360	360	221.5 N ⁻¹	21 mN	12
Electrospun micropyr amid arrays on-skin devices	Poly(vinylidene fluoride) film	0.8	-	19 kPa ⁻¹	0.05 Pa (13 mN)	13
Electronic whiskers	Shape memory polymer and gold strain gauges ¹⁴ Pizeoresistor ¹⁵	0.25 ¹⁴	65 ¹⁵	46 Ω·mN ⁻¹ ²³	1.129 μN ²¹	14-23
		16 ¹⁵	76 ¹⁷	80 kPa ⁻¹ ¹⁹	3.33 μN ¹⁶	
		37 ¹⁶			632 μN ²²	

	MEMS barometers ¹⁶	50 ¹⁷			1.31 mN ²³	
	Graphite pencil trace ¹⁷	90 ¹⁸				
	CNT-Ag NP film ^{18,19}	100 ¹⁹				
	Graphene ²⁰	220 ²⁰				
	Fluorinated ethylene propylene ²¹					
	Giant magnetoresistive sensor ^{22,23}					
Photonic whisker	Silica microbubble	1.24	1.26	3.994 pm·mN ⁻¹	0.9 μN	This work

Ref:

12. Ge, R., Yu, Q., Zhou, F., Liu, S. & Qin, Y. Dual-modal piezotronic transistor for highly sensitive vertical force sensing and lateral strain sensing. *Nat. Commun.* **14**, 6315 (2023).
13. Zhang, J. et al. Versatile self-assembled electrospun micropylamids for high-performance onskin devices with minimal sensory interference. *Nat. Commun.* **13**, 5839 (2022).
14. Reeder, Jonathan T. et al. 3D, reconfigurable, multimodal electronic whiskers via directed air assembly. *Adv. Mater.* **30**, 1706733 (2018).
15. Wang, Q. et al. Mechano-Sensor for proprioception inspired by ultrasensitive trigger hairs of Venus flytrap. *Cyborg Bionic Syst.*
16. Deer, W., & Pounds, P. E. Lightweight whiskers for contact, pre-contact, and fluid velocity sensing. *IEEE Robot Autom Let.* **4**, 1978-1984 (2019).
17. Hua, Q. et al. Bioinspired Electronic Whisker Arrays by Pencil-Drawn Paper for Adaptive Tactile Sensing. *Adv. Electron. Mater.* **2**, 1600093 (2016).
18. Harada, S. et al. Fully printed, highly sensitive multifunctional artificial electronic whisker arrays integrated with strain and temperature sensors. *ACS nano* **8**, 3921-3927 (2014).
19. Takei, K. et al. Highly sensitive electronic whiskers based on patterned carbon nanotube and silver nanoparticle composite films. *Proc. Natl. Acad. Sci. U. S. A.* **111**, 1703-1707 (2014).
20. Gul, J. Z. et al. Fully 3D printed multi-material soft bio-inspired whisker sensor for underwater-induced vortex detection. *Soft Robot.* **5**, 122-132 (2018).
21. An, J. et al. Biomimetic hairy whiskers for robotic skin tactility. *Adv. Mater.* **33**, 2101891 (2021).
22. Ribeiro, P. et al. A miniaturized force sensor based on hair-like flexible magnetized cylinders deposited over a giant magnetoresistive sensor. *IEEE Trans. Magn.* **53**, 1-5 (2017).
23. Alfadhel, A. et al. A magnetoresistive tactile sensor for harsh environment applications. *Sensors* **16**, 650 (2016).

Comment 5: For the durability test, the voltage amplitudes are suggested to be replaced with the practical displacements.

Answer 5: We thank the reviewer for the positive suggestions. The voltage amplitudes have been replaced with the practical displacements in Figure 5e in our revised manuscript (Page 35).

Fig. 5. (e) Displacement response of the mechano-sensor. The 0.1 Hz square signals were applied to the piezo actuator with displacements of 0.054 μm , 0.099 μm , 0.152 μm , 0.249 μm , and 0.316 μm .

Comment 6: How about the anti-interference capability of the device to surrounding perturbations, in consideration of its high sensitivity to tiny mechanical stimuli?

Answer 6: We thank the reviewer for the constructive suggestions.

6.1 Decoupling of temperature and displacement measurements:

Answer 6.1: To demonstrate the anti-interference capability (i.e., temperature interference) of the mechano-sensor, temperature was introduced as a variable during the displacement measurement. To construct the two-dimensional (2D) sensing matrix $M_{T,D}$, the sensitivities of two WGMs are measured to decouple the temperature and displacement measurements. The 2D sensitivity matrix $M_{T,D}$ is defined as:

(R16)

Here, S_T and S_D are the temperature and displacement sensitivities, respectively. The wavelength shifts of the two WGMs ($\Delta\lambda_1$ and $\Delta\lambda_2$) induced by temperature (ΔT) and displacement changes (ΔD) are defined as:

(R17)

Therefore, the changes in temperature and displacement can be solved by the following matrix:

(R18)

First, Figure R35a shows the evolution of the transmission spectra as the temperature increased from 27.7°C to 28.2°C at 0.1°C intervals, while the displacement was kept at 0 μm . The temperature sensitivities (S_{T1} and S_{T2}) of the two tracked WGMs were $-16.495 \text{ pm}\cdot\text{C}^{-1}$ and $-21.665 \text{ pm}\cdot\text{C}^{-1}$, respectively (Figure R35b). The greater negative thermo-optical effect of the polymer matrix and the weaker positive thermo-optical effect of the glass wall both lead to the blueshift in the resonance wavelength with increasing temperature.

Figure R35. (a) Transmission spectra evolution of the mechano-sensor as the temperature increased from 27.7°C to 28.2°C. (b) The temperature sensitivities of two tracked WGMs. (c) Transmission spectra evolution of the mechano-sensor as the displacement increased from 0 μm to 500 μm. (d) The displacement sensitivities of two tracked WGMs. (e) Transmission spectra evolution of the mechano-sensor at different temperatures and displacements (0 μm and 27.7°C, 20 μm and 27.8°C, 40 μm and 28.0°C, 240 μm and 28.1°C, and 280 μm and 28.2°C). (f) Temperature/displacement comparisons between the calculated and measured results.

Second, Figure R35c shows the evolution of the transmission spectra as the displacement increased from 0 μm to 500 μm at 100 μm intervals ($L = 5 \text{ mm}$, $\varphi = 180^\circ$), while the temperature was kept at 27.7°C. The displacement sensitivities (S_{D1} and S_{D2}) of the two above tracked WGMs were $0.0193 \text{ pm} \cdot \mu\text{m}^{-1}$ and $0.0158 \text{ pm} \cdot \mu\text{m}^{-1}$ (Figure R35d).

Because different WGMs have various energy ratios in the polymer matrix and glass wall, mode 1 and mode 2 have distinct temperature and displacement sensitivities. Mode 2 has more energy leakage than mode 1 in the polymer matrix, so mode 2 has a larger temperature sensitivity. However, mode 2 has a lower energy proportion in the glass wall with the largest strain effect, so the displacement sensitivity of mode 2 is smaller.

Finally, the evolution of the transmission spectra as the temperature and displacement changed concurrently was demonstrated (Figure R35e). By tracking the two above tracked WGMs and applying Equation (R18), the displacement and temperature measurements could be decoupled, as shown in Figure R35f. The derived root mean square errors of the displacement and temperature were 20.15 μm and 0.027 $^{\circ}\text{C}$, respectively. The results show that the mechano-sensor has good displacement-temperature decoupling stability. Moreover, compared to electrical mechano-sensors, the optical mechano-sensor has new features, such as all-optical multifunctional perception system.

We have added the relevant discussion in the revised Supplementary Information as:

"To demonstrate the anti-interference capability (i.e., temperature interference) of the mechano-sensor, temperature was introduced as a variable during the displacement measurement¹⁰. To construct the two-dimensional (2D) sensing matrix $M_{T,D}$, the sensitivities of two WGMs are measured to decouple the temperature and displacement measurements. The 2D sensitivity matrix $M_{T,D}$ is defined as:

(S7)

Here, S_T and S_D are the temperature and displacement sensitivities, respectively. The wavelength shifts of the two WGMs ($\Delta\lambda_1$ and $\Delta\lambda_2$) induced by temperature (ΔT) and displacement changes (ΔD) are defined as:

(S8)

Therefore, the changes in temperature and displacement can be solved by the following matrix:

(S9)

First, Supplementary Fig. 10a shows the evolution of the transmission spectra as the temperature increased from 27.7 $^{\circ}\text{C}$ to 28.2 $^{\circ}\text{C}$ at 0.1 $^{\circ}\text{C}$ intervals, while the displacement was kept at 0 μm . The temperature sensitivities (S_{T1} and S_{T2}) of the two tracked WGMs were -16.495 $\text{pm}\cdot^{\circ}\text{C}^{-1}$ and -21.665 $\text{pm}\cdot^{\circ}\text{C}^{-1}$, respectively (Supplementary Fig. 10b). The greater negative thermo-optical effect of the polymer matrix and the weaker positive thermo-optical effect of the glass wall both lead to the blueshift in the resonance wavelength with increasing temperature.

Second, Supplementary Fig. 10c shows the evolution of the transmission spectra as the displacement increased from 0 μm to 500 μm at 100 μm intervals ($L = 5 \text{ mm}$, $\varphi = 180^\circ$), while the temperature was kept at 27.7°C. The displacement sensitivities (S_{D1} and S_{D2}) of the two above tracked WGMs were 0.0193 $\text{pm}\cdot\mu\text{m}^{-1}$ and 0.0158 $\text{pm}\cdot\mu\text{m}^{-1}$ (Supplementary Fig. 10d).

Because different WGMs have various energy ratios in the polymer matrix and glass wall, mode 1 and mode 2 have distinct temperature and displacement sensitivities. Mode 2 has more energy leakage than mode 1 in the polymer matrix, so mode 2 has a larger temperature sensitivity. However, mode 2 has a lower energy proportion in the glass wall with the largest strain effect, so the displacement sensitivity of mode 2 is smaller.

Finally, the evolution of the transmission spectra as the temperature and displacement changed concurrently was demonstrated (Supplementary Fig. 10e). By tracking the two above tracked WGMs and applying **Equation S9**, the displacement and temperature measurements could be decoupled, as shown in Supplementary Fig. 10f. The derived root mean square errors of the displacement and temperature were 20.15 μm and 0.027°C, respectively. The results show that the mechano-sensor has good displacement-temperature decoupling stability. Moreover, compared to electrical mechano-sensors, the optical mechano-sensor have new features, such as all-optical multifunctional perception system." on Page S15 (Part 6 in SI).

6.2 Resistance to saline-alkali environment interference:

Answer 6.2: Moreover, to demonstrate that the mechano-sensor can work properly in saline and alkaline environments, the mechano-sensor was immersed in sea water. The sea water was prepared according to ASTM standard D1141-98 (2013, American Society for Testing Materials). An external force F_r ($\varphi = 0^\circ$, $L = 12 \text{ mm}$) was applied on the micro-hair, with the displacement increasing from 0 μm to 90 μm in steps of 30 μm (Figure R36a). The calculated displacement sensitivity was -0.00255 $\text{pm}\cdot\mu\text{m}^{-1}$ (Figure R36b). The result was comparable to that in Figure R27. Therefore, the force sensitivity is -3.131 $\text{pm}\cdot\text{mN}^{-1}$ according to Figure R27, and the calculated external force F_r is shown in Figure R38c.

Figure R36. (a) Transmission spectra evolution of the mechano-sensor as the displacement increased from $0 \mu\text{m}$ to $90 \mu\text{m}$ in the sea water environment. (b) Displacement sensitivity of the mechano-sensor immersed in sea water. (c) Relation between the calculated F_r and the displacement.

We have added the relevant discussion in the revised Supplementary Information as:

"Moreover, to demonstrate that the mechano-sensor can work properly in saline and alkaline environments, the mechano-sensor was immersed in sea water. The sea water was prepared according to ASTM standard D1141-98 (2013, American Society for Testing Materials)¹¹. An external force F_r ($\varphi = 0^\circ$, $L = 12 \text{ mm}$) was applied on the micro-hair, with the displacement increasing from $0 \mu\text{m}$ to $90 \mu\text{m}$ in steps of $30 \mu\text{m}$ (Supplementary Fig. 11a). The calculated displacement sensitivity was $-0.00255 \text{ pm} \cdot \mu\text{m}^{-1}$ (Supplementary Fig. 11b). The result was comparable to that in Fig. 3e. Therefore, the force sensitivity is $-3.131 \text{ pm} \cdot \text{mN}^{-1}$ according to Fig. 3e, and the calculated external force F_r is shown in Supplementary Fig. 11c." on Page S18 (Part 6 in SI).

In summary, we have addressed all comments made by the Reviewer #3. Accordingly, the manuscript and supplementary materials have been carefully revised. We hope these answers and efforts yield a more complete and insightful manuscript, and we are also very grateful for the reviewer's essential contribution in motivating these improvements.

REVIEWERS' COMMENTS

Reviewer #1 (Remarks to the Author):

The authors have convincingly addressed all the questions and the manuscript has largely improved in clarity and quality of the information provided to the readers.

Reviewer #2 (Remarks to the Author):

I had the opportunity to review the manuscript a few months ago, and I can see how the overall quality of the manuscript has greatly improved. The authors have addressed all my previous comments, and now I can support its publication.

Reviewer #3 (Remarks to the Author):

the authors have done a good work in this 2nd revision. and I think the MS is ready for acceptance.